# Structural basis of CSN-mediated SCF deneddylation

Shan Ding[1,2,5], Julie A. Clapperton [1,5], Märt-Erik Mäeots[1],
Simone Kunzelmann [3], Mohammed Shaaban[1,4] & Radoslav I. Enchev [1] ✉

Cullin-RING ligases (CRLs) are the largest family of E3 ligases, with ubiquitination activity dynamically regulated by neddylation and deneddylation by the COP9 signalosome (CSN). CSN-mediated deneddylation not only deactivates CRLs but also enables substrate receptor exchange. Although CSN is a promising drug target, the structural basis underlying its catalytic mechanism remains unclear. Here, we use cryo-electron microscopy (cryo-EM) to uncover distinct functional states of CSN-CRL (SCF) complexes, capturing key intermediates of the deneddylation cycle. We visualise an autoinhibited docking state and a catalytic intermediate in which CSN5 Ins-1 loop, RBX1 RING and neddylated Cullin WHB domains are repositioned for isopeptide cleavage. We further resolve four dissociation intermediates that define the stepwise release of CSN from its product, with RBX1 RING stabilising key interactions. Additionally, our structures locate CSNAP within a CSN3-CSN8 groove. Together, our study provides a mechanistic model for CSN function and informs the rational design of CSN-targeted therapeutics.

Cullin-RING ligases (CRLs) constitute the largest family of ubiquitin E3 ligases, accounting for nearly half of all E3 ligases in human cells[1]. CRLs regulate diverse biological processes like cell cycle progression, signal transduction, stress responses, and developmental pathways[2,3]. The large numbers and functional diversity of CRLs arise from their modular architecture, which enables dynamic substrate selection via exchangeable substrate receptor (SR) modules[1]. Given their central role in cellular homeostasis, CRLs have also been co-opted for targeted protein degradation, where small molecules such as molecular glues and proteolysis-targeting chimeras hijack their substrate recognition mechanisms to degrade disease-associated proteins[4,5].

The activity and specificity of all CRLs are tightly regulated by three master regulators: neddylation[6], deneddylation by the COP9 signalosome (CSN)[7], and SR exchange facilitated by SR exchange factors, Cullin-associated NEDD8-dissociated proteins 1 and 2 (CAND1 & CAND2)[8-10]. CRL activation is driven by the covalent attachment of the ubiquitin-like protein NEDD8 to the winged-helix B (WHB) domain of Cullins[11], increasing catalytic efficiency by several orders of magnitude

while preventing CAND1 from engaging the complex[12,13]. CSN reverses this modification by removing NEDD8, restoring CRLs to an inactive state, preventing CRLs from auto-ubiquitination, and facilitating SR exchange, resetting the ligase for substrate recognition[14,15]. This activation-inactivation cycle ensures that ubiquitination activity is tuned to substrate availability.

CSN is a conserved multi-subunit complex that orchestrates CRL regulation through its metalloprotease subunit CSN5[16]. The CSN complex consists of six proteasome lid–CSN–initiation factor 3 (PCI) domain-containing subunits (CSN1-4, CSN7, CSN8), which form a horseshoe-shaped scaffold, and an MPN domain heterodimer (CSN5-CSN6) harbouring the catalytic site[17]. An additional subunit, CSNAP, has been identified in biochemical studies as a modulator of CSN, yet its precise structural role remains uncharacterised, as it has not been visualised in prior CSN structures[18-21].

In its inactive state, CSN5 adopts an autoinhibited conformation, where the Ins-1 loop blocks substrate access to the $Zn^{2+}$-coordinated active site[17,22]. Upon binding a neddylated CRL, CSN undergoes a series

[1]The Visual Biochemistry Laboratory, The Francis Crick Institute, London, UK. [2]Department of Chemistry, King's College London, London, UK. [3]Structural Biology Science Technology Platform, The Francis Crick Institute, London, UK. [4]Department of Infectious Disease, Faculty of Medicine, Imperial College London, London, UK. [5]These authors contributed equally: Shan Ding, Julie A. Clapperton. ✉e-mail: radoslav.enchev@crick.ac.uk

of conformational changes, repositioning CSN5 for catalysis[14,15,23–25]. CSN is hypothesised to engage CRLs in a sequential manner, first docking onto the Cullin scaffold through CSN2[23]. The interaction is further stabilised by inositol hexakisphosphate (IP6), a small metabolite that enhances CSN-CRL affinity through electrostatic interactions between CSN2 and the Cullin basic canyon[26–28]. Following initial docking, CSN4 and CSN2 clamp the Cullin C-terminal domain (CTD) and RBX1, ensuring that the WHB domain is positioned correctly for NEDD8 cleavage by CSN5[14,15,23–25].

Despite extensive structural and functional studies of CSN-CRL complexes[14,15,23–25], key mechanistic questions remain unresolved. It is still unclear how CSN selectively recognises neddylated CRLs while excluding ubiquitinated or unmodified complexes. Furthermore, the precise molecular interfaces driving CSN5 activation, NEDD8 cleavage, and subsequent CRL release remain incompletely characterised. A detailed structural understanding of these processes is not only fundamental to CRL biology but also has translational significance. CSN is a validated therapeutic target with roles in cell cycle regulation and tumorigenesis, yet no clinical inhibitors have been developed to date[29–31].

In this study, we reconstitute recombinant CSN-SCF (CRL1) complexes representing distinct catalytic states and determine high-resolution structures using cryo-electron microscopy (cryo-EM) single-particle analysis. These structures capture key mechanistic stages of the CSN-mediated deneddylation cycle, including CSN association, catalytic activation and product dissociation. We identify a pre-activated conformation in which CSN docks to SCF but remains autoinhibited, an activated state involving a conformational rearrangement of the CSN5 Ins-1 loop that positions the isopeptide bond for catalysis, and multiple dissociation intermediates where the RBX1 RING domain stabilises critical interfaces during product release. Supported by biochemical assays, these findings offer a detailed mechanistic model of CSN-mediated CRL deactivation.

## Results

### Cryo-EM and single particle analysis of CSN-$^{N8}$SCF complexes

To investigate how CSN recognises and deneddylates its substrate, we reconstituted a CSN$^{5H138A}$ complex comprising CSN1-8 and CSNAP, where the CSN5$^{H138A}$ mutation abolishes deneddylation activity[16], enabling stable association with a separately prepared neddylated SCF complex consisting of $^{N8}$CUL1/RBX1/SKP1/SKP2/CKS1 ($^{N8}$SCF). The CSN$^{5H138A}$-$^{N8}$SCF complex was purified using analytical size-exclusion chromatography (Supplementary Fig. 1a) and subsequently subjected to cryo-EM and single-particle analysis (Supplementary Fig. 1b–j).

After particle selection through 2D and 3D classification, we obtained an initial map at 3.2 Å. However, the CSN5-CSN6 and NEDD8-WHB ($^{N8}$CUL1$^{WHB}$) regions were poorly resolved, likely due to conformational heterogeneity. To improve subclassification, we applied cryoDRGN[32], identifying two major classes with distinct NEDD8 positions. We generated a single mask incorporating both NEDD8 positions for 3D local classification in RELION-4.0, which resulted in two major conformations with well-resolved but distinct CSN5-CSN6 and $^{N8}$CUL1$^{WHB}$ regions. Each conformation was separately refined to ~3.2 Å resolution. To improve the resolution of the SR region (SKP1-SKP2-CKS1), we performed additional focused classification and refinement (Supplementary Fig. 2), yielding two maps at 3.8 Å and 3.5 Å resolution. However, this process significantly reduced particles numbers, resulting in lower resolution at the CSN5–NEDD8–CUL1$^{WHB}$ interface, which is critical for understanding the deneddylation mechanism. Therefore, we based our model building (Figs. 1a, b and 2a, b) on the maps shown in Supplementary Fig. 1g, j. Fourier Shell Correlation (FSC) analysis confirmed good model-to-map agreement with minimal overfitting (Supplementary Fig. 3).

The two structures contain the expected 15 subunits and are highly similar overall, with a root-mean-square deviation (RMSD) of

0.92 Å. Additionally, they closely resemble previously reported cryo-EM structures of related complexes[15,23–25]. However, these earlier studies lacked the resolution required for atomic-level interpretation. For comparative analysis, we therefore reference previously determined near-atomic resolution structures, including CSN complex in CSN-CRL2 complex (PDB: 6R7I)[24], the SKP1/SKP2/CKS1 SR complex (PDB: 2ASS)[33] and $^{N8}$SCF from a trapped ubiquitination intermediate complex (PDB: 1LDK)[34].

CSN is assembled into the previously described horseshoe-shaped ring[17], formed by association of the winged-helix (WH) subdomains from the six proteasome lid–CSN–initiation factor 3 (PCI ring) containing subunits (CSN1-4 and CSN7-8), along with the globular CSN5-CSN6 heterodimer. These eight core subunits are interconnected through a C-terminal α-helical bundle. In our structures, we observe that the ninth subunit, CSNAP, is incorporated into a groove formed by CSN3 and CSN8. The structural and functional implications of this interaction will be described in detail later.

Overall $^{N8}$SCF also closely resembles previous structures[35] with CUL1, serving as a scaffold protein, interacting with the N-terminal region of RBX1 via its C-terminal domain (CUL1$^{CTD}$) and with SKP1/SKP2/CKS1 via its N-terminal domain (CUL$^{NTD}$). Notably, no density was observed for α-helix 27, which connects CUL1$^{WHB}$ to CUL1$^{CTD}$, consistent with previous reports indicating structural flexibility in this region. Importantly, the CUL1$^{WHB}$ domain and covalently linked NEDD8 are clearly resolved, providing near-atomic resolution insights into the dramatic conformational rearrangements in this region. The most striking structural differences between our two structures are observed around the $^{N8}$CUL1$^{WHB}$ module and the CSN5 active site. Based on these differences, described in detail below, we refer to the two structures as pre-activated and activated conformations, which together reveal the activation and deneddylation mechanisms of CSN.

### CSNAP is an integral component of CSN

In both pre-activated and activated CSN$^{5H138A}$-$^{N8}$SCF structures, we can resolve how CSNAP integrates into the CSN complex, with well-defined density for its C-terminal region, enabling residues 37–57 to be modelled into our Cryo-EM maps (Fig. 1c). Adopting an extended conformation, CSNAP spans across CSN1, CSN3 and CSN8, consistent with structural predictions from an AlphaFold2-generated model[19]. Interestingly, no density is observed for CSNAP's N-terminal region across any of our structures, consistent with a previous study, which found that deleting the first 20 residues did not disrupt its integration into the CSN complex[18].

CSNAP engages with the loop regions following the helical bundle from the CSN1 C-terminus, where CSNAP$^{V39}$ forms hydrophobic interactions with CSN1$^{V466}$ and CSN1$^{P469}$ (Supplementary Fig. 4a). Beyond this, most of its interactions are mediated by a continuous stretch of hydrophobic and acidic residues in CSNAP's characteristic F/D-rich motif. Specifically, CSNAP's hydrophobic stretch (F43, F44, F47, L50 and F51) nestles within a channel along the inner surface of the CSN3 helical repeat extension, forming extensive hydrophobic contacts with CSN3 residues (L151, F155, Y186, Y187, M190, I219) (Supplementary Fig. 4b). Meanwhile, CSNAP's acidic tail (E48, D52, D53, D54 and D55) occupies an electropositive groove formed between CSN3 (K153, K196, K296) and CSN8 (R54, K58), where multiple salt bridges and hydrogen bonds further stabilise the interface (Supplementary Fig. 4c).

These findings align with previous mutational studies[18] and peptide array results[19], which demonstrated that disrupting key residues within the F/D-rich region severely impairs CSNAP's binding to CSN. Finally, CSNAP's placement within the complex is secured by its C-terminal residues, which form direct contacts with the N-terminal helical repeats of CSN8. Notably, CSN8$^{W75}$ and CSN8$^{W82}$ secure CSNAP$^{I56}$, while the indole nitrogen of CSN8$^{W75}$ makes a hydrogen bond with the free carboxyl group of CSNAP$^{Q57}$, firmly anchoring CSNAP's C-terminus (Supplementary Fig. 4d).

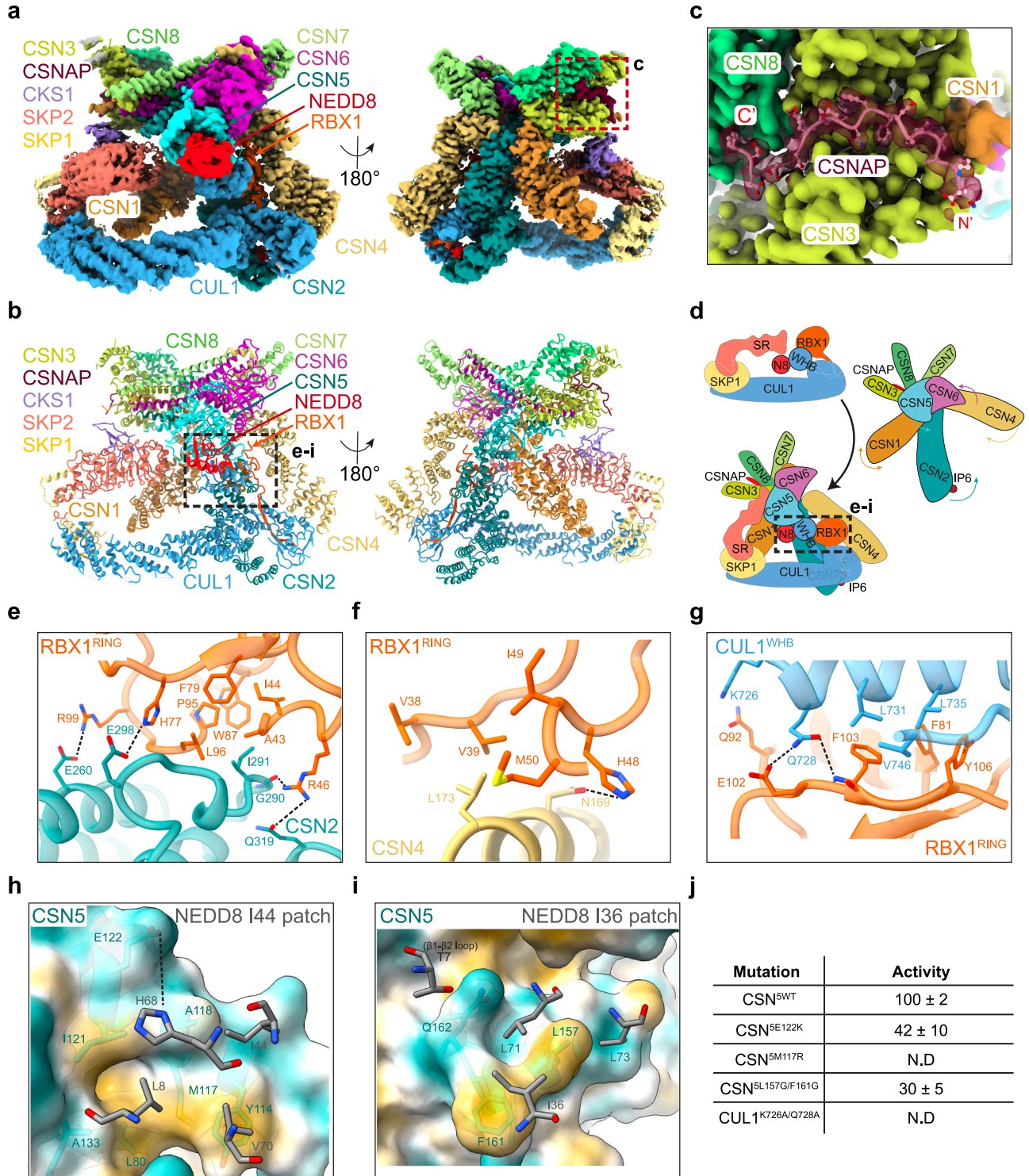

**Fig. 1 | Cryo-EM analysis of a pre-activated CSN$^{5HI38A}$-$^{N8}$SCF complex. a** Cryo-EM density map of pre-activated CSN$^{5HI38A}$-$^{N8}$SCF. **b** Molecular model of pre-activated CSN$^{5HI38A}$-$^{N8}$SCF (ribbon representation). **c** Close-up of CSNAP at its interface with CSN1, CSN3 and CSN8. **d** Schematic depicting the conformational changes from CSN$^{apo}$ and SCF to pre-activated CSN$^{5HI38A}$-$^{N8}$SCF. **e** Interaction interface between CSN2 and RBX1$^{RING}$. **f** Interaction interface between CSN4 and RBX1$^{RING}$. **g** Interaction interface between CUL1$^{WHB}$ and RBX1$^{RING}$. **h** CSN5 engagement with the NEDD8 Ile44

patch, shown on a hydrophobic surface (yellow: hydrophobic; cyan: hydrophilic). CSN5$^{E122}$ side chain is modelled as an approximation due to limited density. **i** Interaction between CSN5 and the NEDD8 I36 patch and β1-β2 loop. **j** In vitro deneddylation assays for key interface mutants in pre-activated CSN$^{5HI38A}$-$^{N8}$SCF. N.D. indicates "not determined". Data represent the mean ± SD from three independent experiments.

To investigate the effect of CSNAP on binding, we performed surface plasmon resonance (SPR) kinetic measurements comparing CSN$^{5HI38A}$ complexes with and without CSNAP (referred to as 9CSN$^{5HI38A}$ and 8CSN$^{5HI38A}$, respectively) with both neddylated and non-

neddylated CUL1/RBX1 (Tables 1 and 2 and Supplementary Fig. 5a–h). Consistent with prior findings[21], incorporation of CSNAP into the CSN complex reduces the binding affinity for both neddylated and non-neddylated CUL1/RBX1. For the neddylated substrate,

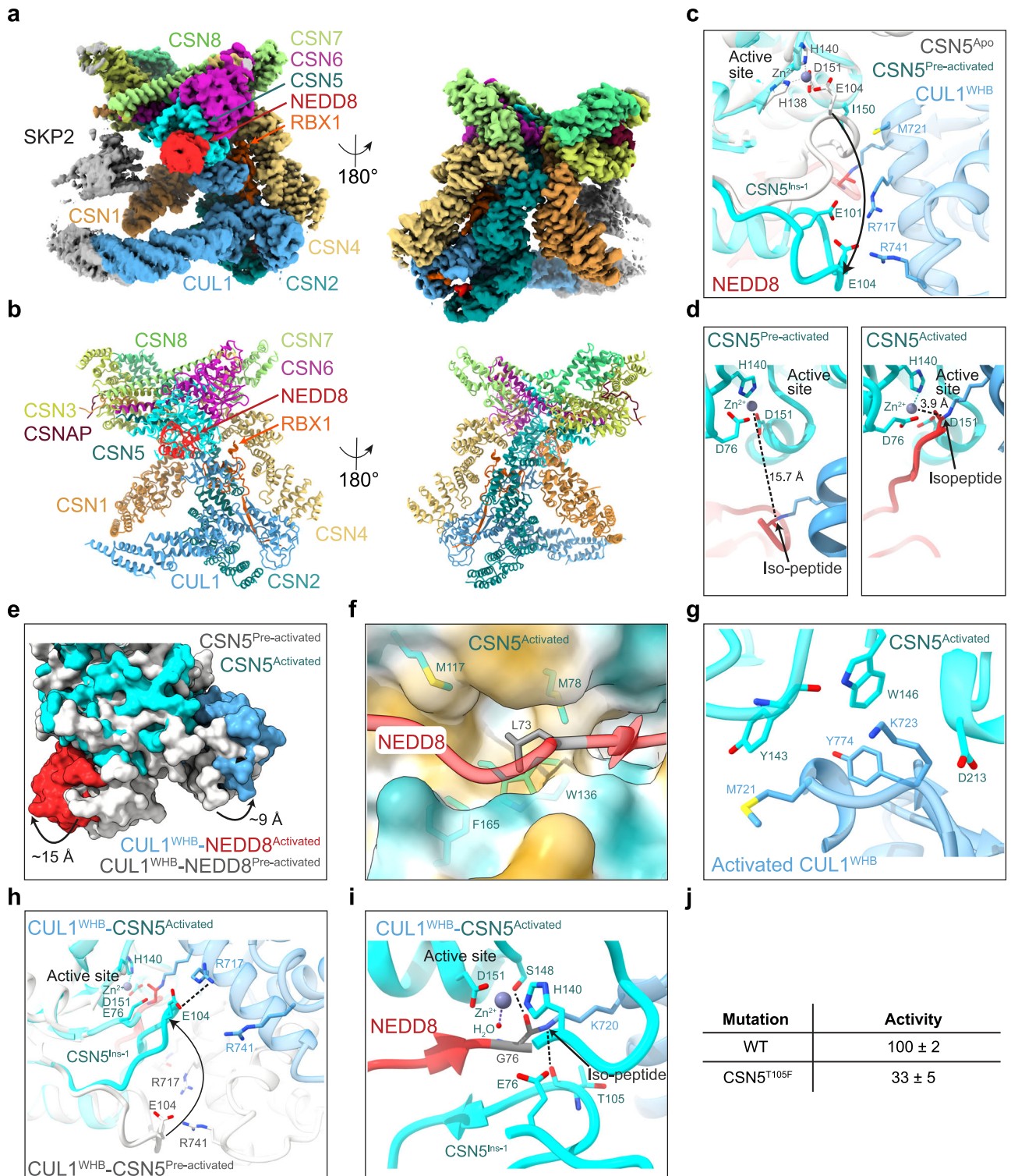

**Fig. 2 | Cryo-EM analysis of an activated CSN$^{SHI38A}$-$^{N8}$SCF complex and structural insights into CSN5 activation. a** Cryo-EM density of activated CSN$^{SHI38A}$-$^{N8}$SCF. **b** Molecular model of activated CSN$^{SHI38A}$-$^{N8}$SCF. **c** CSN5$^{Ins-1}$ remodelling between CSN5$^{apo}$ (grey) and pre-activated CSN5 (cyan), and its interface with CUL1$^{WHB}$. Zn$^{2+}$ is modelled for illustration, while CSN5$^{E101, E104}$ side chains are approximations. **d** Comparison of isopeptide bond positioning in pre-activated CSN5 (left panel) versus activated CSN5 (right panel). **e** Conformational transitions in CSN5, CUL1$^{WHB}$ and NEDD8 from pre-activated (grey) to activated CSN$^{SHI38A}$-$^{N8}$SCF (coloured).

**f** Stabilisation of CSN5-NEDD8 interface via NEDD8$^{L73}$ in the I36 patch. CSN5 is displayed as a hydrophobic surface (yellow: hydrophobic; cyan: hydrophilic). **g** Interface between activated CSN5 and CUL1$^{WHB}$. **h** CSN5$^{Ins-1}$ rearrangement from pre-activated CSN5 (grey) to activated CSN5 (cyan). **i** Proposed deneddylation mechanism of the CUL1$^{K720}$-NEDD8$^{G76}$ isopeptide by activated CSN5. The catalytic Zn$^{2+}$ and H$_2$O are docked from PDB: 4F7O. **j** Deneddylation assay summary. Data represent the mean ± SD from three independent experiments.

## Table 1 | Binding affinity and kinetics measurement results summary with neddylated CUL1/RBX1

| | CSN | $^{N8}$CUL1/RBX1 | $K_d$ (nM) steady-state | $K_d$ (nM) kinetics | $k_{on}$ ($10^6$ M$^{-1}$s$^{-1}$) | $k_{off}$ (s$^{-1}$) |
|---|---|---|---|---|---|---|
| | 8CSN$^{5H138A}$ | WT | 6 ± 3 | 1.6 ± 0.3 | 1.0 ± 0.4 | 0.0016 ± 0.0005 |
| Interface | 9CSN | | | | | |
| | WT | WT | / | | | |
| | 5H138A | WT | 10 ± 3 | 2.6 ± 0.04 | 0.37 ± 0.2 | 0.00097 ± 0.0004 |
| RBX1-WHB | 5H138A | K726A/Q728A | / | | | |
| CSN5-WHB | 5H138A/E122K | WT | 18 ± 3 | 8.1 ± 1 | 0.68 ± 0.02 | 0.0055 ± 0.0003 |
| CSN5-NEDD8 | 5H138A/M117R | WT | 120 ± 9 | 80 ± 30 | 0.095 ± 0.02 | 0.0074 ± 0.003 |
| CSN5-NEDD8 | 5H138A/L157G/F161G | WT | 80 ± 10 | 20 ± 2 | 0.65 ± 0.2 | 0.015 ± 0.006 |
| CSN5-WHB | 5H138A/T105F | WT | 8 ± 0.6 | 4.2 ± 3 | 0.33 ± 0.008 | 0.00055 ± 0.00002 |
| CSN5-WHB | 5H138A | R741E | 12 ± 4 | 1.4 ± 0.4 | 0.39 ± 0.2 | 0.00051 ± 0.0001 |
| CSN5-WHB | 5H138A | R717E | 12 ± 4 | 1.9 ± 0.9 | 0.34 ± 0.1 | 0.00059 ± 0.00008 |
| CSN4-RBX1 | 5E104A | WT | / | | | |

## Table 2 | Binding affinity measurement results summary with non-neddylated CUL1/RBX1

| | CSN | CUL1/RBX1 | $K_d$ (nM) steady-state |
|---|---|---|---|
| | 8CSN$^{5H138A}$ | WT | 90 ± 8 |
| Interface | 9CSN | | |
| | WT | WT | 1400 ± 200 |
| | 5H138A | WT | 260 ± 30 |
| RBX1-WHB | 5H138A | K726A/Q728A | 450 ± 40 |
| CSN5-WHB | 5H138A/E122K | WT | 380 ± 20 |
| CSN5-NEDD8 | 5H138A/M117R | WT | 760 ± 20 |
| CSN5-NEDD8 | 5H138A/L157G/F161G | WT | 530 ± 40 |
| CSN5-WHB | 5H138A/T105F | WT | 370 ± 50 |
| CSN5-WHB | 5H138A | R741E | 140 ± 30 |
| CSN5-WHB | 5H138A | R717E | 140 ± 20 |
| CSN4-RBX1 | 5E104A | WT | 360 ± 30 |

9CSN$^{5H138A}$ exhibits a slower association rate ($k_{on}$ = 0.37 × $10^6$ M$^{-1}$s$^{-1}$) compared to 8CSN$^{5H138A}$ ($k_{on}$ = 1.0 × $10^6$ M$^{-1}$s$^{-1}$), while the dissociation rate remains largely unchanged. Conversely, for the non-neddylated substrate, the association rates are similar between the two complexes, but the dissociation rate is higher for 9CSN$^{5H138A}$ ($k_{off}$ = 0.16 s$^{-1}$) compared to 8CSN$^{5H138A}$ ($k_{off}$ = 0.04 s$^{-1}$). These kinetic findings suggest that CSNAP reduces overall affinity for CRL substrates by modulating complex stability, specifically, by decreasing the association rate for neddylated CRLs and increasing the dissociation rate for non-neddylated CRLs. CSNAP shares key sequence properties with its paralogue DSS1[18], a critical subunit of the 26S proteasome lid complex that enhances proteasome stability[36]. Significantly, DSS1 interacts with the PCI-domain subunits RPN3 and RPN7[37,38], which are evolutionarily related to CSN's CSN3 and CSN1 subunits, respectively[39]. However, despite these similarities, CSNAP and DSS1 exhibit distinct binding modes within their respective complexes. CSNAP adopts an extended conformation bridging CSN3 and CSN8 (Supplementary Fig. 4e), whereas DSS1 runs in the opposite direction, extending from RPN3 toward the PCI ring of RPN7, where its C-terminal residues adopt a α-helical structure (PDB: 3JCK)[40] (Supplementary Fig. 4f).

Our structural findings confirm CSNAP as an integral component of the CSN complex. While its N-terminal region remains flexible across all structures, its C-terminal F/D-rich motif consistently adopts a stable binding conformation anchored by a tripartite scaffold composed of CSN1, CSN3 and CSN8. Our complementary kinetics analysis indicates that CSNAP modulates CSN-substrate interaction dynamics. Specifically, incorporation of CSNAP into the CSN complex reduces the association rate with neddylated CUL1/RBX1 and increases the

dissociation rate from non-neddylated CUL1/RBX1. These findings support a model in which CSN acts as a regulatory subunit that fine-tunes substrate binding and release without directly contributing to deneddylation activity. Although CSNAP shares similarities with DSS1, its features highlight a potential for functional specialisation[41]. Notably, CSNAP has been shown to attenuate CSN binding to Cullins, even in the absence of NEDD8 and SR[21], suggesting that it may modulate this interaction by restricting CSN flexibility, thereby facilitating the efficient disassembly and remodelling of CRL complexes. Thus, we cannot exclude that our ability to obtain seven cryo-EM maps at near-atomic resolution is partly attributed to CSNAP's effect on the overall rigidity and stability of the CSN complex.

### The structure of pre-activated CSN-$^{N8}$SCF uncovers a multivalent recognition mechanism

In pre-activated CSN$^{5H138A}$-$^{N8}$SCF, we observe striking conformational changes in both CSN (Supplementary Fig. 6a–d) and $^{N8}$SCF components (Supplementary Fig. 7a–e), compared with CSN$^{apo}$ (PDB: 4D10) and $^{N8}$SCF (a superimposed complex with $^{N8}$CUL1/RBX1/SKP1 from PDB: 6TTU[35], SKP2/CKS1 were from PDB: 2ASS[33]), driven by a series of multivalent binding and pre-activation events (Supplementary Movie 1). The key structural rearrangements include: (i) the presence of a neddylated SCF triggers the convergence of the N-terminal helical arms of CSN2 and CSN4 (hereafter CSN2$^{arm}$ and CSN4$^{arm}$ respectively), clamping CUL1$^{CTD}$ and RBX1$^{RING}$, (ii) the flexible CUL1$^{WHB}$ is repositioned against the stabilised RBX1$^{RING}$, aligning NEDD8 at the CSN active site and (iii) concurrent disruption of the CSN4-CSN6 interface enables flexibility and remodelling of the CSN5-CSN6 MPN domains, bringing the CSN5 active site closer to NEDD8 (Fig. 1d). As well as coordinating multiple remodelling steps, CSN must also accommodate substantial SR diversity and stabilise the N-terminal end of the Cullin-SR region.

Compared to CSN$^{apo}$ (PDB: 4D10)[17], the presence of the $^{N8}$SCF drives substantial structural shifts in CSN2$^{arm}$ and CSN4$^{arm}$ (Supplementary Fig. 6a), accompanied by an expansion of the PCI ring (Supplementary Fig. 6b). CSN4$^{arm}$ undergoes a dramatic rotation towards CSN2, pivoting around a flexible "hinge" loop (CSN4$^{hinge-loop}$: residues 291–299) just before its WH domain. In contrast, CSN2$^{arm}$ experiences a more subtle rotation near its WH domain, accompanied by an internal twist of its helical repeats. This conformational clamping by CSN2 and CSN4 establishes multiple contact points with the CUL1$^{CTD}$, engaging the WHA, the α/β and the four-helix bundle subdomains (Supplementary Fig. 8a–e). Notably, CSN2 contributes to the majority of these interactions, aligning with previous binding studies that identified its helical arm as the primary recognition module for neddylated SCF[14,15,23–25].

To accommodate CSN, RBX1$^{RING}$ rotates from the position observed in the active UBE2D2-$^{N8}$SCF complex (PDB: 6TTU)[35]

(Supplementary Fig. 7b), adopting an alternative conformation within the CSN2-CSN4 clamp. At this interface, an intricate network of interactions functions to stabilise the RING domain (Fig. 1e, f and Supplementary Fig. 8f). Multiple studies involving CRL1, CRL2 and CRL4A complexes have previously demonstrated that CSN2 and CSN4 and the RBX1$^{RING}$ domain are essential for substrate binding and efficient deneddylation[15,23,24]. However, in this study, we provide high-resolution structural insights into these interactions, made possible by improved cryo-EM map quality. Structurally, the RBX1$^{RING}$ consists of two zinc-chelating loops (RBX1$^{RING-loop1}$ and RBX1$^{RING-loop2}$), an α-helix (RBX1$^{RING-helix3}$) and a three-stranded β-sheet, alongside an insertion motif (RBX1$^{RING-insertion}$: residues 50–70) stabilised by a third zinc ion (Supplementary Fig. 8f). At the CSN2-RBX1$^{RING}$ interface, a shallow groove in the RBX1$^{RING}$ accommodates CSN2$^{arm}$. This groove, predominantly formed by the RBX1$^{RING-helix3}$, RBX1$^{RING-loop1}$ and RBX1$^{RING-loop2}$, features residues RBX1$^{A43}$, RBX1$^{I44}$, RBX1$^{F79}$, RBX1$^{W87}$, RBX1$^{P95}$ and RBX1$^{L96}$, which create a hydrophobic environment for CSN2$^{I291}$ (Fig. 1e). Additional stability comes from two salt bridges, CSN2$^{E260}$-RBX1$^{R99}$ and CSN2$^{D298}$-RBX1$^{H77}$, along with a hydrogen bond between the backbone carbonyl group of CSN2$^{G290}$ and side chain RBX1$^{R46}$ (Fig. 1e). On the opposite side of the clamp, CSN4$^{arm}$ links to RBX1 via RBX1$^{RING-loop1}$ and the RBX1$^{RING-insertion}$ motif, forming further stabilising interactions. The binding surface is predominantly hydrophobic, with residues RBX1$^{V38}$, RBX1$^{V39}$, RBX1$^{I49}$ and RBX1$^{M50}$ creating a pocket for CSN4$^{L173}$ (Fig. 1f). Additionally, the imidazole ring of RBX1$^{H48}$ engages in a hydrogen bond with the side chain of CSN4$^{N169}$, further strengthening the interface (Fig. 1f). We note that these interactions are conserved in the activated CSN$^{5H138A}$-$^{N8}$SCF complex (Supplementary Fig. 9a–c).

CSN directly competes with RBX1$^{RING}$ interactors[14], and a comparison of pre-activated CSN$^{5H138A}$-$^{N8}$SCF with trapped active $^{N8}$SCF complexes[11,35] highlights how E2 interactions are effectively blocked when a CRL is bound by CSN. First, the positioning of the CSN2-CSN4 clamp around the RING in CSN-$^{N8}$SCF creates a significant steric clash with the E2 (Supplementary Fig. 10a, b). Second, CSN2 shares the same interaction surface on RBX1 as the E2, directly obstructing the E2's access to the RING. This effectively "switches off" ubiquitination, safeguarding CRLs from unwanted autoubiquitination in the absence of a substrate. Furthermore, until CSN deactivates the CRL, the SR exchange factor CAND1 is prevented from binding. Structural comparisons with a CAND1-bound SCF complex (PDB: 8OR0)[42] reveal a potential steric clash between CSN1, CSN2 and the RBX1$^{RING}$ with CAND1 (Supplementary Fig. 10c). This suggests that the cycling of CRLs between CSN and CAND1 is tightly regulated, ensuring that CRLs are either protected from auto-destruction or primed for SR exchange as needed.

One of the most striking conformational differences in $^{N8}$SCF, compared to its ubiquitin-transfer active state (PDB: 6TTU)[35], occurs in the region of CUL1$^{WHB}$ and its isopeptide-linked NEDD8. When CSN is present, a significant rearrangement of the flexibly tethered $^{N8}$CUL1$^{WHB}$ module is required to deliver NEDD8 at the CSN active site (Supplementary Fig. 7c and Supplementary Movie 1). While the linker connecting the CUL1$^{WHB}$ to the CUL1 α/β domain could not be resolved, our cryo-EM map reveals densities for both CUL1$^{WHB}$ and NEDD8. The $^{N8}$CUL1$^{WHB}$ module extends toward CSN5, with CUL1$^{WHB}$ firmly positioned against the CSN-stabilised RING. This interaction is supported by a hydrophobic network, where residues RBX1$^{F81}$, RBX1$^{F103}$ and RBX1$^{Y106}$, located within a channel formed by the RBX1$^{RING-helix3}$ and third C-terminal β-strand, establish extensive contacts with CUL1$^{L731}$, CUL1$^{L735}$ and CUL1$^{V746}$ of CUL1$^{WHB}$ (Fig. 1g). Additionally, the side chain of CUL1$^{Q728}$ forms a hydrogen bond with the backbone amide group of RBX1$^{F103}$ and may also interact with the carboxyl group of RBX1$^{E102}$, although the glutamate side-chain density is unresolved. Likewise, while the side-chain densities for RBX1$^{Q92}$ and CUL1$^{K726}$ are ambiguous, a potential interaction at this site cannot be ruled out. This CUL1$^{WHB}$-

RBX1$^{RING}$ interaction network is also retained in the activated CSN$^{5H138A}$-$^{N8}$SCF complex (Supplementary Fig. 9d).

The pre-activated CSN$^{5H138A}$-$^{N8}$SCF complex reveals an intricate network of interactions among the CSN2-CSN4 clamp, RBX1$^{RING}$ and CUL1$^{WHB}$, ensuring their stable conformation within the complex. Since CUL1$^{WHB}$ positioning is critical for aligning NEDD8 with CSN5, we hypothesised that disrupting the interface between RBX1$^{RING}$ and CUL1$^{WHB}$ would impair CSN activity. To test this, we introduced a CUL1-K726A/Q728A double mutation in CUL1$^{WHB}$ (Supplementary Fig. 11a) and employed two complementary assays to validate the functional significance of the interface: SPR to measure the binding affinities and in vitro deneddylation assays[43,44] to assess enzymatic activity. Binding assays confirmed that the non-neddylated CUL1$^{K726A/Q728A}$/RBX1 construct interacts with CSN$^{5H138A}$ similarly to wild-type CUL1/RBX1 (Table 2 and Supplementary Figs. 5h and 12b). However, the mutant exhibited reduced catalytic activity in the deneddylation assay compared to wild-type $^{N8}$CUL1/RBX1 (Fig. 1j and Supplementary Fig. 13a, b, g), suggesting that disrupting the RBX1$^{RING}$-CUL1$^{WHB}$ interface destabilises NEDD8's positioning, ultimately impairing its cleavage by CSN5.

The clamping motion of CSN2 and CSN4 sets off a cascade of structural rearrangements throughout the CSN complex (Supplementary Movie 1). In pre-activated CSN$^{5H138A}$-$^{N8}$SCF, the expansion of the CSN PCI ring and rearrangements within the CSN helical bundle (Supplementary Fig. 6b, c) facilitate the release of the CSN6 Ins-2 loop (CSN6$^{Ins-2}$) from CSN4[17]. This increased flexibility, in turn, triggers a substantial rotation of the CSN5-CSN6 MPN heterodimer (Supplementary Fig. 6d), repositioning CSN5 and NEDD8 closer to each other. CSN5 interacts with three key conserved regions on NEDD8. First, CSN5$^{Q162}$ stabilises the flexible β1-β2 loop of NEDD8 through van der Waals contacts and a hydrogen bond with the hydroxyl group of NEDD8$^{T7}$. Second and third, NEDD8's surface-exposed hydrophobic regions, the I44 and I36 patches, form additional stabilising interactions. The I44 patch (L8-I44-V70) makes multiple van der Waals contacts with CSN5$^{YI14}$, CSN5$^{MI17}$, CSN5$^{AI18}$ and CSN5$^{I121}$ (Fig. 1h), while the I36 patch (I36-L71-L73) engages CSN5$^{L157}$, CSN5$^{F161}$ and CSN5$^{Q162}$ (Fig. 1i). At the core of NEDD8's I44 patch lies a conserved histidine residue, NEDD8$^{H68}$. The side chain of NEDD8$^{H68}$ is oriented toward CSN5$^{E122}$, and although the density for this glutamate side chain is poor, we investigated whether these residues contribute to the interaction interface. To test this idea, we introduced an E122K mutation in CSN5 (Supplementary Fig. 11b, c) and applied this charge swap mutant in binding and activity assays. While CSN$^{5H138A/E122K}$ bound neddylated CUL1/RBX1 with comparable affinity to wild-type CSN (Table 1 and Supplementary Fig. 12c), it exhibited an ~50% reduction in deneddylation activity (Fig. 1j and Supplementary Fig. 13c, g). To elucidate a functional role for CSN5$^{E122}$, we performed kinetic analysis of the SPR data (Table 1 and Supplementary Figs. 14a and 5c). While the association rate constant ($k_{on}$) to neddylated CUL1/RBX1 was similar to that of CSN$^{5H138A}$, the dissociation rate constant ($k_{off}$) was accelerated nearly sixfold, indicating that CSN5$^{E122}$ is not directly involved in substrate recognition, but instead contributes to stabilising the enzyme-substrate complex, possibly through an interaction with NEDD8$^{H68}$. The weakened complex stability observed in this mutant likely accounts for its reduced deneddylation activity.

To explore the hydrophobic contributions between CSN5-NEDD8 interactions, we introduced targeted mutations in CSN5. An M117R mutation (Supplementary Fig. 11d, e) was designed to disrupt contacts with the I44 patch, while a double L157G/F161G mutation (Supplementary Fig. 11f, g) targeted the I36 patch. Kinetic SPR analysis revealed that the CSN$^{5H138A/M117R}$ mutant exhibited an approximately 30-fold reduction in binding affinity for neddylated CUL1/RBX1, resulting from an ~fourfold decrease in the $k_{on}$ rate and an ~eightfold increase in the $k_{off}$ rate, relative to CSN$^{5H138A}$. Interestingly, the same mutation also led to a modest ~threefold reduction in binding affinity for non-neddylated CUL1/RBX1, resulting from an ~threefold slower

association rate (Tables 1 and 2 and Supplementary Figs. 12e, f, 14b and 5g–i). The CSN$^{5H138A/L157G/F161G}$ mutant showed an eightfold decrease in binding to neddylated CUL1/RBX1, primarily due to a faster $k_{off}$. This mutant had little impact on binding to non-neddylated CUL1/RBX1 (Tables 1 and 2 and Supplementary Fig. 12g, h). Both mutations impaired deneddylation activity, with CSN$^{MI17R}$ nearly abolishing catalysis (Fig. 1j and Supplementary Fig. 13d, e, g). These findings demonstrate that CSN5's interaction with the I36 and I44 patches of NEDD8 plays a fundamental role in CSN-mediated deneddylation. First, residues CSN5$^{L157}$ and CSN5$^{F161}$, which contact the I36 patch, do not seem to affect initial binding but instead stabilise the enzyme-substrate complex. Along with CSN5$^{E122}$, they form a stabilising interface that promotes substrate retention and efficient catalysis. Second, although disrupting the CSN5-NEDD8 I44 interface through the CSN$^{5MI17R}$ mutation impaired binding and catalysis, the modest impact on binding to the non-neddylated substrate suggests that CSN5$^{MI17}$ may also influence complex formation independently of NEDD8. Conserved residues across Cullin and RBX family members align with many of the interfaces observed in the pre-activated CSN$^{5H138A}$-$^{N8}$SCF complex, suggesting that the CSN recognition mechanism described in our study is likely shared throughout the entire Cullin-RING E3 ligase family (Supplementary Fig. 15)[9,45].

SKP2, alongside its co-receptor CKS1, is nestled within a space formed by CSN1, CSN3, the distal end of the helical bundle, and CSN5 (Supplementary Fig. 16a). The N-terminal helical arm of CSN1 (CSN1$^{arm}$) has been implicated in interactions with various SRs, including SKP2 and FBW7 in $^{N8}$CRL1[14], DDB1 in $^{N8}$CRL4[15] and KBTBD2 in $^{N8}$CRL3[25] complexes. However, the nature and extent of these interactions appear variable. In the pre-activated CSN$^{5H138A}$-$^{N8}$SCF complex, CSN1$^{arm}$ approaches the final helix of the SKP2 LRR domain but remains just beyond reach for direct interaction. Cryo-EM maps reveal that the N-terminal regions of CUL1, SKP1 and SKP2 exhibit poorly resolved density and lower local resolution (Supplementary Fig. 1g), reflecting significant conformational flexibility of this region. Similarly, while CSN5 is positioned close to SKP2, the lack of defined density prevents the identification of specific contacts. Interestingly, CKS1 appears to be stabilised by CSN3. Well-defined density suggests that CSN3$^{Q247}$ and CSN3$^{R251}$ may form van der Waals interactions with CKS1$^{Y7}$ and CKS1$^{Y8}$, though the absence of side chain density for CKS1 residues leaves this interaction as speculative (Supplementary Fig. 16b). The flexibility observed near the SR region of the complex likely reflects the structural adaptability CSN requires to engage a diverse range of SRs, accommodating their variations in size and geometry while maintaining partial stabilisation (Supplementary Fig. 17a, b). Given that CSN can only act after the ubiquitinated substrate has dissociated from the CRL, ensuring an efficient catalytic cycle[14,43,46], we docked p27 (PDB: 2ASS)[33], the substrate of SKP2-CKS1, onto our structure (Supplementary Fig. 16c). Strikingly, the model suggests that CSN cannot engage with substrate-bound SCF, as the presence substrate induces steric clashes with CSN, highlighting a potential regulatory mechanism in substrate detection.

## Activation of the CSN5 deneddylation machinery

To understand how $^{N8}$SCF binding orchestrates CSN5 activation, we compared the active site architecture across the three key states: CSN$^{apo}$, and the determined pre-activated and activated CSN$^{5H138A}$-$^{N8}$SCF complexes. In the CSN$^{apo}$ state (PDB: 4D10)[17], CSN5 remains autoinhibited, with the catalytic site shielded by CSN5$^{E104}$ within the Ins-1 loop (CSN5$^{Ins-1}$: residues 98–109). This loop serves as a fourth ligand to the zinc-binding site, keeping the isopeptidase inactive. However, in the pre-activated CSN$^{5H138A}$-$^{N8}$SCF complex, autoinhibition is relieved as the flexible CSN5$^{Ins-1}$ undergoes a displacement from the active site, allowing access for the incoming isopeptide bond (Fig. 2c and Supplementary Movie 2). Comparison with CSN$^{apo}$ reveals that this shift is driven by the $^{N8}$CUL1$^{WHB}$ module. Specifically, CUL1$^{WHB}$

residue CUL1$^{M721}$ and the CUL1$^{720}$-NEDD8$^{G76}$ isopeptide now pack against CSN5$^{I150}$, a residue that, in the autoinhibited state, had been engaged in multiple van der Waals interactions with CSN5$^{Ins-1}$. Due to the inherent flexibility of the CSN5$^{Ins-1}$ region, we were unable to accurately model all side chains within the loop (Supplementary Fig. 18a, b). However, it is notable that CSN5$^{E101}$ and CSN5$^{E104}$ are positioned near CUL1$^{WHB}$ residues CUL1$^{R717}$ and CUL1$^{R741}$, respectively, suggesting that charge complementarity between these residues may contribute to stabilising the transient pre-activated conformation of the loop. To complete the tetrahedral coordination of the Zn$^{2+}$-binding site, we propose that the JAMM motif residue CSN5$^{E76}$ replaces CSN5$^{E104}$, consistent with the crystallographic structure of isolated CSN5[22]. However, as we worked with CSN5$^{H138A}$ to facilitate complex assembly, we did not observe density corresponding to the catalytic Zn$^{2+}$ and, therefore, cannot directly confirm this interaction.

Surprisingly, despite the CSN5 active site appearing catalytically competent in pre-activated CSN$^{5H138A}$-$^{N8}$SCF, the CUL1$^{K720}$-NEDD8$^{G76}$ isopeptide bond remains positioned too far from the active site to enable deneddylation (Fig. 2d, left panel). This indicates that further structural rearrangements are necessary for full activation. In the activated CSN$^{5H138A}$-$^{N8}$SCF complex, targeted remodelling around CSN5, CUL1$^{WHB}$ and NEDD8 creates interaction surfaces that are specific to the activated state (Fig. 2e and Supplementary Movie 2). Specifically, the C-terminal tail of NEDD8 is repositioned closer to the CSN5 active site. This shift is stabilised by NEDD8$^{L73}$, which moves away from its original I36 patch position, extending the NEDD8 C-terminal tail. In this conformation, NEDD8$^{L73}$ is accommodated within a hydrophobic pocket formed by CSN5$^{M78}$, CSN5$^{MI17}$, CSN5$^{W136}$ and CSN5$^{F165}$ (Fig. 2f). Simultaneously, a concerted repositioning of CUL1$^{WHB}$ and CSN5 establishes an additional interface between the two proteins. Whereas pre-activated CSN$^{5H138A}$-$^{N8}$SCF featured only a modest contact between CUL1$^{M721}$ and CSN5$^{I150}$, in activated CSN$^{5H138A}$-$^{N8}$SCF, the interaction surface expands significantly, with CUL1$^{M721}$, now aided by CUL1$^{K723}$ and CUL1$^{Y774}$, making multiple van der Waals contacts with CSN5$^{Y143}$, CSN5$^{W146}$ and CSN5$^{D213}$ (Fig. 2g). This increased interface between the $^{N8}$CUL1$^{WHB}$ module and CSN5 is critical for stabilising the isopeptide bond in the activated state.

Additionally, in the activated CSN$^{5H138A}$-$^{N8}$SCF complex, CSN5$^{Ins-1}$ undergoes further remodelling. No longer displaced, it moves back toward the catalytic site, aligning precisely with the C-terminal tail of NEDD8 and enclosing the CUL1$^{K720}$-NEDD8$^{G76}$ isopeptide bond within the CSN5 active site (Fig. 2h and Supplementary Movie 2). The C-terminal section of CSN5$^{Ins-1}$ also transitions from a flexible conformation into a short β-strand, forming a two-stranded antiparallel β-sheet with NEDD8's C-terminal tail. Comparisons with the crystal structure of isolated CSN5 (PDB: 4F7O)[22] highlight the structural versatility of CSN5$^{Ins-1}$ (Supplementary Fig. 18e). In its isolated form, the loop adopts a helical conformation that sterically shields the active site from NEDD8. Furthermore, the overall geometry of the Zn$^{2+}$ and substrate binding sites in CSN5 mirrors that of other MPN+/JAMM family proteases. Parallels emerge when comparing CSN with AMSH-LP bound to di-ubiquitin (PDB: 2ZNV)[47] and ubiquitin-bound RPN11 within a substrate-engaged 26S proteasome complex (PDB: 6MSE)[48,49]. In both structures, the conserved Ins-1 loop adopts a β-hairpin motif that stabilises the β-stranded C-terminal ubiquitin tail for isopeptide cleavage (Supplementary Fig. 19). Notably, the density for CSN5$^{Ins-1}$ in the activated CSN$^{5H138A}$-$^{N8}$SCF position is well-defined, enabling the modelling of most side chains (Supplementary Fig. 18c, d). This activated conformation of CSN5$^{Ins-1}$ is further reinforced by a salt bridge between CSN5$^{E104}$ and WHB CUL1$^{R717}$. Meanwhile, the CUL1$^{K720}$-NEDD8$^{G76}$ isopeptide bond, now positioned within the Zn$^{2+}$-binding site, is stabilised by a hydrogen bond between the NEDD8$^{G76}$ carbonyl oxygen and the hydroxyl group of CSN5$^{S148}$, as well as an interaction between the ε-amino group of CUL1$^{K720}$ and the main-chain carbonyl group of CSN5$^{T105}$ (Fig. 2i). With this final activation step, the isopeptide bond is

perfectly aligned within the CSN5 active site, primed for cleavage and the release of NEDD8 from the CUL1$^{WHB}$ (Fig. 2d, right panel).

To explore the importance of the geometry at the CSN5$^{Ins-1}$ and isopeptide-CUL1$^{WHB}$ interface, we designed and assayed several key mutations. A bulky T105F substitution was added into CSN5$^{Ins-1}$ (Supplementary Fig. 11h, i), while charge-swaps at CUL1$^{WHB}$ residues CUL1$^{R741}$ and CUL1$^{R717}$ (R741E (Supplementary Fig. 11j) and R717E (Supplementary Fig. 11k), respectively) targeted key electrostatic interactions. Incidentally, CUL1$^{R717}$ is highly conserved across Cullins, and structural analysis confirms a conserved positive charge at or very close to CUL1$^{R741}$ (Supplementary Fig. 20). SPR analysis showed that CSN$^{5H138/T105F}$ retained wild-type-like binding to neddylated CUL1/RBX1 (Table 1 and Supplementary Fig. 12i–n), as did $^{N8}$CUL1$^{R741E}$/RBX1 and $^{N8}$CUL1$^{R717E}$/RBX1 when binding CSN$^{5H138A}$. However, in deneddylation assays, CSN$^{5T105F}$ reduced CSN activity by over 50% (Fig. 2j and Supplementary Fig. 13f, g). Due to stability challenges with the CUL1$^{WHB}$ mutants, quantification using our standard assay was not possible. However, Coomassie-stained gel assays confirmed that CSN was less efficient at deneddylating these CUL1$^{WHB}$ mutants compared to wild-type CUL1/RBX1 (Supplementary Fig. 13h). These findings highlight the necessity of precise structural coordination between CSN5$^{Ins-1}$ and the isopeptide-linked CUL1$^{WHB}$ domain for CSN function. While T105F may still allow main-chain hydrogen bonding with the isopeptide, its bulky side chain perturbs the CSN5$^{Ins-1}$ interface with CUL1$^{WHB}$. Likewise, electrostatic interactions between CSN5$^{Ins-1}$ and CUL1$^{WHB}$ are important for stabilising this catalytic interface.

Although ubiquitin and NEDD8 share 59% sequence identity and adopt similar folds[50–52], CSN5 exhibits strict selectivity for NEDD8. The key to this specificity appears to lie in NEDD8's C-terminal tail, which features an alanine at position 72 instead of ubiquitin's arginine. Docking the ubiquitin structure (PDB: 1UBQ)[53] onto the pre-activated CSN$^{5H138A-N8}$SCF complex reveals that while its C-terminal tail aligns well with NEDD8 at the active site, the bulky, charged Ub$^{R72}$ is poorly accommodated within CSN5's hydrophobic pocket (Supplementary Fig. 21a). This distinction becomes even clearer in activated CSN$^{5H138A-N8}$SCF, where NEDD8$^{A72}$ is deeply buried within a hydrophobic channel formed by CSN5$^{YI14}$ and CSN5$^{L157}$, leaving no space for an arginine side chain (Supplementary Fig. 21b). This structural constraint explains CSN5's selectivity for NEDD8 over ubiquitin. Interestingly, NEDD8$^{A72}$ and Ub$^{R72}$ have also been shown to play key roles in selecting the correct E1 enzyme, thereby ensuring correct entry into the appropriate NEDD8 or ubiquitin pathway[50,54].

## Alternative conformational intermediates for CSN-SCF association and dissociation

We previously proposed the E-vict mechanism by which CSN ensures both high specificity and efficient substrate turnover[23]. Central to this process is CSN5$^{E76}$, which initially stabilises the Zn$^{2+}$-bound catalytic water molecule but is displaced by CSN5$^{E104}$ in the autoinhibited state. This exchange weakens CSN's affinity for its product, accelerating its dissociation and preventing product inhibition. The model was validated using a CSN5$^{E104A}$ mutant, which disrupts this ligand exchange, significantly prolonging CSN's residence time on the deneddylated product and leading to a product-inhibited enzyme. Specifically, CSN$^{5E104A}$ binds deneddylated CRL, the reaction product, with more than tenfold higher affinity than wild-type CSN and exhibits an eightfold reduction in the product dissociation rate[23].

To leverage the high-affinity state of this mutant, we mixed CSN$^{5E104A}$ with non-neddylated SCF and subjected the complex to cryo-EM and single-particle analysis, aiming to (i) visualise the product-inhibited enzyme complex, (ii) capture early CSN-SCF association intermediates that would normally be rejected, and (iii) obtain snapshots of CSN during product dissociation. Extensive classification identified five distinct complexes, including CSN$^{apo}$ and four non-neddylated CSN-SCF assemblies (Supplementary Figs. 22 and 23). The

maps vary in the density observed for CUL1$^{WHB}$, RBX1$^{RING}$ and the CSN5-CSN6 heterodimer, indicating conformational flexibility within the system. Intriguingly, while these structures resemble both pre-activated and activated CSN$^{5H138A-N8}$SCF states, they display remarkable conformational diversity, with three conformations showing the N-terminal helical CSN4$^{arm}$ disengaged from CUL1$^{CTD}$ in its "up" conformation. To provide a coherent framework, we describe these structures as dissociation-state-1 through to dissociation-state-4 in a sequence that most plausibly reflects the stepwise mechanism of CSN-SCF dissociation.

Dissociation-state-1 bears the closest resemblance to our activated CSN$^{5H138A-N8}$SCF model, with an RMSD of 1.3 Å (Fig. 3a and Supplementary Fig. 24a, b). In this state, the CSN2$^{arm}$ and CSN4$^{arm}$ maintain their stabilising positions, clamping CUL1$^{CTD}$ and RBX1$^{RING}$. However, in the absence of NEDD8, CUL1$^{WHB}$ is noticeably more flexible, reducing its interaction with RBX1$^{RING}$ (Fig. 3b and Supplementary Movie 3). CSN4$^{arm}$ is still engaged with CUL1$^{CTD}$, and the MPN domains of CSN5 and CSN6 are still in a position comparable to the activated state. Yet, with CSN5 released from NEDD8, flexibility is introduced, resulting in insufficient CSN5-CSN6 density for model building. Given the striking similarity of dissociation-state-1 to the activated complex, we argue that this conformational state represents the earliest step in CSN dissociation from SCF, capturing the post-NEDD8 dissociation phase while largely preserving the activated state conformation.

Dissociation-state-2 represents a significant departure from dissociation-state-1 (Fig. 3c and Supplementary Fig. 24c, d). Among the four states identified, dissociation-state-2 exhibits the lowest resolution at 4.6 Å, likely reflecting a transient intermediate state along the dissociation pathway. Yet, despite its lower resolution, this structure provides valuable insights into dynamic rearrangements within the complex. A defining feature of dissociation-state-2 is the repositioning of CSN4$^{arm}$, which detaches from CUL1$^{CTD}$ and pivots around the CSN4$^{hinge-loop}$ (residues 291–299) to adopt an "up" conformation. (Supplementary Fig. 25a, b). This movement is coupled with a retraction of CSN5 and CSN6, shifting them back to their CSN$^{apo}$ conformation, a transition marked by the re-engagement of a conserved β-hairpin loop, CSN6$^{Ins-2}$, with CSN4 (Fig. 3d)[17]. Interestingly, the final RELION-4.0[55] refined cryo-EM map features weak but discernible density for RBX1$^{RING}$ (Supplementary Fig. 26a). However, post-processing with DeepEMhancer[56] to aid model building in other parts of the map, eliminated this density. To aid interpretation, we manually positioned an RBX1 model into the non-post-processed map and improved its fit using real-space refinement in Coot[57]. This analysis uncovered a striking conformational shift. RBX1$^{RING}$ appears to track the movement of the CSN4$^{arm}$, potentially forming transient interactions with CSN4 via the RBX1$^{RING-insertion}$ (residues 50–70) (Supplementary Fig. 26b and Supplementary Movie 3).

The highest-resolution conformation, dissociation-state-3, at 3 Å (Fig. 3e and Supplementary Fig. 24e, f), represents the most stable of the observed CSN-SCF complexes, as reflected by the consistent local resolution across the complex (Supplementary Fig. 22n). Analogous with dissociation-state-2, CSN4$^{arm}$ is disengaged from CUL1$^{CTD}$ (Supplementary Fig. 27), and CSN5-CSN6 are held in their autoinhibited positions by CSN4 (Supplementary Fig. 28a). However, in dissociation-state-3, RBX1 adopts a third distinct conformation, one that could be confidently built to assign molecular contacts. In this state, RBX1$^{RING}$ undergoes a dramatic rearrangement: the N-terminal stretch of RBX1$^{RING-loop1}$ extends outward, while the RBX1$^{RING}$ core rotates from its activated CUL1$^{WHB}$-CSN2-CSN4-engaged orientation, transitions through its position in dissociation-state-2, before reaching its final, extended conformation in dissociation-state-3, where it establishes an interface with the disengaged CSN4$^{arm}$ (Fig. 3f). This previously uncharacterised interaction increases the binding surface area by over 300 Å$^2$ compared to the RBX1$^{RING}$-CSN4 interface in the activated CSN$^{5H138A-N8}$SCF structure.

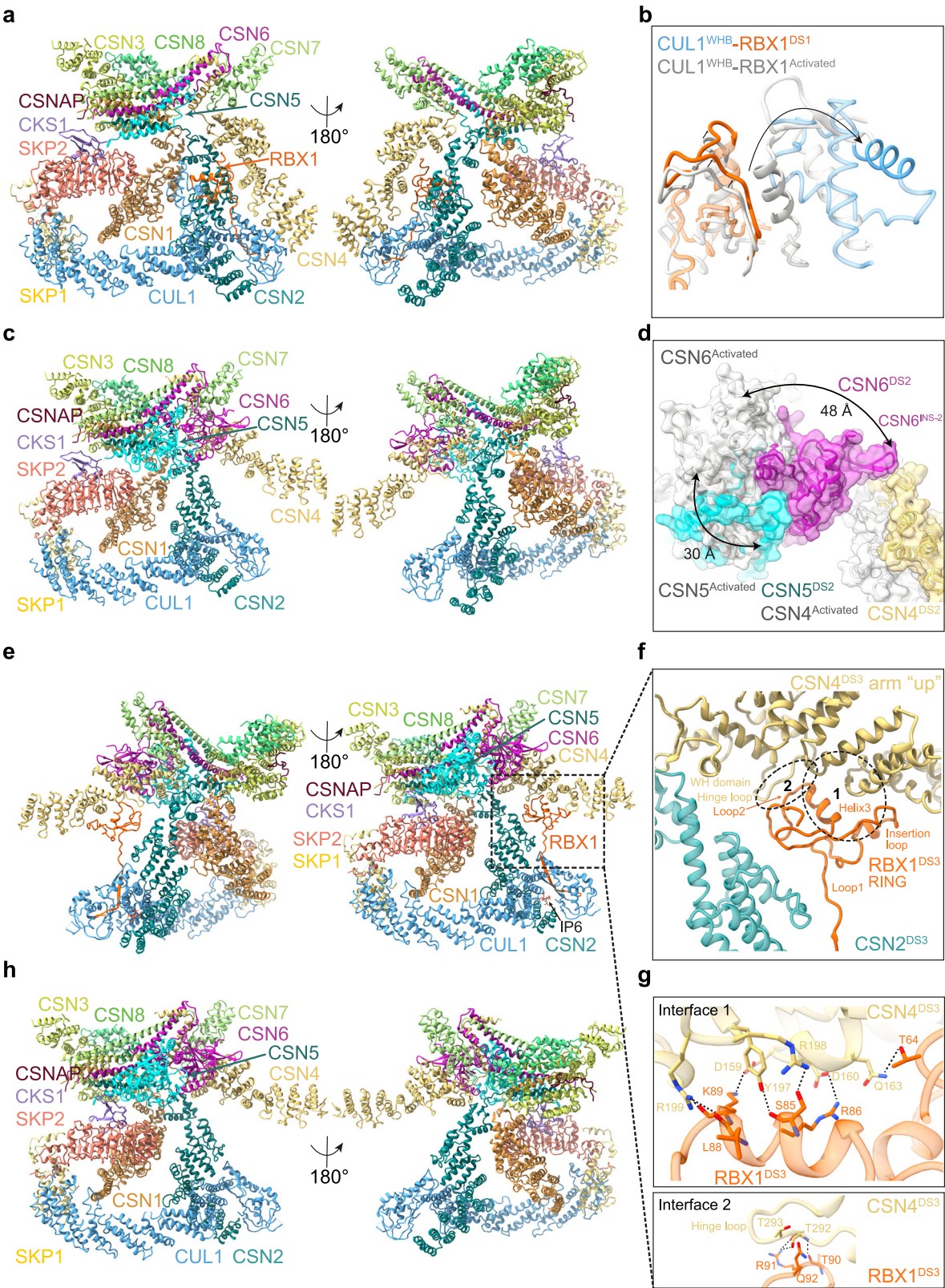

**Fig. 3 | Cryo-EM analysis of four CSN^EI04A-SCF dissociation states. a** Molecular model of CSN^EI04A-SCF in dissociation-state-1 (ribbon representation). **b** Structural comparison of the CUL1^WHB-RBX1^RING interface between activated CSN^5HI38A_N8SCF (grey) and dissociation-state-1 (coloured), highlighting conformational rearrangements upon transition. **c** Molecular model of CSN^EI04A-SCF in dissociation-state-2 (ribbon representation). **d** Conformational changes in the CSN5-CSN6 heterodimer between activated CSN^5HI38A_N8SCF (grey) and dissociation-state-2 (coloured).

**e** Molecular model of CSN^EI04A-SCF in dissociation-state-3 (ribbon representation). **f** In dissociation-state-3, RBX1^RING engages with the CSN4^arm in its "up" conformation. The RBX1^RING-CSN4 interface is divided into two zones: interface 1, which describes RBX1^RING interactions with the CSN4^arm, and interface 2, which describes RBX1^RING interactions with the CSN4^hinge-loop. **g** Close-up views of interface 1 (upper panel) and interface 2 (lower panel) described in (**f**). **h** Molecular model of CSN^EI04A-SCF in dissociation-state-4 (ribbon representation).

The RBX1-CSN4 interface in dissociation-state-3 can be divided into two key interaction zones (Fig. 3f). First, a V-shaped groove formed by RBX1$^{RING-helix3}$ and RBX1$^{RING-insertion}$ creates extensive interactions with CSN4$^{arm}$. These include two salt bridges, CSN4$^{D159}$-RBX1$^{K89}$ and CSN4$^{D160}$-RBX1$^{R86}$, along with hydrogen bonds between CSN4$^{Q163}$-RBX1$^{T64}$ and CSN4$^{R198}$-RBX1$^{S85}$ side chains. Additionally, CSN4$^{R199}$ forms multiple hydrogen bonds with the main-chain carbonyl groups of RBX1$^{L88}$ and RBX1$^{K89}$, while CSN4$^{Y197}$ hydrogen bonds to the carbonyl group of RBX1$^{S85}$. These contacts are further stabilised by multiple van der Waals interactions (Fig. 3g upper panel). Second, RBX1$^{RING-loop2}$ is now capped by the CSN4$^{hinge-loop}$. This CSN4$^{hinge-loop}$, largely disordered across other complexes identified in this study, is remarkably well-defined in structure-3, enabling an additional interaction surface with RBX1$^{RING}$. At this second interface, RBX1$^{R91}$ establishes hydrogen bonds with both the side-chain hydroxyl and main-chain carbonyl group of CSN4$^{T292}$, while the main-chain carbonyl of RBX1$^{T90}$ forms a hydrogen bond with the backbone amide group of CSN4$^{T292}$ (Fig. 3g, lower panel). Collectively, these findings suggest that in dissociation-state-3, the CSN4$^{hinge-loop}$ is effectively "locked" in position and would prevent the CSN4$^{arm}$ from re-engaging with CUL1$^{CTD}$ until RBX1$^{RING}$ shifts out of this state. While we have proposed that structure-1 represents an early stage in the dissociation pathway, it is equally plausible that it could represent a late-stage association step. However, extending this logic to dissociation-state-3 proves more challenging. The inherent asymmetry between CSN association and dissociation, dictated by the presence or absence of NEDD8, introduces a fundamental difference in pathway dynamics. Furthermore, the "locked" CSN4$^{arm}$ conformation in dissociation-state-3 suggests that this state cannot simply be mirrored for association but instead represents a functionally distinct conformation of CSN dissociation.

As the RBX1$^{RING}$ transitions from its conformation adopted during deneddylation to its extended position in dissociation-state-3, there is a significant loss of its prior interactions with CSN2. A direct comparison with our activated CSN$^{5HI38A}$-$^{N8}$SCF model reveals that the only contacts retained in structure-3 involve RBX1$^{W35}$, the final residue of the N-terminal β-strand that integrates into CUL1$^{CTD}$. Here, the indole ring of RBX1$^{W35}$ maintains van der Waals interactions with the main-chain and side chains of CSN2$^{Q297}$ and CSN2$^{E298}$ (Supplementary Fig. 28b). Beyond this, the RBX1$^{RING}$-CSN2 interface is substantially reduced, limited to a single interaction between RBX1$^{R99}$ and CSN2$^{N320}$, in stark contrast to the extensive interactions observed in the activated complex (Fig. 1e). Interestingly, dissociation-state-3 also shows a lateral repositioning of CSN2$^{arm}$, which shifts away from the core of the CSN complex (Supplementary Fig. 29a). Furthermore, superimposing the structures over CSN4 reveals that the RBX1$^{RING}$ position in dissociation-state-3 is incompatible with the activated conformation of CSN$^{5HI38A}$-$^{N8}$SCF, as it would cause a significant clash with CSN2 (Supplementary Fig. 29b). Thus, some repositioning of the CSN2$^{arm}$ in dissociation-state-3 is essential for accommodating the extended RBX1$^{RING}$ conformation.

This lateral movement of the CSN2$^{arm}$ has broader structural implications, causing a related movement of the CSN2-bound CUL1$^{CTD}$. As a result, the entire CUL1 scaffold and its associated SR undergo remodelling within the complex. A striking trend emerges when comparing dissociation-states-1, 2 and 3: there is a clear conformational trajectory in which dissociation-state-2 represents an intermediate state between −1 and −3 (Supplementary Fig. 30a–c). Consequently, the SR occupies an alternative orientation within the CSN complex, where SKP2 and CKS1 subunits are now located within a space formed by the CSN1$^{arm}$, the CSN5 helical bundle and CSN5 MPN domain (Supplementary Fig. 16d). A consequence of this rearrangement is the stabilisation of the SR subunits. While the SR is highly flexible across all other complexes identified in this study, in dissociation-state-3, the formation of a more extensive interface between CSN and the SR results in a higher resolution assembly at this region.

The conformational changes associated with dissociation-state-3 bring the C-terminal end of the SKP2 LRR domain closer to CSN1$^{arm}$ (Supplementary Fig. 16e). This proximity allows for the formation of a small interface, potentially stabilised by a salt bridge between CSN1$^{E233}$ and SKP2$^{K363}$ (Supplementary Fig. 16f). In parallel, CKS1 is repositioned between the CSN5 helical bundle and the CSN5 Ins-2 loop (CSN5$^{Ins-2}$: residues 197–219) (Supplementary Fig. 16g). At the helical bundle, CSN5$^{K299}$ and CSN5$^{Q275}$ form hydrogen bonds with the main-chain carbonyl groups of CKS1$^{Y8}$ and CKS1$^{I6}$, respectively, and are further stabilised by van der Waal interactions involving CSN5$^{A278}$, CSN5$^{Q279}$ and CKS1$^{Y7}$ (Supplementary Fig. 16h). The strongest contact between CSN5$^{Ins-2}$ and CKS1 is mediated by a dual salt bridge between CSN5$^{E212}$ and CKS1$^{R20}$, along with additional van der Waals interactions between CSN5$^{V216}$ and CKS1$^{R71}$. CSN5$^{Ins-2}$ has a lesser-understood role in CSN activation, and while the discovery of its interface with the SR in structure-3 adds another layer of complexity to its function, further work will be required to determine its functional significance.

The metabolite IP6 is known to bind a distinct surface patch on CSN2, where a cluster of conserved lysine residues interact electrostatically with its phosphate groups[26–28]. In every complex analysed in this study, we identified additional density in our cryo-EM maps that could not be assigned to CSN or SCF. However, this density precisely matched the location and geometry of IP6 in the CSN2 co-crystal structure (PDB: 6A73)[28], suggesting that the extra density was indeed IP6 and was likely to have been co-purified from insect cells. Analysis of the high-resolution dissociation-state-3 structure pinpoints three residues from CUL1/RBX1 in the vicinity of IP6: RBX1$^{K25}$ and RBX1$^{K26}$ and CUL1$^{K522}$ (Supplementary Fig. 31). Notably, RBX1$^{K25}$ and RBX1$^{K26}$ are conserved in RBX2 (Supplementary Fig. 15a) and have previously been implicated in modulating CSN-CUL4A/RBX1 interactions in cells[28]. Therefore, our study reinforces the notion that IP6 serves as a molecular bridge between CSN, and the sequence variations found across all human Cullins. Beyond this, our structural insights provide a valuable foundation for designing CSN-CRL molecular glue-type inhibitors that could selectively regulate complex formation and function.

The final CSN$^{E104A}$-SCF dissociation complex, dissociation-state-4 (Fig. 3h and Supplementary Fig. 24g, h), closely resembles dissociation-state-3 (Supplementary Fig. 32a) with no significant changes identified in the CSN1, CSN2, CSN3 and CSN8 subunits, or the CUL1 and SR components. However, one prominent difference emerges in the lack of density for RBX1$^{RING}$, only identifiable at very low contour levels in the non-post-processed map (Supplementary Fig. 26c). A second key difference occurs in the CSN4$^{arm}$, which undergoes a subtle yet consequential shift relative to CSN2. Pivoting around the CSN4$^{hinge-loop}$, CSN4$^{arm}$ adopts a "fully up" conformation, moving further from the CSN2$^{arm}$ (Supplementary Fig. 32b). This repositioning disrupts the critical CSN4-RBX1$^{RING}$ interface observed in structure-3, suggesting that the RING can no longer extend the extra distance required to maintain contact with CSN4. This CSN4$^{arm}$ shift is accompanied by a coordinated movement of the CSN5-CSN6 heterodimer, ensuring the preservation of the essential interface between CSN4 and CSN6$^{Ins-2}$ (Supplementary Fig. 32c).

The final structure we determined in this dataset, CSN$^{apo}$, captures the fully dissociated state of the complex (Supplementary Fig. 24i, j). We built an atomic model that closely resembles the previously reported crystallographic structure[17] with the notable addition of CSNAP. Expectedly, regions corresponding to flexible elements exhibit greater variability compared to the crystal structure of 8-subunit CSN$^{apo}$ (PDB: 4D10)[17], consistent with a structure in solution (Supplementary Fig. 33a). We were also intrigued to compare 8-subunit CSN$^{apo}$ to the regions of the complex that bind with CSNAP. While the overall architecture of CSN3 and CSN8 remains largely conserved between the two complexes, subtle variations hint at potential structural adaptations induced by CSNAP binding (Supplementary Fig. 33b, c).

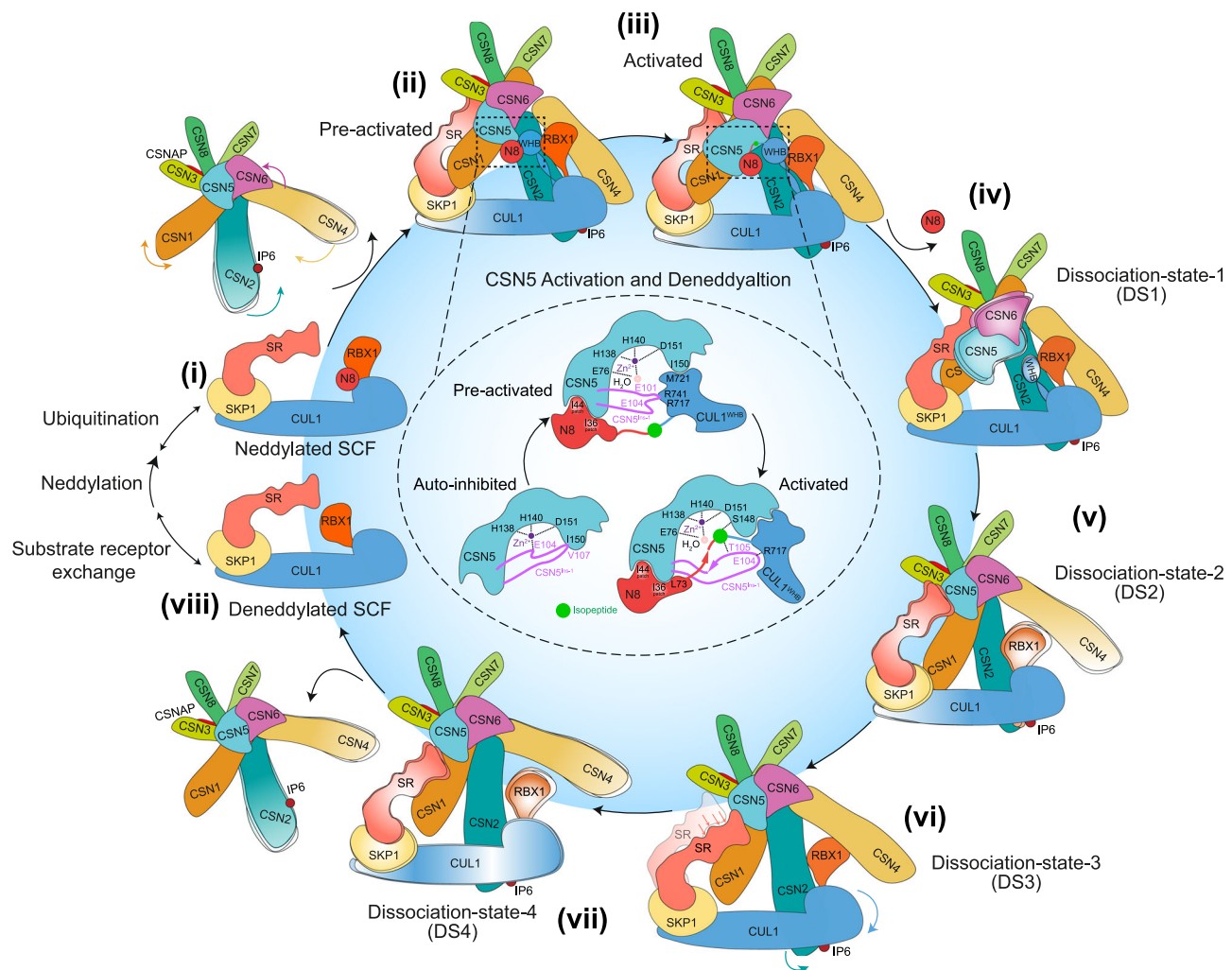

**Fig. 4 | Schematic of CSN-mediated SCF deneddylation mechanism.** CSN binding to neddylated SCF (**i**) triggers a series of conformational rearrangements in both complexes, resulting in (**ii**) a pre-activated state. In this high-affinity intermediate, the active site is sterically accessible and catalytically competent. However, the isopeptide bond remains misaligned (inset). A further set of local rearrangements in NEDD8, CUL1$^{WHB}$, and CSN5$^{Ins-1}$ produces (**iii**) the activated state, where the C-terminal tail of NEDD8 is repositioned and forms an antiparallel β-sheet with the remodelled CSN5$^{Ins-1}$, precisely aligning the isopeptide bond within the active site (inset). Following NEDD8 cleavage, the SCF complex dissociates from CSN through a series of four distinct intermediates (**iv**–**vii**, dissociation states DS1–DS4). In DS1 (**iv**), NEDD8 departure induces increased flexibility in CSN5-6 and CUL1$^{WHB}$. Next, CSN4 rotates upward and re-engages CSN6, stabilising the CSN5-6 heterodimer (DS2, **v**), while RBX1$^{RING}$ detaches from CUL1$^{WHB}$ and adopts a flexible conformation. In DS3 (**vi**), RBX1$^{RING}$ rebinds the CSN4$^{arm}$ in its "up" orientation, and CUL1 shifts away from the CSN core, repositioning its SR between CSN1 and CSN5. In DS4 (**vii**), CSN4 moves further upward and releases RBX1, ultimately leading to complete dissociation of the deneddylated SCF (**viii**), which is now poised for substrate receptor exchange, substrate binding and reactivation through neddylation.

## Discussion

In this study, we reconstituted and structurally characterised key intermediates in the CSN-mediated deneddylation cycle of SCF. We captured a comprehensive set of high-resolution structures representing distinct mechanistic stages, including CSN association, catalytic activation, and product dissociation (Fig. 4 and Supplementary Movie 4). The pre-activated conformation represents an initial docking state in which CSN is poised for catalysis but remains autoinhibited. The activated conformation reveals a fully engaged, catalytically competent complex, in which CSN5$^{Ins-1}$ undergoes a conformational rearrangement to position the isopeptide bond of $^{N8}$CUL1$^{WHB}$ in the active site. Four additional structures capture a continuum of dissociation intermediates, with RBX1$^{RING}$ playing a critical role in stabilising key interfaces throughout the release process.

Our results build upon and significantly expand prior mechanistic studies of CSN-mediated regulation of CRLs, encompassing CRL1[23], CRL2[24], CRL3[25] and CRL4[15]. A unifying theme across these studies is the conserved role of CSN2 and CSN4 in clamping both the CUL1$^{CTD}$ and RBX1$^{RING}$, along with the release mechanism of the CSN5/CSN6 heterodimer from CSN4. In our work, these key interfaces are resolved at near-atomic resolution, providing additional structural insight. Notably, our study structurally characterises the interface between RBX1$^{RING}$ and the CUL1$^{WHB}$ domain, an interaction essential for positioning NEDD8 in a catalytically competent configuration, and one that we have now experimentally validated. Faull et al. previously proposed that NEDD8 triggers remodelling of the CSN5 active site[24]. Our structural data support and extend this model, showing that although NEDD8 facilitates CSN5 activation indirectly, it is the interaction between CSN5 and the CUL1$^{WHB}$ domain that directly drives remodelling of the active site, initiating a two-step mechanism. Concerning the dissociation phase, our study presents the most comprehensive structural characterisation to date of CSN-CRL disengagement. The four conformational states resolved from a complex assembled with the product-inhibited mutant CSN5$^{E104A}$, which is known to bind

deneddylated CRLs with over tenfold higher affinity and significantly reduced dissociation kinetics compared to wild-type CSN[23], likely represent a post-catalytic ensemble transitioning toward complex dissociation. While we favour the interpretation that these structures correspond to sequential dissociation intermediates, we explicitly acknowledge that some may additionally reflect early association states, given the inherent reversibility of protein-protein interactions. Nonetheless, the defined assembly conditions, high-resolution features, and specificity of the CSN-SCF interactions observed in the absence of NEDD8 support their classification as bona fide dissociation intermediates. Ultimately, complementary approaches such as time-resolved cryo-EM[58–60] will be essential to fully elucidate the temporal dynamics of the CSN−CRL regulatory cycle.

## Methods

### Cloning, protein expression and purification of human CUL1/RBX1 and mutations

All oligonucleotides were synthesised by Merck Life Science, as listed in Supplementary Table 3. N-terminally tagged $^{StrepII2x}$CUL1 and RBX1 were subcloned into the baculovirus vector pFBDM[61]. Mutagenesis on CUL1/RBX1 was performed by Q5 Site-Directed Mutagenesis (NEB). CUL1/RBX1 complexes were expressed and purified as described by Enchev et al.[62].

### Neddylation assay

The NEDD8-activating enzyme NAE (GST-APPBP1 and UBA3), GST-UBE2M (UBC12), NEDD8 with a C-terminal StrepII$^{2x}$-tag and N-terminally tagged StrepII$^{2x}$-DEN1 were expressed in BL21 (DE3) *E. coli* cells and purified as described by Enchev et al.[14].

The neddylation assay, adapted from Enchev et al.[62], was initially performed in a 100 µl reaction volume. The reaction included 50 mM Hepes pH 7.5, 150 mM NaCl, 2.5 mM MgCl$_2$, 1.25 mM ATP, 0.7 µM NAE1, 3.6 µM UBC12, 12 µM NEDD8 and 8 µM StrepII2x-CUL1/RBX1. Reactions were incubated at 37 °C for 30 min. To terminate the reaction, 50 mM DTT was added, effectively quenching the activity of the enzymes and preventing further NEDD8 conjugation. Mono-neddylation was achieved by the addition of 0.3 µM DEN1, followed by a further incubation of 30 min. Purification of the neddylated complex utilised the StrepII$^{2x}$ tag of CUL1/RBX1, absent on other proteins in the mix. The reaction sample was bound to a Strep-Tactin Superflow Cartridge (QIAGEN) and washed with a buffer containing 50 mM Hepes pH 7.5, 150 mM NaCl and 10 mM DTT. Neddylated CUL1/RBX1 was eluted with the same buffer supplemented with 2.5 mM d-desthiobiotin and analysed via SDS-PAGE. The elution was concentrated and then purified by gel filtration on a Superdex 200 Increase 10/300 GL column (Cytiva) in 15 mM Hepes pH 7.4, 120 mM NaCl, and 0.5 mM DTT.

### Cloning, protein expression and purification of 9-subunit CSN and mutagenesis

CSN1, CSN2, StrepII$^{2x}$-CSN3 and His$_6$-CSN5 were cloned into a single pFBDM vector, while CSN4, CSN6, CSN7b and CSN8 were inserted into a second pFBDM, as described by Mosadeghi et al.[23]. CSNAP was inserted into an empty pFBDM vector, the expression cassette excised with PmeI and AvrII (NEB) and then transferred to BstZ17I/SpeI linearised pFBDM$^{CSN1/2/StrepII2x-CSN3/His6-CSN5}$. The insertion of CSNAP into the plasmid was confirmed by DNA sequencing (Full circle). Point mutations on CSN5 were made by digesting pFBDM$^{CSN1/2/StrepII2x-CSN3/His6-CSN5/CSNAP}$ with SacII and Bsu36I to produce two fragments. A 3217 bp fragment containing CSN5 was used as a template for mutagenesis by PCR to produce two smaller fragments depending on the point mutation site. The larger backbone fragment of 9301 bp was left unchanged. The ligation of the resulting three fragments was performed using the NEBuilder HiFi DNA Assembly Cloning Kit (NEB). CSN was expressed and purified as described by Mosadeghi et al.[23].

### CSN$^{5H138A}$-$^{N8}$SCF assembly and CSN$^{5E104A}$-SCF assembly

SKP1/SKP2 were co-expressed and purified as described by Enchev et al.[14]. To purify CKS1, a protocol was adapted from previously published work, where GST-CKS1 was expressed in BL21(DE3) cells[33]. SKP1/SKP2 and CKS1 were mixed with neddylated CUL1/RBX1 at a ratio of 1.2:1.2:1 to form $^{N8}$SCF, while non-neddylated CUL1/RBX1 was mixed at the same ratio to form SCF[63].

To form the CSN$^{5H138A}$-$^{N8}$SCF complex, $^{N8}$SCF was mixed with CSN$^{5H138A}$ at a ratio of 1.2:1 at 6.5 µM final concentration, followed by a 10 min incubation. The sample was purified using size exclusion chromatography with a Superose 6 Increase 10/300 GL column (Cytiva) in 15 mM Hepes pH 7.5, 120 mM NaCl and 0.5 mM DTT. To form the CSN$^{5E104A}$-SCF complex, SCF was mixed with CSN$^{5E104A}$ at a ratio of 1.2:1 at 6.5 µM final concentration, followed by a 10 min incubation.

### Cryo-EM sample preparation and data collection

The CSN$^{5H138A}$-$^{N8}$SCF and CSN$^{5E104A}$-SCF complexes were diluted to 3.3 mg/ml in 25 mM Hepes pH 7.5, 150 mM NaCl, 1 mM TCEP and 0.1% octyl glucoside. Cryo-EM holey UltrAUfoil grids (Quantifoil) were glow-discharged for one minute at 40 mA in air. Four microlitres of complex were placed on a grid, then blotted for 4.5 s (blot force −10) using a Vitrobot mark IV (Thermo Fisher Scientific) at 4 °C and 100 % humidity. The grids were plunge-frozen in liquid ethane.

Data was collected using a Titan Krios cryo-EM (Thermo Fisher Scientific), operated at 300 kV, equipped with a Falcon III detector. Micrographs were auto-collected with EPU software (Thermo Fisher Scientific) in counting mode with 32 frames per movie and nominal magnification 96,000×, (1.08 Å per pixel). The total electron dose was 47 electrons per Å$^2$, with defocus range from −0.5 to −5 µm.

### Cryo-EM data processing

Micrographs were motion-corrected in RELION-4.0 with MotionCor2, 5 by 5 patches[64,65]. CTF estimation was done by GCTF[66]. Particle picking was done in crYOLO using a purpose-trained model[67].

8,107,132 particles were extracted in RELION-4.0 with 2 times binning and 160 box size from the $^{N8}$SCF-CSN$^{5H138A}$ complex dataset, and 6,955,138 particles from SCF-CSN$^{5E104A}$ complex dataset. Particles were transferred to cryoSPARC v3.3.2 for several runs of 2D classification and ab-initio reconstruction to remove junk particles[68]. Subsequent particles were transferred back to RELION-4.0 for re-extraction without binning (320 box size), 3D refinement, CTF refinement and Bayesian polishing. CryoDRGN v1.0.0 and RELION-4.0 were applied for heterogeneous cryo-EM reconstruction[32,55]. The cryoDRGN-generated maps informed the design of local masks centred around the NEDD8 region. Focused classification was done with RELION-4.0. Final reconstructions were sharpened by DeepEMhancer[56]. The 3D Fourier Shell Correlation (3D FSC) analysis was done using the Remote 3DFSC Processing Server (https://3dfsc.salk.edu/upload/)[69]. The data processing flowchart is shown in Supplementary Figs. 1, 2 and 22. Comprehensive statistical information regarding all the cryo-EM datasets and structures are available in the Supplementary Tables 1 and 2.

### Model building and refinement

Rigid body fitting was performed in UCSF Chimera v1.16 and ChimeraX v1.5[70,71]. Initial models for atomic model building were obtained as follows; CSN1-8 and NEDD8 from CSN-NEDD8/CUL2/RBX1/ELOB/C (PDB: 6R7I)[24], CUL1/RBX1/SKP1/SKP2 from CUL1/RBX1/SKP1/SKP2 (PDB: 1LDK)[34], and CKS1 from SKP1/SKP2/CKS1 (PDB: 2ASS)[33]. Coot v0.9.6 was used for manual fitting and model building[57]. Flexible fitting was performed using Namdinator (https://namdinator.au.dk/). The atomic models were subjected to multiple rounds of real-space refinement in Phenix v1.21.1[72]. All models were validated using the MolProbity server (http://molprobity.biochem.duke.edu/) and wwPDB Validation (https://deposit-pdbe.wwpdb.org/). All the structural

figures were prepared using ChimeraX[71]. Schematic cartoons generated for Figs. 1d and 4 were adapted from Shaaban et al. (open access)[42].

### In vitro deneddylation assay

The in vitro neddylation assay is adapted from a previously published protocol[23,43]. His-PKA-N8, tagged with MHHHHHHHHHRRGSL at the N-terminus, was used for CUL1 neddylation and deneddylation assays. 6 μM neddylated $^{His-PKA-N8}$CUL1 was labelled with $^{P32}$ATP (Hartmann Analytic) by PKA (cAMP-dependent protein kinase, NEB) at 30 °C for 60 min. The reaction was quenched by placing on ice.

Deneddylation assays were performed in a buffer containing 25 mM Tris-HCl, pH 7.5, 100 mM NaCl, 25 mM trehalose (Sigma-Aldrich), 1 mM DTT, 1% (v/v) glycerol, 0.01% (v/v) Triton X-100 (Sigma-Aldrich) and 0.1 mg/ml ovalbumin (Sigma-Aldrich). 800 nM $^{P32-His-PKA-N8}$CUL1/RBX1 was cleaved with 2 nM CSN at 24 °C. Samples were taken at predetermined times (15, 30, 45, 60, 75, 90 and 120 s) and the reaction was stopped with the addition of reducing SDS-PAGE buffer. Samples were analysed on a 4–12% gradient SDS-PAGE gel (Thermo Fisher). SDS-PAGE gels were dried and subjected to overnight exposure on a phosphor screen. Imaging was performed using a Typhoon FLA 9500 (Cytiva). The resulting images were analysed with ImageJ[73]. The ratio of the radioactive signal from free HIS-PKA-N8 to the total radioactive signal (comprising both free HIS-PKA-N8 and His-PKA-N8-CUL1) was calculated for the various time points, revealing a linear relationship. The deneddylation activity was represented by the slope of the linear regression curve. Each reaction was performed in triplicate, and the standard deviation was calculated accordingly. The data were plotted using Prism 10 (GraphPad). To compare the differences in activity among various CSN and CUL1/RBX1 mutations, the data were normalised against the activity of CSN$^{5WT}$.

For CUL1$^{R741E}$/RBX1 and CUL1$^{R717E}$/RBX1, the neddylation assay was performed with unlabelled NEDD8. The deneddylation assay was performed at the same condition as $^{P32-His-PKA-N8}$CUL1/RBX1. Gels were stained with SimplyBlue SafeStain (Thermo Fisher).

### Surface plasmon resonance (SPR)

SPR experiments were performed on a Biacore S200 instrument (Cytiva) using a Twin-Strep-tag® capture approach. Measurements were conducted in HBS-P⁺ running buffer, supplemented with TCEP (10 mM HEPES pH 7.4, 150 mM NaCl, 0.05% Tween 20 and 1 mM TCEP) at 25 °C. A Strep-Tactin coated CM5 chip was prepared using the Twin-Strep-tag® Capture Kit for SPR (IBA Lifesciences) according to the manufacturer's instructions.

StrepII$^{2x-N8}$CUL1/RBX1 and StrepII$^{2x}$-CUL1/RBX1 were separately immobilised on a sensor chip at concentrations of 25 nM and 2.5 nM, respectively, for 60 s at a flow rate of 30 μl/min, achieving a surface density of approximately 100 response units (RU). The protein CSN$^{5H138A}$ was subjected to buffer exchange into HBS-P+ running buffer, enhanced with 1 mM TCEP. Following a 300-s stabilisation period, increasing concentrations of CSN$^{5H138A}$ at 0.625, 1.25, 2.5, 5.0, 10.0, 20.0, 40.0, 80.0 nM for StrepII$^{2x-N8}$CUL1/RBX1, and 40.0, 80.0, 160, 320, 640, 1280, 2560, 5120 nM for StrepII$^{2x}$-CUL1/RBX1, were injected in a single cycle with a contact time of 80 s (StrepII$^{2x-N8}$CUL1/RBX1) and 180 s (StrepII$^{2x}$-CUL1/RBX1), followed by a 600-s dissociation period at a flow rate of 30 μl/min. After each run, the Strep-Tactin chip was regenerated with two 60-s injections of 3 M guanidine hydrochloride and two 30-s injections of 10 mM NaOH/500 mM NaCl, both at 10 μl/min. Single-cycle kinetics sensorgrams were processed in Biacore S200 Evaluation Software v1.0 (Cytiva). Responses were double-referenced (reference-flow-cell subtraction plus blank buffer injections), and baselines were aligned. The steady-state report point was defined as the mean response over the final 3–5 s of each concentration step (after signal stabilisation and before the valve switch), yielding Req values for each analyte concentration C. The steady state affinity is determined from a plot of the steady state response Req against concentration C. Req-C data were fit by nonlinear least squares to a 1:1 Langmuir (one-site hyperbolic) steady-state model and replotted in GraphPad Prism Version 10.0. The kinetic analysis was based on either a 1:1 Langmuir binding model (a, c) or a 1:1 binding model with drift correction, in both cases Rmax was fitted globally.

### Ethical approval

This study does not involve human participants, animal experiments, or clinical data. All experimental protocols involving recombinant protein expression were conducted under institutional biosafety guidelines.

### Reporting summary

Further information on research design is available in the Nature Portfolio Reporting Summary linked to this article.

### Data availability

The cryo-EM density maps generated in this study have been deposited in the Electron Microscopy Data Bank (EMDB) under accession codes EMD-53252 (pre-activated CSN$^{5H138A-N8}$SCF), EMD-53253 (activated CSN$^{5H138A-N8}$SCF), EMD-53254 (CSN$^{E104A}$-SCF dissociation-state-1), EMD-53255 (CSN$^{E104A}$-SCF dissociation-state-2), EMD-53256 (CSN$^{E104A}$-SCF dissociation-state-3), EMD-53257 (CSN$^{E104A}$-SCF dissociation-state-4), and EMD-53258 (9-subunit CSN$^{apo}$). The corresponding atomic coordinates have been deposited in the Protein Data Bank (PDB) under accession codes 9QO0 (pre-activated CSN$^{5H138A-N8}$SCF), 9QO1 (activated CSN$^{5H138A-N8}$SCF), 9QO2 (CSN$^{E104A}$-SCF dissociation-state-1), 9QO3 (CSN$^{E104A}$-SCF dissociation-state-2), 9QO4 (CSN$^{E104A}$-SCF dissociation-state-3), 9QO5 (CSN$^{E104A}$-SCF dissociation-state-4), and 9QO6 (9-subunit CSN$^{apo}$). The SPR and deneddylation assay data generated in this study are provided in the Supplementary Information and Source Data file. Source data are provided with this paper.

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

## Acknowledgements

We thank members of the Enchev lab for their comments and suggestions throughout this work. We are grateful to members of the Structural Biology STP: Andrea Nans for cryo-EM data collection and assistance with computing, Philip Walker, Donald Benton, and Andrew Purkiss, as well as members of the Scientific Computing STP for software and computer support. Work in the Enchev laboratory was supported by the Francis Crick Institute, with funding from Cancer Research UK (CC2059), the UK Medical Research Council (CC2059), and the Wellcome Trust (CC2059). For the purpose of Open Access, the author has applied a CC BY public copyright license to any Author Accepted Manuscript version arising from this submission.

## Author contributions

S.D. expressed and purified proteins with help from J.A.C. S.D. and R.I.E. prepared cryo-EM samples and collected data. S.D. processed cryo-EM data with help and guidance from M.-E.M. and R.I.E. Atomic models were built and refined by S.D. and J.A.C. with initial help from M.S. S.D., J.A.C. and R.I.E. performed structural analysis and designed validation experiments. Biochemical and kinetic experiments were performed and analysed by S.D. with help from S.K. R.I.E. conceived and supervised the research. S.D., J.A.C. and R.I.E. wrote the manuscript with input from all authors.

## Funding

## Competing interests

The authors declare no competing interests.
