## [Transparent Peer Review file · Nature Communications]

Structural Basis of CSN-mediated SCF Deneddylation

Corresponding Author: Dr Radoslav Enchev

Version 0:

Reviewer comments:

Reviewer #1

(Remarks to the Author)

The CSN signalosome plays an important role in regulating both the neddylation and deneddylation states of Cullin-RING ligases (CRL). Enchev and colleagues reconstituted and applied cryo-EM analysis on CSN-N8SCF complex, revealing distinct functional states and capturing critical intermediates involved in the deneddylation cycle. Their work highlights the conformational rearrangements involving the CSN5 Ins-1 loop, the RBX1 RING domain and the neddylated Cullin WHB domain in achieving catalytic activation. They further detailed the sequential release of the CSN from the deneddylated product. Additionally, they also provided the structural positioning of CSNAP, which integrates into a groove formed by CSN3 and CSN8. Overall, the structures are presented with clarity and the work provides new insights into this biological process. However, the following concerns should be addressed before acceptance for publication.

- In Line 144, the PDB identifier 6R7I corresponds to the structure of the CSN bound to CRL2, not the apo CSN. Please clarify this.
- Line 177 and Extended Data Fig. 2a: the authors state that CSNAP engages with the C-terminal helical bundle of CSN1. However, upon examining the structure, it is evident that CSNAP does not directly interact with the helical bundle. Instead, it engages with the loop regions following the helical bundle from the CSN1 C-terminus. Please revise the text to reflect this distinction accurately.
- Lines 183-186: please specify the residues within CSN3 and CSN8 that contribute to the hydrophobic and hydrogen bonding interactions with CSNAP. Including these details will help readers better understand the content and align the text more effectively with the corresponding figures.
- Line 220, please state which structure(s) was used to compare with the pre-activated CSN-5H138A-N8-SCF.
- Line 295-310, from the pre-activated CSN-N8-SCF structure, the author hypothesised that disrupting the interface between RBX1-RING and CUL1-WHB would impair CSN activity, therefore they introduced mutations K726A/Q728A on CUL1-WHB and applied them for further SPR/in vitro deneddylation assays. First, isn't it more reasonable to test the interface between CSN2 and CSN4 with RBX1. Second, could the author examine the structure based on activated state and design the experiment.
- Line 295-310: based on the pre-activated CSN-N8-SCF structure, the authors hypothesized that disrupting the interface between RBX1-RING and CUL1-WHB would impair CSN activity, leading to the introduction of K726A/Q728A mutations in CUL1-WHB, followed by SPR and in vitro deneddylation assays. First, wouldn't it be more reasonable to test the interface between CSN2 and CSN4 with RBX1 instead? This could provide additional insights into the interaction critical for CSN activity.
- In line 329, the authors propose that residue E122 in CSN5 contributes to interactions with NEDD8. However, the observation that the CSN5 H138A/E122K mutant still retains binding to N8-CRL1 (albeit with reduced activity) raises questions about the functional role of E122, particularly given its position far from the Ins-1 loop and the catalytic site. Similarly, the basis for the reduced activity in the L157G and F161G mutants remains unclear. Could the authors experimentally test whether the E122K, L157G, or F161G mutations affect stabilization or any other functional roles?
- In line 457, the authors state that "CSN5 H138A/T105F reduced CSN activity by over 50%." However, it is unclear whether they used the CSN5 H138A/T105F double mutant or just the CSN5 T105F single mutant for the assay. Given that the CSN5 H138A mutation is known to abolish deneddylation activity, this clarification is essential. Additionally, even though the CSN5 H138A mutant reportedly loses activity, could the authors still use this mutant to perform the same deneddylation activity assay for consistency and comparison?
- In Extended Data Fig. 9, the results presented for the deneddylation assay are incomplete. All CSN mutations indicated in panel c should be supplemented with SDS-PAGE (4–12%) analysis, coupled with phosphate radioactive imaging, to provide a more comprehensive and detailed dataset, similar in panel a and b.

- For the cryo-EM analysis, please provide the models to map the Fourier Shell Correlation (FSC) for the individual structures. Including this data will enhance the transparency and rigor of the structural analysis. Have the authors tried local refinement on the N-terminal regions of CUL1, SKP1 and SKP2 to increase the quality of the maps?
- Could the authors summarize the discoveries presented in this manuscript alongside the previously reported findings on the CSN-neddylated CRL2 complex (PMID: 31444342)? Adding a comparative summary in the discussion section would help provide a more cohesive understanding of the CSN working mechanisms against diverse CRL.
- Please provide marker in Extended Data Fig1a and Fig17a.
- Given that the dimensions of the protein assemblies described in this paper are approximately 150 × 170 angstroms, Please confirm whether the scale bars in Extended Data Fig. 1c and Extended Data Fig. 17 are accurate.
- Please indicate the corresponding figures between line 252-263. It is easier to follow the text and figures.
- Extended Data Fig.8n "Rbx1" should be "RBX1"; Fig 2J, "Csn5" should be "CSN5"; ED Fig2c, "csn8" should be "CSN8".
- The figure legend for Extended Data Fig 7e does not match the label in the figure.
- In Extended Data Fig. 8d and 8j, the reported affinities should be similar, as the M117R mutation was specifically designed to disrupt the CSN5-NEDD8 interaction. Since non-neddylated protein is used in this context, the mutation is not expected to influence the affinity. Please clarify or address this discrepancy.
- Line 343-344, while CSN5 E122 does not directly mediate NEDD8 binding, it likely plays a crucial role in stabilizing NEDD8 for efficient catalysis, possibly through interactions with NEDD8 H68. This sentence is confusing. Please revise.

Reviewer #2

(Remarks to the Author)

Reviewer #3

(Remarks to the Author)

Ding and colleagues present a comprehensive structural analysis of the COP9 signalosome (CSN) interaction with its substrate, the SCF ubiquitin ligase complex. Using cryo-electron microscopy, the authors have captured multiple states of this molecular machinery, including pre-activated, activated, and four distinct dissociation intermediates. Their findings provide unprecedented insight into the mechanisms of CSN-mediated deneddylation, including the first structural visualization of CSNAP, a previously uncharacterized component of the CSN complex. While the structural work is a real tour de force, I have several significant concerns about the data interpretation and experimental validation, but I would still recommend publication once they are addressed

Major Concerns:

1. The authors designate structures as "pre-activated," "activated," and "dissociation" states with insufficient experimental evidence. The assignment of these states to a sequential reaction coordinate is largely speculative and not supported by kinetic data. The authors provide no compelling evidence that these aren't simply different conformations of the complex that exist in equilibrium or, in some cases, potentially damaged particles. Without kinetic experiments or complementary biophysical approaches, the sequential nature of these states remains unproven, undermining the mechanistic model.
2. Despite providing detailed structural information about CSNAP's integration into the CSN complex, the authors present no functional data whatsoever. Critical questions remain entirely unaddressed: Does CSNAP affect deneddylation activity? Does it alter binding kinetics? Does it influence complex stability? A basic comparison of 8-subunit versus 9-subunit CSN would provide essential functional context for the structural observations.
3. The cryo-EM maps show clear signs of preferred orientation, as evident from the angular distribution plots. While DeepEMhancer has been used for post-processing, the potential artifacts introduced by preferred orientation are hidden to some extent. But they should still be adequately acknowledged or addressed. A 3D FSC analysis would help quantify anisotropy in the maps. Moreover, the local resolution estimation appears overly optimistic, particularly in peripherally located regions where resolution is likely significantly lower than 5 Å as indicated.
4. The methods section lacks critical details regarding image processing decisions. The implementation of cryoDRGN is inadequately described - specifically, how particles were selected for different classes, what criteria determined inclusion or exclusion, and whether additional conformational states might exist in the dataset but were not pursued. What about the other clusters? These details are essential for evaluating the robustness of the structural classifications.
5. The modeling of the CSN5Ins-1 loop in the pre-activated state is problematic. This region shows fragmented density yet serves as the basis for significant mechanistic interpretations. Given the central importance of this loop to the authors' mechanistic arguments, a more rigorous approach to modeling this region is required, with explicit acknowledgement of the limitations imposed by map quality.

Version 1:

Reviewer comments:

Reviewer #1

(Remarks to the Author)

I am happy that the authors have addressed my concerns, therefore I recommend publication in Nature Communications.

Reviewer #3

(Remarks to the Author)

The authors have addressed all my concerns sufficiently and extensively and clarified everything. Thus, I recommend publication.

Reviewer #1 (Remarks to the Author):

The CSN signalosome plays an important role in regulating both the neddylation and deneddylation states of Cullin-RING ligases (CRL). Enchev and colleagues reconstituted and applied cryo-EM analysis on CSN-N8SCF complex, revealing distinct functional states and capturing critical intermediates involved in the deneddylation cycle. Their work highlights the conformational rearrangements involving the CSN5 Ins-1 loop, the RBX1 RING domain and the neddylated Cullin WHB domain in achieving catalytic activation. They further detailed the sequential release of the CSN from the deneddylated product. Additionally, they also provided the structural positioning of CSNAP, which integrates into a groove formed by CSN3 and CSN8. Overall, the structures are presented with clarity, and the work provides new insights into this biological process.

We thank the reviewer as well as the early career co-reviewer from their lab for the thoughtful and constructive summary of our work, and for the overall positive and encouraging feedback. We greatly appreciate your comments, which have helped us further refine and strengthen the manuscript. We have carefully addressed all points in the detailed responses below.

However, the following concerns should be addressed before acceptance for publication.

- In Line 144, the PDB identifier 6R7I corresponds to the structure of the CSN bound to CRL2, not the apo CSN. Please clarify this.**

Thank you for pointing this out. We apologise for the oversight. In our analysis, we only used the CSN model from the structure deposited as PDB: 6R7I. To avoid confusion, we have revised the manuscript text on page 5, lines 149-150: *“including CSN complex in CSN-CRL2 complex (PDB: 6R7I)”*.

• **Line 177 and Extended Data Fig. 2a:** the authors state that CSNAP engages with the C-terminal helical bundle of CSN1. However, upon examining the structure, it is evident that CSNAP does not directly interact with the helical bundle. Instead, it engages with the loop regions following the helical bundle from the CSN1 C-terminus. Please revise the text to reflect this distinction accurately.

To improve this imprecise wording, we have revised the manuscript text accordingly on page 6, lines 184-185: “CSNAP engages with the loop regions following the helical bundle from the CSN1 C-terminus, ...”

• **Lines 183-186:** please specify the residues within CSN3 and CSN8 that contribute to the hydrophobic and hydrogen bonding interactions with CSNAP. Including these details will help readers better understand the content and align the text more effectively with the corresponding figures.

To address this point and align the text with the corresponding figures, we have included specific residue numbers for CSN3 and CSN8 that contribute to CSNAP binding.

Page 7, Line 190-194: “...forming extensive hydrophobic contacts with CSN3 residues (L151, F155, Y186, Y187, M190, I219) (Extended Data Fig. 4b). Meanwhile, CSNAP’s acidic tail (E48, D52, D53, D54 and D55) occupies an electropositive groove formed between CSN3 (K153, K196, K296) and CSN8 (R54, K58), where multiple salt bridges and hydrogen bonds further stabilise the interface (Extended Data Fig. 4c).”

• **Line 220,** please state which structure(s) was used to compare with the pre-activated CSN-5H138A-N8-SCF.

We have clarified the comparative structural references used in our analysis.

Specifically, CSN^{apo} (PDB: 4D10) and the individual components of N⁸SCF were used for comparison. The N⁸CUL1/RBX1/SKP1 subcomplex was taken from PDB: 6TTU, while

SKP2/CKS1 was taken from PDB: 2ASS and superimposed onto the 6TTU structure using SKP1 as the alignment reference. We have revised the text as below:

Page 8-9, Line 251-253: "...compared with CSN^{apo} (PDB: 4D10) and N⁸SCF (a superimposed complex with N⁸CUL1/RBX1/SKP1 from PDB: 6TTU (Baek, Krist et al. 2020), SKP2/CKS1 were from PDB: 2ASS (Hao, Zheng et al. 2005)),"

• Line 295-310, from the pre-activated CSN-N8-SCF structure, the author hypothesised that disrupting the interface between RBX1-RING and CUL1-WHB would impair CSN activity, therefore they introduced mutations K726A/Q728A on CUL1-WHB and applied them for further SPR/in vitro deneddylation assays. First, isn't it more reasonable to test the interface between CSN2 and CSN4 with RBX1. This could provide additional insights into the interaction critical for CSN activity.

We agree with the reviewer that the interfaces between CSN2-RBX1 and CSN4-RBX1 are essential for CSN activity, as demonstrated by multiple studies involving CRL1, CRL2 and CRL4A complexes. Notably, Mosadeghi *et al.* (2016) highlighted the essential functional role of the RBX1^{RING} domain. Deletion of this domain (RBX1^{ΔRING}) led to a ~18,000-fold reduction in the k_{cat} for CSN-mediated cleavage of NEDD8 conjugated CUL1/RBX1^{ΔRING}, while the binding affinity was only modestly affected (Mosadeghi, Reichermeier *et al.* 2016). Similarly, Cavadini *et al.* (2016) showed that mutating residues in the RBX1-binding regions of CSN2 or CSN4, residues also identified in our structural characterisation of CSN-N⁸SCF complexes, impaired both CSN's binding to N⁸CRL4A^{DDB2} and catalytic efficiency (Cavadini, Fischer *et al.* 2016). Further support comes from Faull *et al.* (2019), who used cross-linking mass spectrometry (XL-MS) and hydrogen-deuterium exchange mass spectrometry (HDX-MS) to validate the importance of these interfaces in CSN-CRL2^{VHL} complexes (Faull, Lau *et al.* 2019).

These studies motivated our decision to instead investigate a previously uncharacterised interaction: the interface between the RBX1^{RING} domain and the CUL1^{WHB} domain. To our knowledge, our study provides the first structural visualisation of this interface at a resolution sufficient for detailed analysis. By characterising the

RBX1^{RING}-CUL1^{WHB} interface, we sought to determine whether this novel interface also contributes to the regulation of CSN-mediated deneddylation.

We have now revised the manuscript to acknowledge these previous studies concerning the functional importance of the interaction between CSN2 and CSN4 with RBX1 as follows:

Page 9-10, Line 281-286: *“Multiple studies involving CRL1, CRL2 and CRL4A complexes have previously demonstrated that CSN2 and CSN4 and the RBX1^{RING} domain are essential for substrate binding and efficient deneddylation (Cavadini, Fischer et al. 2016, Mosadeghi, Reichermeier et al. 2016, Faull, Lau et al. 2019). However, in this study, we provide high-resolution structural insights into these interactions, made possible by improved cryo-EM map quality.”*

• **Second, could the author examine the structure based on activated state and design the experiment.**

We have carefully compared the pre-activated and activated CSN^{5H138A_N8}SCF structures, focusing specifically on the interfaces between CSN2 and CSN4 with CUL1/RBX1. Structural comparisons, presented in Extended Data Fig. 9 (new), along with a calculated overall r.m.s.d of 0.797 Å, indicate that these interfaces remain largely unchanged between the two states, within the resolution limits of our data.

We have thus revised the text on page 10, line 301-302: *“We note that these interactions are conserved in the activated CSN^{5H138A_N8}SCF complex (Extended Data Fig. 9a-c).”*

As well as on page 11, line 333-335: *“This CUL1^{WHB}-RBX1^{RING} interaction network is also retained in the activated CSN^{5H138A_N8}SCF complex (Extended Data Fig. 9d).”*

Extended Data Fig. 9. Structural comparison of pre-activated and activated CSN^{5H138A-N8}SCF at the RBX1^{RING} domain

a, Superimposition of CSN2, CSN4 and the RBX1^{RING} domain from pre-activated and activated CSN^{5H138A-N8}SCF structures. **b**, Close-up view comparing the CSN2-RBX1^{RING} interface in pre-activated (grey) and activated (coloured) states. **c**, Comparison of the CSN4-RBX1^{RING} interface between pre-activated (grey) and activated (coloured) states. **d**, Comparison of the CUL1^{WHB}-RBX1^{RING} interface in pre-activated (grey) and activated (coloured) states.

- In line 329, the authors propose that residue E122 in CSN5 contributes to interactions with NEDD8. However, the observation that the CSN5 H138A/E122K mutant still retains binding to N8-CRL1 (albeit with reduced activity) raises questions about the functional role of E122, particularly given its position far from the Ins-1 loop and the catalytic site. Similarly, the basis for the reduced activity in the L157G and F161G mutants remains unclear. Could

the authors experimentally test whether the E122K, L157G, or F161G mutations affect stabilization or any other functional roles?

We agree that the ability of the CSN^{5H138A/E122K} and CSN^{5H138A/L157G/F161G} mutants to bind neddylated CUL1/RBX1, despite markedly reduced catalytic activity, raises important questions about the roles of E122, L157 and F161 in CSN function.

To address this, we revisited our SPR data and performed additional kinetic analysis (see Table 2 and Extended Data Fig. 5c and 14a, c). For both mutants, the association rate (k_{on}) with ^{N8}CUL1/RBX1 remained largely unchanged relative to CSN^{5H138A}, indicating that initial substrate binding is not significantly impaired. However, the dissociation rate (k_{off}) was significantly increased, ~6-fold for CSN^{5H138A/E122K} and ~15-fold for CSN^{5H138A/L157G/F161G}. These findings suggest that E122K, L157G and F161G are not directly involved in substrate recognition but instead contribute to stabilising the enzyme-substrate complex. Although E122 is spatially distant from the CSN5^{InS-1} loop and the active site, it appears to play a supportive role in complex retention. Similarly, L157 and F161 form a stabilising interface that secures NEDD8 during catalysis. The weakened complex stability observed in these mutants likely accounts for their reduced deneddylation activity. We have revised the manuscript to incorporate these results and interpretations.

CSN	^{N8} CUL1/RBX1	K _d (nM) steady-state	K _d (nM) kinetics	k _{on} (10 ⁶ M ⁻¹ s ⁻¹)	k _{off} (s ⁻¹)
5H138A	WT	10 ± 3	2.6 ± 0.04	0.37 ± 0.2	0.00097 ± 0.0004
5H138A E122K	WT	18 ± 3	8.1 ± 1	0.68 ± 0.02	0.0055 ± 0.0003
5H138A/L157G/F161G	WT	80 ± 10	20 ± 2	0.65 ± 0.2	0.015 ± 0.006

Page 12, lines 372-379: *To elucidate a functional role for CSN5^{E122}, we performed kinetic analysis of the SPR data (Table 2, and Extended Data Fig. 14a, 5c). While the association rate constant (k_{on}) to neddylated CUL1/RBX1 was similar to that of CSN^{5H138A}, the dissociation rate constant (k_{off}) was accelerated nearly 6-fold, indicating that CSN5^{E122} is not directly involved in substrate recognition, but instead contributes to stabilising the enzyme-substrate complex, possibly through an interaction with NEDD8^{H68}. The*

weakened complex stability observed in this mutant likely accounts for its reduced deneddylation activity.

Page 13, Lines 390-393: *The CSN^{5H138A/L157G/F161G} mutant showed an 8-fold decrease in binding to neddylated CUL1/RBX1, primarily due to a faster k_{off} . The mutant had little impact on binding to non-neddylated CUL1/RBX1 (Table 2-3 and Extended Data Fig. 12g-h).*

Page 13, Lines 395-400: *These findings demonstrate that CSN5's interaction with the I36 and I44 patches of NEDD8 plays a fundamental role in CSN-mediated deneddylation. First, residues CSN5^{L157} and CSN5^{F161}, which contact the I36 patch, do not seem to affect initial binding but instead stabilise the enzyme-substrate complex. Along with CSN5^{E122}, they form a stabilising interface that promotes substrate retention and efficient catalysis.*

• In line 457, the authors state that "CSN5 H138A/T105F reduced CSN activity by over 50%." However, it is unclear whether they used the CSN5 H138A/T105F double mutant or just the CSN5 T105F single mutant for the assay. Given that the CSN5 H138A mutation is known to abolish deneddylation activity, this clarification is essential. Additionally, even though the CSN5 H138A mutant reportedly loses activity, could the authors still use this mutant to perform the same deneddylation activity assay for consistency and comparison?

Thank you for pointing this out. We apologise for the confusion. In the deneddylation activity assay, we used the CSN^{5T105F} single mutant, in which the catalytic histidine (H138) remains intact. To clarify this, we have corrected the manuscript text from "CSN^{5H138/T105F}" to "CSN^{5T105F}" to accurately reflect that only a single point mutation (T105F) was introduced.

Regarding your suggestion to include the CSN^{5H138A} mutant as a control for comparison, we agree that such an assay might be informative. However, as the complete loss of deneddylation activity in CSN^{5H138A} has been well established and widely reported (Cope, Suh et al. 2002), we focused our assay efforts on variants that retain activity .

• In Extended Data Fig. 9, the results presented for the deneddylation assay are incomplete. All CSN mutations indicated in panel c should be supplemented with SDS-PAGE (4–12%) analysis, coupled with phosphate radioactive imaging, to provide a more comprehensive and detailed dataset, similar in panel a and b.

To provide a more complete and consistent presentation of the deneddylation assay data across all panels, we have now included the corresponding SDS-PAGE (4–12%) analyses and radioactive phosphate imaging for all CSN mutants (Extended Data Fig. 13a-f).

a De-neddylation activity of CSN^{5WT}
on His-PKA-N8CUL1/RBX1

b De-neddylation activity of CSN^{5WT}
on His-PKA-N8CUL1^{K726A/Q728A}/RBX1

c De-neddylation activity of CSN^{5E122K}
on His-PKA-N8CUL1/RBX1

d De-neddylation activity of CSN^{5M117R}
on His-PKA-N8CUL1/RBX1

e De-neddylation activity of CSN^{5L157G/F161G}
on His-PKA-N8CUL1/RBX1

f De-neddylation activity of CSN^{5T105F}
on His-PKA-N8CUL1/RBX1

Extended Data Fig. 13. In vitro De-neddylation activity assays.

a-f, SDS-PAGE (4–12%) analysis combined with phosphorimaging of in vitro de-neddylation assays. The assays compare CSN5 variants, CSN^{5WT} (**a**), CSN^{5E122K} (**c**), CSN^{5M117R} (**d**), CSN^{5L157G/F161G} (**e**), and CSN^{5T105F} (**f**), with $^{His-PKA-N8}CUL1/RBX1$ as substrate.

Panel **b** shows CSN^{5WT} acting on a mutant substrate $^{His-PKA-N8}CUL1^{K726A/Q728A}/RBX1$. **g**, Quantification of de-neddylation activity measured as the increasing ratio of free NEDD8 to total NEDD8 (free NEDD8 + CUL1-bound NEDD8) over time. The de-neddylation activity is indicated by the slope of the linear regression curve, with mutations in CSN or CUL1 distinguished by colour-coded data points. **h**, SDS-PAGE (4–12%) and Coomassie staining of in vitro de-neddylation assays comparing 1: $^{N8}CUL1/RBX1$; 2: $^{N8}CUL1^{R741E}/RBX1$; and 3: $^{N8}CUL1^{R717E}/RBX1$.

• For the cryo-EM analysis, please provide the models to map the Fourier Shell Correlation (FSC) for the individual structures. Including this data will enhance the transparency and rigour of the structural analysis.

To improve the transparency and rigour of our structural analysis, we have calculated model–map FSC curves with Phenix version 1.21.1 (Adams, Afonine et al. 2010) for all individual structures. These data have been included as additional figures (Extended Data Fig. 3 and 23), now provided below and in the revised manuscript.

Also, we have revised the main text as below:

Page 5, lines 140-142: *Fourier Shell Correlation (FSC) analysis confirmed good model-to-map agreement with minimal overfitting (Extended Data Fig. 3).*

Extended Data Fig. 3. Models-to-map Fourier Shell Correlation (FSC) analysis for the CSN^{5H138A}-N8 SCF complexes.

a, FSC curves for the pre-activated CSN^{5H138A}-N8 SCF complex. **b**, FSC curves for the activated CSN^{5H138A}-N8 SCF complex. For each structure, the FSC was calculated between the refined atomic model and the corresponding cryo-EM map with Phenix version 1.21.1 (Adams, Afonine et al. 2010). Both masked (orange) and unmasked (blue) correlations are shown. The resolution at the FSC = 0.5 criterion is indicated by the vertical dashed line.

Extended Data Fig. 23. Models-to-map Fourier Shell Correlation (FSC) analysis for the CSN^{E104A}-SCF dissociation states.

a, FSC curves for dissociation-state-1 CSN-SCF complex. **b**, FSC curves for dissociation-state-2 CSN-SCF complex. **c**, FSC curves for dissociation-state-3 CSN-SCF complex. **d**, FSC curves for dissociation-state-4 CSN-SCF complex. **e**, FSC curves for dissociated CSN^{apo} complex. For each structure, the FSC was calculated between the refined atomic model and the corresponding cryo-EM map using the *phenix.validation_cryoem* tool. Both masked (orange) and unmasked (blue) correlations are shown. The resolution at the FSC = 0.5 criterion is indicated by the vertical dashed line.

- **Have the authors tried local refinement on the N-terminal regions of CUL1, SKP1 and SKP2 to increase the quality of the maps?**

Yes, we performed local refinement to improve map quality for the N-terminal regions of CUL1, SKP1, and SKP2.

The data processing is shown in new Extended Data Fig. 2, local 3D classification followed by refinement of the pre-activated dataset (Class A) resulted in a map at 3.8 Å resolution with improved density in the N-terminal regions of these components (left panel of Extended Data Fig. 2a and b). A similar strategy was applied to the activated sub-dataset (Class B) (right panel), yielding a 3.5 Å map with enhanced density in the same regions (right panel of Extended Data Fig. 2a and b).

However, this approach required extensive particle binning: 75% of the particles were binned in the pre-activated dataset and 66% in the activated dataset. This led to a noticeable compromise in the local resolution of the CSN5–NEEDD8–CUL1^{WHB} interface (Extended Data Fig. 2c-d), which is central to the deneddylation mechanism.

Given this trade-off, we prioritised the higher-resolution global reconstructions for model building, even though the density for the N-terminal regions of CUL1, SKP1, and SKP2 was suboptimal in those maps. For completeness, we are happy to deposit the locally refined maps with improved N-terminal density alongside the final reconstructions to support further exploration by interested researchers.

Extended Data Fig. 2. Focused 3D classification of the substrate receptor region in the CSN^{5H138A-N8} SCF dataset.

a, Workflow of 3D local classification targeting the substrate receptor (SKP1–SKP2–CKS1) in both Class A and Class B sub-datasets. The red circle highlights the mask used for focused classification. **b**, Resolution estimates of the final maps in (a). **c-d**, Local resolution estimates of the map generated from Class A and Class B, respectively.

• **Could the authors summarize the discoveries presented in this manuscript alongside the previously reported findings on the CSN-neddylated CRL2 complex (PMID: 31444342)? Adding a comparative summary in the discussion section would help provide a more cohesive understanding of the CSN working mechanisms against diverse CRL.**

In the revised Discussion section, we now include a direct comparison in the discussion between the findings from this study and those reported in the 2019 *Nature Communications* paper on the CSN–neddylated CRL2 complex (PMID: 31444342) as well as other CSN-CRL structural research (Cavadini, Fischer et al. 2016, Mosadeghi, Reichermeier et al. 2016, Faull, Lau et al. 2019, Hu, Zhang et al. 2024). We trust that this comparison further helps contextualise the significance of our findings within the broader understanding of CSN–CRL regulation and have included this summary in the revised manuscript as requested.

Discussion (Page 23, lines 735-749): *Our results build upon and significantly expand prior mechanistic studies of CSN-mediated regulation of CRLs, encompassing CRL1 (Mosadeghi, Reichermeier et al. 2016), CRL2 (Faull, Lau et al. 2019), CRL3 (Hu, Zhang et al. 2024), and CRL4 (Cavadini, Fischer et al. 2016). A unifying theme across these studies is the conserved role of CSN2 and CSN4 in clamping both the CUL1^{CTD} and RBX1^{RING}, along with the release mechanism of the CSN5/CSN6 heterodimer from CSN4. In our work, these key interfaces are resolved at near-atomic resolution, providing new structural insight. Notably, our study structurally characterises the interface between RBX1^{RING} and the CUL1^{WHB} domain, an interaction essential for positioning NEDD8 in a catalytically competent configuration, and one that we have now experimentally validated. Faull et. al., previously proposed that NEDD8 triggers remodelling of the CSN5 active site (Faull, Lau et al. 2019). Our structural data support and extend this model, showing that although NEDD8 facilitates CSN5 activation indirectly, it is the interaction between CSN5 and the CUL1^{WHB} domain that directly drives remodelling of the active site, initiating a two-step mechanism.*

- **Please provide marker in Extended Data Fig1a and Fig17a.**

We have now added molecular weight markers to Extended Data Fig. 1a and Extended Data Fig. 22a (old 17a). We hope these additions help improve the clarity and interpretability of the results.

- **Given that the dimensions of the protein assemblies described in this paper are approximately 150 × 170 angstroms, please confirm whether the scale bars in Extended Data Fig. 1c and Extended Data Fig. 17 are accurate.**

We thank the reviewer for drawing our attention to this point. The 2D class averages shown in Extended Data Fig. 1c and Extended Data Fig. 17. were generated from images that had been two-fold binned (pixel size = 2.16 Å, not the unbinned 1.08 Å). Consequently, the original 50 Å scale bars were incorrect. We have replaced them with the correct 100 Å scale bars (Extended Data Fig. 1c and 22c). We apologise for the oversight.

- **Please indicate the corresponding figures between line 252-263. It is easier to follow the text and figures.**

We have now added references to the corresponding figures on page 10, lines 286–301 to improve clarity and help guide the reader through the results more effectively. These changes are reflected in the revised manuscript.

“Structurally, the RBX1^{RING} consists of two zinc-chelating loops (RBX1^{RING-loop1} and RBX1^{RING-loop2}), an α -helix (RBX1^{RING-helix3}) and a three-stranded β -sheet, alongside an insertion motif (RBX1^{RING-insertion}: residues 50–70) stabilised by a third zinc ion (Extended Data Fig. 8f). At the CSN2-RBX1^{RING} interface, a shallow groove in the RBX1^{RING} accommodates CSN2^{arm}. This groove, predominantly formed by the RBX1^{RING-helix3}, RBX1^{RING-loop1} and RBX1^{RING-loop2}, features residues RBX1^{A43}, RBX1^{I44}, RBX1^{F79}, RBX1^{W87}, RBX1^{P95} and RBX1^{L96}, which create a hydrophobic environment for CSN2^{I291} (Fig. 1e).

Additional stability comes from two salt bridges, CSN2^{E260}-RBX1^{R99} and CSN2^{D298}-RBX1^{H77}, along with a hydrogen bond between the backbone carbonyl group of CSN2^{G290} and side chain RBX1^{R46} (Fig. 1e). On the opposite side of the clamp, CSN4^{arm} links to RBX1 via RBX1^{RING-loop1} and the RBX1^{RING-insertion} motif, forming further stabilising interactions. The binding surface is predominantly hydrophobic, with residues RBX1^{V38}, RBX1^{V39}, RBX1^{I49} and RBX1^{M50} creating a pocket for CSN4^{L173} (Fig. 1f). Additionally, the imidazole ring of RBX1^{H48} engages in a hydrogen bond with the side chain of CSN4^{N169}, further strengthening the interface (Fig. 1f)."

- **Extended Data Fig.8n “Rbx1” should be “RBX1”; Fig 2J, “Csn5” should be “CSN5”; ED Fig2c, “csn8” should be “CSN8”.**

Thank you for pointing this out. We have corrected the capitalisation of all protein names in the corresponding figures to maintain consistency with standard nomenclature. The revised figures have been updated in the revised manuscript as new Extended Data Fig. 12j, Extended Data Fig.4c.

- **The figure legend for Extended Data Fig 7e does not match the label in the figure.**

We have corrected the figure legend to ensure consistency with the figure panel, as shown in the revised manuscript (new Extended Data 11e).

- **In Extended Data Fig. 8d and 8j, the reported affinities should be similar, as the M117R mutation was specifically designed to disrupt the CSN5-NEDD8 interaction. Since non-neddylated protein is used in this context, the mutation is not expected to influence the affinity. Please clarify or address this discrepancy.**

We agree that the CSN^{5M117R} mutation was originally designed to disrupt the CSN5–NEDD8 interface within the CSN-N⁸SCF complex and indeed resulted in a 12-fold reduction in binding affinity, as determined by our original, steady-state SPR data analysis. Interestingly, this mutation also led to a ~3-fold decrease in binding affinity to

non-neddylated CUL/RBX1 ($K_d = 760$ nM for CSN^{5H138A/M117R} vs. 260 nM for CSN^{5H138A}). While this reduction is relatively modest, we further investigated its impact by reanalysing existing SPR datasets for this mutant and performing additional kinetic measurements, now included in Extended Data Fig. 5i and 14b (new). Kinetic analysis revealed that CSN^{5H138A/M117R} had no significant effect on the dissociation rate (k_{off}) but showed a ~3-fold decrease in the association rate (k_{on}) with non-neddylated CUL1/RBX1, consistent with its reduced affinity (Table 2 and Extended Data Fig. 5i and 14b). A comparable reduction in k_{on} rate was also observed for experiments performed with neddylated CUL1/RBX1, suggesting that the M117R mutation may induce subtle conformational changes that influence initial complex formation, even in the absence of NEDD8. Notably, the most pronounced effect of CSN^{5H138A/M117R} was observed in the k_{off} when NEDD8 was present, supporting the importance of the CSN5-NEDD8 interface in stabilising the neddylated complex.

CSN	CUL1/RBX1	K_d (nM) steady-		k_{on} (10^6 M ⁻¹ s ⁻¹)	k_{off} (s ⁻¹)
		state	K_d (nM) kinetics		
5H138A	WT	260 ± 30	220 ± 90	0.74 ± 0.04	0.16 ± 0.06
5H138A/M117R	WT	760 ± 20	760 ± 20	0.19 ± 0.02	0.14 ± 0.02

CSN	N ⁸ CUL1/RBX1	K_d (nM) steady-		k_{on} (10^6 M ⁻¹ s ⁻¹)	k_{off} (s ⁻¹)
		state	K_d (nM) kinetics		
5H138A	WT	10 ± 3	2.6 ± 0.04	0.37 ± 0.2	0.0004
5H138A/M117R	WT	120 ± 9	80 ± 30	0.095 ± 0.02	0.0074 ± 0.003

We have now clarified this point in the revised text.

Page12-13, lines 383-387: *Kinetic SPR analysis revealed that the CSN^{5H138A/M117R} mutant exhibited an approximately ~30-fold reduction in binding affinity for neddylated CUL1/RBX1, resulting from a ~4-fold decrease in the k_{on} rate and an ~8-fold increase in*

the k_{off} rate, relative to CSN^{5H138A}. Interestingly, the same mutation also led to a relatively modest ~3-fold reduction in binding affinity for non-neddylated CUL1/RBX1, resulting from a ~3-fold slower association rate (Table 2-3, Extended Data Fig. 12e-f, Extended Data Fig. 14b, Extended Data Fig. 5g-i).

Page13, lines 400-403: *Second, although disrupting the CSN5-NEDD8 I44 interface through the CSN^{5M117R} mutation severely impaired binding and catalysis, the modest impact on binding to the non-neddylated substrate suggests that CSN^{5M117} may also influence complex formation independently of NEDD8.*

• **Line 343-344, while CSN5 E122 does not directly mediate NEDD8 binding, it likely plays a crucial role in stabilizing NEDD8 for efficient catalysis, possibly through interactions with NEDD8 H68. This sentence is confusing. Please revise.**

We have revised the sentence in the main text for clarity. The revised version reads:

Page 12, lines 372-379: *To elucidate a functional role for CSN5^{E122}, we performed kinetic analysis of the SPR data (Table 2, and Extended Data Fig. 14a, 5c). While the association rate constant (k_{on}) to neddylated CUL1/RBX1 was similar to that of CSN^{5H138A}, the dissociation rate constant (k_{off}) was accelerated nearly 6-fold, indicating that CSN5^{E122} is not directly involved in substrate recognition, but instead contributes to stabilising the enzyme-substrate complex, possibly through an interaction with NEDD8^{H68}. The weakened complex stability observed in this mutant likely accounts for its reduced deneddylation activity.*

Reviewer #2 (Remarks to the Author):

Reviewer #3 (Remarks to the Author):

Ding and colleagues present a comprehensive structural analysis of the COP9 signalosome (CSN) interaction with its substrate, the SCF ubiquitin ligase complex. Using cryo-electron microscopy, the authors have captured multiple states of this molecular machinery, including pre-activated, activated, and four distinct dissociation intermediates. Their findings provide unprecedented insight into the mechanisms of CSN-mediated deneddylation, including the first structural visualization of CSNAP, a previously uncharacterized component of the CSN complex. While the structural work is a real tour de force, I have several significant concerns about the data interpretation and experimental validation, but I would still recommend publication once they are addressed

We thank the reviewer for the thoughtful and constructive summary of our work. We truly appreciate these comments, which have been valuable in guiding our revisions and improving the clarity and completeness of the manuscript. All points have been carefully addressed in the detailed responses below.

Major Concerns:

1. The authors designate structures as "pre-activated," "activated," and "dissociation" states with insufficient experimental evidence. The assignment of these states to a sequential reaction coordinate is largely speculative and not supported by kinetic data. The authors provide no compelling evidence that these aren't simply different conformations of the complex that exist in equilibrium or, in some cases, potentially damaged particles. Without kinetic experiments or complementary biophysical approaches, the sequential nature of these states remains unproven, undermining the mechanistic model.

We agree that definitive validation for a stepwise reaction sequence would require additional, complementary biophysical approaches, including time-resolved cryo-EM or kinetic trapping. These approaches are beyond the scope of the present study but remain active goals for future investigation, as acknowledged in the original manuscript (see Discussion, page24 lines 749-764).

Our use of the terms "pre-activated", "activated" and "dissociation" is not meant to imply a fully resolved or experimentally validated reaction pathway. Rather, these designations are intended to provide a mechanistic framework based on the structural snapshots we obtained, interpreted within their defined biochemical contexts. Each structure was assembled under specific biochemical conditions designed to favour distinct functional states, allowing us to derive a plausible, hypothesis-driven model for understanding CSN-CRL regulation.

While we acknowledge that our interpretations are model based, we believe that the biochemical context in which these complexes were assembled, coupled with the structural features observed at high resolution, lends strong support to our proposed descriptions. In response to the reviewer's point 1, we have now revised the manuscript in the following ways:

Activated state:

To enable structural studies of the active CSN^{-N8}SCF complex, we reconstituted the 9-subunit CSN^{5H138A} enzyme with its neddylated SCF substrate. The H138A mutation in the CSN5 active site abolishes deneddylation activity (Cope, Suh et al. 2002) (page 4, lines 120-121) and is a well-established structural tool to trap the activated transition state of CSN by stably retaining NEDD8 over the time scale of an experiment (Enchev, Scott et al. 2012, Cavadini, Fischer et al. 2016, Mosadeghi, Reichermeier et al. 2016, Faull, Lau et al. 2019, Hu, Zhang et al. 2024). In the resulting CSN^{5H138A}-N8SCF complex, NEDD8 is positioned within the CSN5 active site, with its isopeptide bond located ~3 Å from the catalytic zinc-coordinating residues (lines 500-502, fig.2d, right panel). This geometry is consistent with a catalytically poised conformation, as observed in other MPN+ enzymes such as RPN11 and AMSH-LP (lines 486-491). The use of the catalytically impaired CSN^{5H138A} mutant supports the conclusion that our complex represents a trapped, activated intermediate (see Extended Data Fig. 19).

Pre-activated state:

The structure designated as pre-activated CSN^{5H138A}-N8SCF reveals distinct structural hallmarks of activation, including a clear remodelling of the CSN5 Ins-1 loop away from the catalytic site and a coordinated engagement of CUL1^{WHB} and conjugated NEDD8 with CSN5. These features suggest that the substrate is properly aligned in readiness for catalysis, consistent with a trajectory towards full enzymatic activation. While we do not claim this represents the sole intermediate along the activation pathway, we chose the term “pre-activated” as a general term to reflect its preparatory conformation (see Results, Fig. 2c-d left panel).

Dissociation states:

The four CSN-SCF dissociation intermediates were obtained from a complex assembled with the CSN^{5E104A} mutant, previously characterised by us (Mosadeghi, Reichermeier et al. 2016). This mutation is known to inhibit product release, resulting in a product-inhibited form of the enzyme. Specifically, CSN^{5E104A} binds deneddylated

CRL, the reaction product, with significantly higher affinity than wild-type CSN ($K_d = 26$ nM vs. 310 nM) and exhibits a markedly slower dissociation rate ($k_{off} = 0.13 \text{ s}^{-1}$ vs. 1.1 s^{-1} for wild type; see Figure 3C in Mosadeghi, Reichermeier *et al.*, 2016). To aid the reader, we have included additional details of this kinetic work within the text.

Page 18, lines 547-550: *Specifically, CSN^{5E104A} binds deneddylated CRL, the reaction product, with more than 10-fold higher affinity than wild-type CSN and exhibits an 8-fold reduction in the product dissociation rate (Mosadeghi, Reichermeier *et al.* 2016).*

These kinetic findings strongly support the interpretation that the resolved conformations in our cryo-EM dataset correspond to post-catalytic complexes that are stalled along the dissociation pathway. All four states were resolved at reasonably high resolution and display distinct structural differences with a trajectory toward complex disassembly. Based on the biochemical context of the CSN^{5E104A} mutant and the structural features of the maps, we believe this evidence provides a strong rationale for interpreting these as dissociation intermediates, but with appropriate caveats clearly stated. Therefore, we explicitly acknowledge in the revised manuscript that alternative interpretations are possible, namely that some of these states could reflect early assembly intermediates rather than post-catalytic ones. We have revised the Discussion accordingly to present a more balanced and transparent interpretation as below:

Discussion, page 24, lines 755–764: *While we favour the interpretation that these structures correspond to sequential dissociation intermediates, we explicitly acknowledge that some may additionally reflect early association states, given the inherent reversibility of protein-protein interactions. Nonetheless, the defined assembly conditions, high-resolution features, and specificity of the CSN-SCF interactions observed in the absence of NEDD8 support their classification as bona fide dissociation intermediates. Ultimately, complementary approaches such as time-resolved cryo-EM (Maeots, Lee *et al.* 2020, Maeots and Enchev 2022, Märt-Erik Mäeots and Rodriguez Molina 2025) will be essential to fully elucidate the temporal dynamics of the CSN–CRL regulatory cycle.*

We hope these clarifications address the reviewer's concerns and reinforce that while the sequence of states remains a model, it is grounded in a combination of structural, biochemical, and mechanistic reasoning.

2. Despite providing detailed structural information about CSNAP's integration into the CSN complex, the authors present no functional data whatsoever.

Critical questions remain entirely unaddressed:

Does CSNAP affect deneddylation activity? Does it alter binding kinetics?

Does it influence complex stability? A basic comparison of 8-subunit versus 9-subunit CSN would provide essential functional context for the structural observations.

These are indeed important points that have attracted the attention of the field and been the subject of previous work. Since the identification of CSNAP as the ninth subunit of CSN (Rozen, Fuzesi-Levi et al. 2015), its biochemical role has been systematically studied. *Füzesi-Levi et. al. (2020)*, in collaboration with our laboratory, conducted extensive *in vitro* and cellular analysis demonstrating that CSNAP does not impact CSN's deneddylation activity (Fuzesi-Levi, Fainer et al. 2020). Based on those findings, we did not pursue additional activity assays related to CSNAP in the current study.

However, CSNAP was previously reported to reduce the binding affinity of CSN for both neddylated and non-neddylated CUL1/RBX1 (Fuzesi-Levi, Fainer et al. 2020). To further examine the effect of CSNAP on substrate binding, we performed detailed SPR kinetic analysis comparing CSN^{5H138A} complexes with and without CSNAP (referred to as 9CSN^{5H138A} and 8CSN^{5H138A}, respectively) against both forms of CUL1/RBX1. The results are presented in Tables 2-3 and Extended Data Fig. 5a-h.

CSN	N ⁶ CUL1/RBX1	K _d (nM) steady-state	K _d (nM) kinetics	k _{on} (10 ⁶ M ⁻¹ s ⁻¹)	k _{off} (s ⁻¹)
8CSN5H138A	WT	6 ± 3	1.6 ± 0.3	1.0 ± 0.4	0.0016 ± 0.0005
9CSN5H138A	WT	10 ± 3	2.6 ± 0.04	0.37 ± 0.2	0.00097 ± 0.0004

CSN	CUL1/RBX1	K _d (nM) steady-state	K _d (nM) kinetics	k _{on} (10 ⁶ M ⁻¹ s ⁻¹)	k _{off} (s ⁻¹)
8CSN5H138A	WT	90 ± 8	77 ± 20	0.62 ± 0.3	0.044 ± 0.01
9CSN5H138A	WT	260 ± 30	220 ± 90	0.74 ± 0.04	0.16 ± 0.06

Our data indicate that CSNAP modulates the binding kinetics and stability of CSN-substrate interactions. Consistent with previous findings (Fuzesi-Levi, Fainer et al. 2020), incorporation of CSNAP into the CSN complex reduces binding affinity for both neddylated and non-neddylated CUL1/RBX1. For the neddylated substrate, 9CSN^{5H138A} exhibits a slightly slower association rate ($k_{on} = 0.37 \times 10^6 \text{ M}^{-1}\text{s}^{-1}$) compared to 8CSN^{5H138A} ($k_{on} = 1.0 \times 10^6 \text{ M}^{-1}\text{s}^{-1}$), while the dissociation rate remains largely unchanged. Conversely, for the non-neddylated substrate, the association rates are similar between the two complexes, but the dissociation rate is significantly higher for 9CSN^{5H138A} ($k_{off} = 0.16 \text{ s}^{-1}$) compared to 8CSN^{5H138A} ($k_{off} = 0.04 \text{ s}^{-1}$).

These kinetic findings suggest that CSNAP reduces overall affinity for CRL substrates by modulating complex stability, specifically, by decreasing the association rate for neddylated CRLs and increasing the dissociation rate for non-neddylated CRLs. Importantly, this decreased binding affinity does not compromise the structural integrity of the complex: all seven cryo-EM reconstructions of 9-subunit CSN reveal well-defined density for CSNAP, indicating stable incorporation in both neddylated and non-neddylated complexes. Furthermore, removal of CSNAP does not disrupt the integrity of the remaining eight CSN subunits, as evidenced by both cryo-EM analysis and biochemical purification profiles (Lingaraju, Bunker et al. 2014).

Notably, our ability to obtain seven near-atomic resolution cryo-EM maps may, in part, reflect CSNAP's stabilising effect on the overall rigidity of the CSN complex, offering a potential advantage over previous structural studies of CSN lacking CSNAP (Enchev,

Scott et al. 2012, Cavadini, Fischer et al. 2016, Mosadeghi, Reichermeier et al. 2016, Faull, Lau et al. 2019, Hu, Zhang et al. 2024).

Although CSNAP does not directly participate in catalysis, our data support its role as a regulatory subunit that fine-tunes the dynamics of CSN-substrate interactions. These findings have been incorporated into the revised manuscript and supporting figures as below.

Page 7, Lines 203-217: *To investigate the effect of CSNAP on binding, we performed surface plasmon resonance (SPR) kinetic measurements comparing CSN^{5H138A} complexes with and without CSNAP (referred to as 9CSN^{5H138A} and 8CSN^{5H138A}, respectively) with both neddylated and non-neddylated CUL1/RBX1 (Table 2 and Extended Data Fig. 5a-h). Consistent with prior findings (Fuzesi-Levi, Fainer et al. 2020), incorporation of CSNAP into the CSN complex reduces the binding affinity for both neddylated and non-neddylated CUL1/RBX1. For the neddylated substrate, 9CSN^{5H138A} exhibits a slower association rate ($k_{on} = 0.37 \times 10^6 \text{ M}^{-1}\text{s}^{-1}$) compared to 8CSN^{5H138A} ($k_{on} = 1.0 \times 10^6 \text{ M}^{-1}\text{s}^{-1}$), while the dissociation rate remains largely unchanged. Conversely, for the non-neddylated substrate, the association rates are similar between the two complexes, but the dissociation rate is significantly higher for 9CSN^{5H138A} ($k_{off} = 0.16 \text{ s}^{-1}$) compared to 8CSN^{5H138A} ($k_{off} = 0.04 \text{ s}^{-1}$). These kinetic findings suggest that CSNAP reduces overall affinity for CRL substrates by modulating complex stability, specifically, by decreasing the association rate for neddylated CRLs and increasing the dissociation rate for non-neddylated CRLs.*

Page 8, Lines 232-238: *Our complementary kinetics analysis indicates that CSNAP modulates CSN-substrate interaction dynamics. Specifically, incorporation of CSNAP into the CSN complex reduces the association rate with neddylated CUL1/RBX1 and increases the dissociation rate from non-neddylated CUL1/RBX1. These findings support a model in which CSN acts as a regulatory subunit that fine-tunes substrate binding and release without directly contributing to deneddylation activity.*

Extended Data Fig. 5. Comparative SPR analysis of CSN variants binding to neddylated and non-neddylated CUL1/RBX1

a, c, e, g, SPR sensorgrams showing binding of CSN^{5H138A} variants (with or without CSNAP) to immobilised StrepII^{2x}-tagged neddylated (**a, c**) or non-neddylated (**e, g**) CUL1/RBX1. Sensorgrams were analysed by global fitting using either a 1:1 Langmuir binding model (**a, c**) or a 1:1 binding model with drift correction (**e, g**), to extract

association (k_{on}) and dissociation (k_{off}) rate constants and calculate kinetic K_d values (black = experimental; red = fit). **b, d, f, h**, Corresponding steady-state binding curves derived from (**a, c, e, g**). Data were analysed by fitting a hyperbolic one-site binding model to determine steady-state K_d . For panels **b** and **f**, responses were normalised to the R_{max} value to account for variability in immobilisation levels across triplicates and for the graph adjusted to the median R_{max} . Raw responses were used in panels **d** and **h, i**, SPR sensorgrams showing binding of CSN^{5H138A/M117R} to immobilised *Streptococcus* CUL1/RBX1. Sensorgrams were analysed by global fitting using either a 1:1 binding model with drift correction, to extract association (k_{on}) and dissociation (k_{off}) rate constants and calculate kinetic K_d values (black = experimental; red = fit). All the data represent mean \pm SD from three independent experiments.

To investigate whether CSNAP influences the structure of the CSN complex, we compared our 9-subunit CSN^{apo} structure with the previously determined 8-subunit CSN crystal structure (PDB: 4D10) (Lingaraju, Bunker et al. 2014), as detailed in lines 783-802 and Extended Data Fig. 33 (formerly Extended Data Fig. 27). Subtle conformational differences are observed in regions near the CSN3–CSN8 interface, where CSNAP is located, as well as within the PCI Ring. These local shifts suggest that CSNAP may modulate structural dynamics in this region, although the resolution of both structures limits definitive interpretation (Extended Data Fig. 33b-c).

Together with our kinetic data, this structural comparison supports the conclusion that CSNAP does not impact the core assembly of the CSN complex. Rather, it acts as a regulatory component modulating substrate engagement. These insights add important functional context to the role of CSNAP within the holo-CSN complex.

3. The cryo-EM maps show clear signs of preferred orientation, as evident from the angular distribution plots. While DeepEMhancer has been used for post-processing, the potential artifacts introduced by preferred orientation are hidden to some extent But they should still be adequately acknowledged or addressed. A 3D FSC analysis would help quantify anisotropy in the maps.

Preferred orientations have been a well-documented challenge across all reported cryo-EM studies of CSN-CRLs. We acknowledge that also in this work the angular distribution plots indicate evidence of preferred particle orientation in our cryo-EM datasets. To rigorously assess the impact of this anisotropy on map quality, as suggested by the reviewer, we performed 3D Fourier Shell Correlation (3D FSC) analysis using the Remote 3DFSC Processing Server (<https://3dfsc.salk.edu/upload/>) (Tan, Baldwin et al. 2017).

The resulting plots are now included in Extended Data Figs. 1f, i, and 22g, j, m, p, s, respectively. This analysis confirms the presence of moderate anisotropy, but with overall sphericity values within acceptable limits. Importantly, no severe directional artefacts were observed that would compromise interpretation. Additionally, local resolution estimates and direct visual inspection of the density maps in critical regions, such as the CSN5 active site and the CSN-NEDD8 interface, demonstrate that these areas are well-resolved and suitable for reliable model building.

We have now acknowledged this limitation and included the 3D FSC results in the Extended Data Figs.1 and 22.

Extended Data Fig. 1. Reconstitution, cryo-electron microscopy, and single particle analysis of CSN^{5H138A}-N8SCF.

e, Euler angle distribution plots of pre-activated CSN^{5H138A}-N8SCF complex. **f**, Directional 3DFSC plots of pre-activated CSN^{5H138A}-N8SCF complex (Tan, Baldwin et al. 2017). **g**, Local resolution estimates of pre-activated CSN^{5H138A}-N8SCF complex. **h**, Euler angle distribution plots of activated CSN^{5H138A}-N8SCF complex. **i**, Directional 3DFSC plots of activated CSN^{5H138A}-N8SCF complex (Tan, Baldwin et al. 2017). **j**, Local resolution estimates of activated CSN^{5H138A}-N8SCF complex.

Extended Data Fig.22. Cryo-EM and single particle analysis on CSN^{5E104A}-SCF dissociation states.

e, Resolution estimates of maps resolved from the CSN^{E104A}-SCF dataset. **f**, Euler angle distribution plots for dissociation-state-1. **g**, Directional 3DFSC plots of dissociation-state-1. **h**, Local resolution estimates for dissociation-state-1. **i**, Euler angle distribution plots for dissociation-state-2. **j**, Directional 3DFSC plots of dissociation-state-2. **k**, Local resolution estimates of dissociation-state-2. **l**, Euler angle distribution plots for dissociation-state-3. **m**, Directional 3DFSC plots of dissociation-state-3. **n**, Local resolution estimates of dissociation-state-3. **o**, Euler angle distribution plots of dissociation-state-4. **p**, Directional 3DFSC plots of dissociation-state-4. **q**, Local resolution estimates of dissociation-state-4. **r**, Euler angle distribution plots of free CSN (CSN^{Apo}). **s**, Directional 3DFSC plots of free CSN (CSN^{Apo}). **t**, Local resolution estimates of free CSN (CSN^{Apo}). Resolutions for all maps in this figure were estimated using the gold-standard FSC 0.143 criterion.

Moreover, the local resolution estimation appears overly optimistic, particularly in peripherally located regions where resolution is likely significantly lower than 5 Å as indicated.

In response, we have recalculated the local resolution using an updated scale bar to ensure a more accurate estimation. The revised local resolution maps are now included in Extended Data Fig. 1g, j.

4. The methods section lacks critical details regarding image processing decisions. The implementation of cryoDRGN is inadequately described - specifically,

how particles were selected for different classes,

We have revised the Methods section to clarify the cryoDRGN workflow, as well as the figure legend for Extended Data Fig. 1d.

Page 36, Lines 995-997: The cryoDRGN-generated maps informed the design of local masks centred around the NEDD8 region. Focused classification was done with RELION-4.0.

Page 2-3 in SuppFigs, Lines 8-16: Extended Data Fig. 1c: Single particle analysis workflow for the CSN-^{N8}SCF dataset, resulting in 3D reconstructions of pre- and activated CSN^{5H138A-N8}SCF. CryoDRGN-based heterogeneity analysis is shown in the inset. Left: UMAP projection of the latent space. Middle: Representative cryoDRGN-generated structures at indicated UMAP coordinates. Right: Overlay of structures 3 and 8 highlighting distinct NEDD8 positions (circled in red). Densities from structures 3 and 8 were combined to create a mask for 3D local classification. 3D local classification was performed in RELION-4.0 (Kimanius, Dong et al. 2021) utilising a NEDD8-specific mask (generated from classification results from CryoDRGN (Zhong, Bepler et al. 2021)).

Please note that we did not use cryoDRGN for direct particle classification. Instead, we employed cryoDRGN to explore conformational heterogeneity within the dataset, particularly focusing on the NEDD8 moiety, which exhibited the greatest degree of structural flexibility. The cryoDRGN-generated maps informed the design of local masks in RELION-4.0 centred around the NEDD8 region.

These masks were subsequently applied in focused 3D classification using RELION. Particle subsets were then selected from the resulting classes based on improvements in local resolution within the CSN5/CSN6–NEDD8 region.

what criteria determined inclusion or exclusion,

The primary criterion guiding particle inclusion or exclusion was the local map quality of the NEDD8-CSN5 catalytic interface, which represents the core functional region relevant to our mechanistic studies. Our classification strategy focused on exploring conformational variability in the vicinity of NEDD8 and the CSN5 active site.

Therefore, particle selection was not based on global conformational differences of the CRL scaffold, but rather on the quality of local resolution at the CSN5/CSN6–NEDD8 interface.

And whether additional conformational states might exist in the dataset but were not pursued.

During our analysis, we did observe a small subset of particles that appeared to correspond to alternative conformational states. However, these were represented at low resolution, likely due to high intrinsic flexibility and limited particle numbers. Given their poor interpretability and relatively low representation within the dataset, we indeed did not pursue further structural analysis of these states. We have now clarified this rationale in the revised Methods section to ensure transparency in our analysis strategy.

What about the other clusters? These details are essential for evaluating the robustness of the structural classifications.

The additional clusters identified through cryoDRGN analysis primarily correspond to conformational variability within the CUL1–SKP1–SKP2 region, which is known to exhibit flexibility relative to the CSN core (as reported for instance in Mosadeghi et al. 2016). In addition, we observed a small number of clusters corresponding to minor particle subsets, which may represent alternative states. However, again, due to low particle numbers and insufficient resolution, we did not pursue further classification of these states.

Our focused analysis prioritised clusters that provided the highest resolution and most reliable density in the CSN5-NEDD8 region.

5. The modelling of the CSN5^{Ins-1} loop in the pre-activated state is problematic. This region shows fragmented density yet serves as the basis for significant mechanistic interpretations. Given the central importance of this loop to the authors' mechanistic arguments, a more rigorous approach to modelling this region is required, with explicit acknowledgement of the limitations imposed by map quality.

We fully acknowledge the limitations in modelling the CSN5^{Ins-1} loop due to its intrinsic flexibility and the corresponding weak density in our cryo-EM maps. In the original manuscript (lines 439-441), we stated, *“Due to the inherent flexibility of the CSN5^{Ins-1} region, we were unable to accurately model all side chains within the loop (Extended Data Fig. 13a).”*

In response to the reviewer’s suggestion for a more rigorous approach to modelling this region, we have now quantified the local resolution estimation of CSN5^{Ins-1} and calculated Q-scores for each residue in both pre-activated and activated CSN^{5H138A}-N⁸SCF structures. These data are presented in Extended Data Fig. 18 (previously Extended Data Fig. 13) and help to illustrate the resolution constraints that informed our model building. Side chains such as CSN5^{E101} and CSN5^{E104} were modelled conservatively as approximations, and this is explicitly stated in the figure legend.

Given the mechanistic significance of this loop, we complemented our structural studies with functional validation. Specifically, we performed a mutagenesis experiment targeting CUL1^{R741E}, designed based on our interpretation of the CSN5^{Ins-1} interaction interface observed in the pre-activated CSN^{5H138A}-N⁸SCF complex. As shown in Extended Data Fig. 13d (new), the CUL1^{R741E} mutation led to a marked reduction in deneddylation activity compared with wild-type N⁸CUL1/RBX1, supporting the functional relevance of our proposed model despite local resolution limitations.

Extended Data Fig. 18. Structural versatility of CSN5^{Ins-1}.

a, Local resolution estimation of CSN5^{Ins-1} and CUL1^{R741}, CUL1^{R717} in pre-activated CSN5^{H138A}-N8 SCF. **b**, Cryo-EM density and molecular model of CSN5^{Ins-1}, as well as CUL1^{R741} and CUL1^{R717} in pre-activated CSN5^{H138A}-N8 SCF. The side chains of CSN5^{E101} and CSN5^{E104} are modelled as approximations due to density limitations. For validation, the Q-score (Pintilie, Zhang et al. 2020) is calculated for the backbone and quoted under each residue. **c**, Local resolution estimation of CSN5^{Ins-1} in activated CSN5^{H138A}-N8 SCF. **d**, Cryo-EM density and molecular model of CSN5^{Ins-1} in activated CSN5^{H138A}-N8 SCF. **e**, Structural comparison of CSN5^{Ins-1} from pre-activated and activated CSN5^{H138A}-N8 SCF, alongside the isolated crystal structure of CSN5 (PDB: 4F70), demonstrating the conformational adaptability of this loop.

Adams, P. D., P. V. Afonine, G. Bunkoczi, V. B. Chen, I. W. Davis, N. Echols, J. J. Headd, L. W. Hung, G. J. Kapral, R. W. Grosse-Kunstleve, A. J. McCoy, N. W. Moriarty, R. Oeffner, R. J. Read, D. C. Richardson, J. S. Richardson, T. C. Terwilliger and P. H. Zwart (2010). "PHENIX: a comprehensive Python-based system for macromolecular structure solution." *Acta Crystallogr D Biol Crystallogr* **66**(Pt 2): 213-221.

Baek, K., D. T. Krist, J. R. Prabu, S. Hill, M. Klugel, L. M. Neumaier, S. von Gronau, G. Kleiger and B. A. Schulman (2020). "NEDD8 nucleates a multivalent cullin-RING-UBE2D ubiquitin ligation assembly." *Nature* **578**(7795): 461-466.

Cavadini, S., E. S. Fischer, R. D. Bunker, A. Potenza, G. M. Lingaraju, K. N. Goldie, W. I. Mohamed, M. Faty, G. Petzold, R. E. Beckwith, R. B. Tichkule, U. Hassiepen, W. Abdulrahman, R. S. Pantelic, S. Matsumoto, K. Sugasawa, H. Stahlberg and N. H. Thoma (2016). "Cullin-RING ubiquitin E3 ligase regulation by the COP9 signalosome." *Nature* **531**(7596): 598-603.

Cope, G. A., G. S. Suh, L. Aravind, S. E. Schwarz, S. L. Zipursky, E. V. Koonin and R. J. Deshaies (2002). "Role of predicted metalloprotease motif of Jab1/Csn5 in cleavage of Nedd8 from Cul1." *Science* **298**(5593): 608-611.

Enchev, R. I., D. C. Scott, P. C. da Fonseca, A. Schreiber, J. K. Monda, B. A. Schulman, M. Peter and E. P. Morris (2012). "Structural basis for a reciprocal regulation between SCF and CSN." *Cell Rep* **2**(3): 616-627.

Faull, S. V., A. M. C. Lau, C. Martens, Z. Ahdash, K. Hansen, H. Yebenes, C. Schmidt, F. Beuron, N. B. Cronin, E. P. Morris and A. Politis (2019). "Structural basis of Cullin 2 RING E3 ligase regulation by the COP9 signalosome." *Nat Commun* **10**(1): 3814.

Fuzesi-Levi, M. G., I. Fainer, R. Ivanov Enchev, G. Ben-Nissan, Y. Levin, M. Kupervaser, G. Friedlander, T. M. Salame, R. Nevo, M. Peter and M. Sharon (2020). "CSNAP, the smallest CSN subunit, modulates proteostasis through cullin-RING ubiquitin ligases." *Cell Death Differ* **27**(3): 984-998.

Hao, B., N. Zheng, B. A. Schulman, G. Wu, J. J. Miller, M. Pagano and N. P. Pavletich (2005). "Structural basis of the Cks1-dependent recognition of p27(Kip1) by the SCF(Skp2) ubiquitin ligase." *Mol Cell* **20**(1): 9-19.

Hu, Y., Z. Zhang, Q. Mao, X. Zhang, A. Hao, Y. Xun, Y. Wang, L. Han, W. Zhan, Q. Liu, Y. Yin, C. Peng, E. M. Y. Moresco, Z. Chen, B. Beutler and L. Sun (2024). "Dynamic molecular architecture and substrate recruitment of cullin3-RING E3 ligase CRL3(KBTBD2)." *Nat Struct Mol Biol* **31**(2): 336-350.

Kimanius, D., L. Dong, G. Sharov, T. Nakane and S. H. W. Scheres (2021). "New tools for automated cryo-EM single-particle analysis in RELION-4.0." *Biochem J* **478**(24): 4169-4185.

Lingaraju, G. M., R. D. Bunker, S. Cavadini, D. Hess, U. Hassiepen, M. Renatus, E. S. Fischer and N. H. Thoma (2014). "Crystal structure of the human COP9 signalosome." *Nature* **512**(7513): 161-165.

Maeots, M. E. and R. I. Enchev (2022). "Structural dynamics: review of time-resolved cryo-EM." *Acta Crystallogr D Struct Biol* **78**(Pt 8): 927-935.

Maeots, M. E., B. Lee, A. Nans, S. G. Jeong, M. M. N. Esfahani, S. Ding, D. J. Smith, C. S. Lee, S. S. Lee, M. Peter and R. I. Enchev (2020). "Modular microfluidics enables kinetic insight from time-resolved cryo-EM." *Nat Commun* **11**(1): 3465.

Märt-Erik Mäeots, S. T., Mohammad M. N. Esfahani, Juan B. and J. A. C. Rodriguez Molina, Aran Amin, Albane Imbert, Radoslav I. Enchev (2025). "Chronobot: Deep learning guided time-resolved cryo-EM

captures molecular choreography of RecA in homology search." *bioRxiv preprint*.

Mosadeghi, R., K. M. Reichermeier, M. Winkler, A. Schreiber, J. M. Reitsma, Y. Zhang, F. Stengel, J. Cao, M. Kim, M. J. Sweredoski, S. Hess, A. Leitner, R. Aebersold, M. Peter, R. J. Deshaies and R. I. Enchev (2016). "Structural and kinetic analysis of the COP9-Signalosome activation and the cullin-RING ubiquitin ligase deneddylation cycle." *Elife* **5**.

Pintilie, G., K. Zhang, Z. Su, S. Li, M. F. Schmid and W. Chiu (2020). "Measurement of atom resolvability in cryo-EM maps with Q-scores." *Nat Methods* **17**(3): 328-334.

Rozen, S., M. G. Fuzesi-Levi, G. Ben-Nissan, L. Mizrachi, A. Gabashvili, Y. Levin, S. Bendor, M. Eisenstein and M. Sharon (2015). "CSNAP Is a Stoichiometric Subunit of the COP9 Signalosome." *Cell Rep* **13**(3): 585-598.

Tan, Y. Z., P. R. Baldwin, J. H. Davis, J. R. Williamson, C. S. Potter, B. Carragher and D. Lyumkis (2017). "Addressing preferred specimen orientation in single-particle cryo-EM through tilting." *Nat Methods* **14**(8): 793-796.

Zhong, E. D., T. Bepler, B. Berger and J. H. Davis (2021). "CryoDRGN: reconstruction of heterogeneous cryo-EM structures using neural networks." *Nat Methods* **18**(2): 176-185.

Reviewer #1 (Remarks to the Author):

The CSN signalosome plays an important role in regulating both the neddylation and deneddylation states of Cullin-RING ligases (CRL). Enchev and colleagues reconstituted and applied cryo-EM analysis on CSN-N8SCF complex, revealing distinct functional states and capturing critical intermediates involved in the deneddylation cycle. Their work highlights the conformational rearrangements involving the CSN5 Ins-1 loop, the RBX1 RING domain and the neddylated Cullin WHB domain in achieving catalytic activation. They further detailed the sequential release of the CSN from the deneddylated product. Additionally, they also provided the structural positioning of CSNAP, which integrates into a groove formed by CSN3 and CSN8. Overall, the structures are presented with clarity, and the work provides new insights into this biological process.

We thank the reviewer as well as the early career co-reviewer from their lab for the thoughtful and constructive summary of our work, and for the overall positive and encouraging feedback. We greatly appreciate your comments, which have helped us further refine and strengthen the manuscript. We have carefully addressed all points in the detailed responses below.

However, the following concerns should be addressed before acceptance for publication.

- In Line 144, the PDB identifier 6R7I corresponds to the structure of the CSN bound to CRL2, not the apo CSN. Please clarify this.**

Thank you for pointing this out. We apologise for the oversight. In our analysis, we only used the CSN model from the structure deposited as PDB: 6R7I. To avoid confusion, we have revised the manuscript text on page 5, lines 149-150: *“including CSN complex in CSN-CRL2 complex (PDB: 6R7I)”*.

• **Line 177 and Extended Data Fig. 2a:** the authors state that CSNAP engages with the C-terminal helical bundle of CSN1. However, upon examining the structure, it is evident that CSNAP does not directly interact with the helical bundle. Instead, it engages with the loop regions following the helical bundle from the CSN1 C-terminus. Please revise the text to reflect this distinction accurately.

To improve this imprecise wording, we have revised the manuscript text accordingly on page 6, lines 184-185: “*CSNAP engages with the loop regions following the helical bundle from the CSN1 C-terminus, ...*”

• **Lines 183-186:** please specify the residues within CSN3 and CSN8 that contribute to the hydrophobic and hydrogen bonding interactions with CSNAP. Including these details will help readers better understand the content and align the text more effectively with the corresponding figures.

To address this point and align the text with the corresponding figures, we have included specific residue numbers for CSN3 and CSN8 that contribute to CSNAP binding.

Page 7, Line 190-194: “*...forming extensive hydrophobic contacts with CSN3 residues (L151, F155, Y186, Y187, M190, I219) (Extended Data Fig. 4b). Meanwhile, CSNAP’s acidic tail (E48, D52, D53, D54 and D55) occupies an electropositive groove formed between CSN3 (K153, K196, K296) and CSN8 (R54, K58), where multiple salt bridges and hydrogen bonds further stabilise the interface (Extended Data Fig. 4c).*”

• **Line 220,** please state which structure(s) was used to compare with the pre-activated CSN-5H138A-N8-SCF.

We have clarified the comparative structural references used in our analysis. Specifically, CSN^{apo} (PDB: 4D10) and the individual components of ^{N8}SCF were used for comparison. The ^{N8}CUL1/RBX1/SKP1 subcomplex was taken from PDB:

6TTU, while SKP2/CKS1 was taken from PDB: 2ASS and superimposed onto the 6TTU structure using SKP1 as the alignment reference. We have revised the text as below:

Page 8-9, Line 251-253: "...compared with CSN^{apo} (PDB: 4D10) and N⁸SCF (a superimposed complex with N⁸CUL1/RBX1/SKP1 from PDB: 6TTU (Baek, Krist et al. 2020), SKP2/CKS1 were from PDB: 2ASS (Hao, Zheng et al. 2005)),"

• **Line 295-310, from the pre-activated CSN-N8-SCF structure, the author hypothesised that disrupting the interface between RBX1-RING and CUL1-WHB would impair CSN activity, therefore they introduced mutations K726A/Q728A on CUL1-WHB and applied them for further SPR/in vitro deneddylation assays. First, isn't it more reasonable to test the interface between CSN2 and CSN4 with RBX1. This could provide additional insights into the interaction critical for CSN activity.**

We agree with the reviewer that the interfaces between CSN2-RBX1 and CSN4-RBX1 are essential for CSN activity, as demonstrated by multiple studies involving CRL1, CRL2 and CRL4A complexes. Notably, Mosadeghi *et al.* (2016) highlighted the essential functional role of the RBX1^{RING} domain. Deletion of this domain (RBX1^{ΔRING}) led to a ~18,000-fold reduction in the k_{cat} for CSN-mediated cleavage of NEDD8 conjugated CUL1/RBX1^{ΔRING}, while the binding affinity was only modestly affected (Mosadeghi, Reichermeier et al. 2016). Similarly, Cavadini *et al.* (2016) showed that mutating residues in the RBX1-binding regions of CSN2 or CSN4, residues also identified in our structural characterisation of CSN-N⁸SCF complexes, impaired both CSN's binding to N⁸CRL4A^{DDB2} and catalytic efficiency (Cavadini, Fischer et al. 2016). Further support comes from Faull *et al.* (2019), who used cross-linking mass spectrometry (XL-MS) and hydrogen-deuterium exchange mass spectrometry (HDX-MS) to validate the importance of these interfaces in CSN-CRL2^{VHL} complexes (Faull, Lau et al. 2019).

These studies motivated our decision to instead investigate a previously uncharacterised interaction: the interface between the RBX1^{RING} domain and the CUL1^{WHB} domain. To our knowledge, our study provides the first structural

visualisation of this interface at a resolution sufficient for detailed analysis. By characterising the RBX1^{RING}-CUL1^{WHB} interface, we sought to determine whether this novel interface also contributes to the regulation of CSN-mediated deneddylation.

We have now revised the manuscript to acknowledge these previous studies concerning the functional importance of the interaction between CSN2 and CSN4 with RBX1 as follows:

Page 9-10, Line 281-286: *“Multiple studies involving CRL1, CRL2 and CRL4A complexes have previously demonstrated that CSN2 and CSN4 and the RBX1^{RING} domain are essential for substrate binding and efficient deneddylation (Cavadini, Fischer et al. 2016, Mosadeghi, Reichermeier et al. 2016, Faull, Lau et al. 2019). However, in this study, we provide high-resolution structural insights into these interactions, made possible by improved cryo-EM map quality.”*

• **Second, could the author examine the structure based on activated state and design the experiment.**

We have carefully compared the pre-activated and activated CSN^{5H138A}-N⁸SCF structures, focusing specifically on the interfaces between CSN2 and CSN4 with CUL1/RBX1. Structural comparisons, presented in Extended Data Fig. 9 (new), along with a calculated overall r.m.s.d of 0.797 Å, indicate that these interfaces remain largely unchanged between the two states, within the resolution limits of our data.

We have thus revised the text on page 10, line 301-302: *“We note that these interactions are conserved in the activated CSN^{5H138A}-N⁸SCF complex (Extended Data Fig. 9a-c).”*

As well as on page 11, line 333-335: *“This CUL1^{WHB}-RBX1^{RING} interaction network is also retained in the activated CSN^{5H138A}-N⁸SCF complex (Extended Data Fig. 9d).”*

Extended Data Fig. 9. Structural comparison of pre-activated and activated CSN^{5H138A_N8} SCF at the RBX1^{RING} domain

a, Superimposition of CSN2, CSN4 and the RBX1^{RING} domain from pre-activated and activated CSN^{5H138A_N8} SCF structures. **b**, Close-up view comparing the CSN2-RBX1^{RING} interface in pre-activated (grey) and activated (coloured) states. **c**, Comparison of the CSN4-RBX1^{RING} interface between pre-activated (grey) and activated (coloured) states. **d**, Comparison of the CUL1^{WHB}-RBX1^{RING} interface in pre-activated (grey) and activated (coloured) states.

- In line 329, the authors propose that residue E122 in CSN5 contributes to interactions with NEDD8. However, the observation that the CSN5 H138A/E122K mutant still retains binding to N8-CRL1 (albeit with reduced activity) raises questions about the functional role of E122, particularly given its position far from the Ins-1 loop and the catalytic site. Similarly, the basis for the reduced activity in the L157G and F161G mutants remains unclear. Could

the authors experimentally test whether the E122K, L157G, or F161G mutations affect stabilization or any other functional roles?

We agree that the ability of the CSN^{5H138A/E122K} and CSN^{5H138A/L157G/F161G} mutants to bind neddylated CUL1/RBX1, despite markedly reduced catalytic activity, raises important questions about the roles of E122, L157 and F161 in CSN function.

To address this, we revisited our SPR data and performed additional kinetic analysis (see Table 2 and Extended Data Fig. 5c and 14a, c). For both mutants, the association rate (k_{on}) with ^{N8}CUL1/RBX1 remained largely unchanged relative to CSN^{5H138A}, indicating that initial substrate binding is not significantly impaired. However, the dissociation rate (k_{off}) was significantly increased, ~6-fold for CSN^{5H138A/E122K} and ~15-fold for CSN^{5H138A/L157G/F161G}. These findings suggest that E122K, L157G and F161G are not directly involved in substrate recognition but instead contribute to stabilising the enzyme-substrate complex. Although E122 is spatially distant from the CSN5^{Ins-1} loop and the active site, it appears to play a supportive role in complex retention. Similarly, L157 and F161 form a stabilising interface that secures NEDD8 during catalysis. The weakened complex stability observed in these mutants likely accounts for their reduced deneddylation activity. We have revised the manuscript to incorporate these results and interpretations.

CSN	^{N8} CUL1/RBX1	K_d (nM) steady-state	K_d (nM) kinetics	k_{on} ($10^6 M^{-1}s^{-1}$)	k_{off} (s^{-1})
5H138A	WT	10 ± 3	2.6 ± 0.04	0.37 ± 0.2	0.00097 ± 0.0004
5H138A E122K	WT	18 ± 3	8.1 ± 1	0.68 ± 0.02	0.0055 ± 0.0003
5H138A/L157G/F161G	WT	80 ± 10	20 ± 2	0.65 ± 0.2	0.015 ± 0.006

Page 12, lines 372-379: *To elucidate a functional role for CSN5^{E122}, we performed kinetic analysis of the SPR data (Table 2, and Extended Data Fig. 14a, 5c). While the association rate constant (k_{on}) to neddylated CUL1/RBX1 was similar to that of CSN^{5H138A}, the dissociation rate constant (k_{off}) was accelerated nearly 6-fold, indicating that CSN5^{E122} is not directly involved in substrate recognition, but instead contributes to stabilising the enzyme-substrate complex, possibly through an interaction with NEDD8^{H68}. The weakened complex stability observed in this mutant likely accounts for its reduced deneddylation activity.*

Page 13, Lines 390-393: *The CSN^{5H138A/L157G/F161G} mutant showed an 8-fold decrease in binding to neddylated CUL1/RBX1, primarily due to a faster k_{off} . The mutant had little impact on binding to non-neddylated CUL1/RBX1 (Table 2-3 and Extended Data Fig. 12g-h).*

Page 13, Lines 395-400: *These findings demonstrate that CSN5's interaction with the I36 and I44 patches of NEDD8 plays a fundamental role in CSN-mediated deneddylation. First, residues CSN5^{L157} and CSN5^{F161}, which contact the I36 patch, do not seem to affect initial binding but instead stabilise the enzyme-substrate complex. Along with CSN5^{E122}, they form a stabilising interface that promotes substrate retention and efficient catalysis.*

• In line 457, the authors state that "CSN5 H138A/T105F reduced CSN activity by over 50%." However, it is unclear whether they used the CSN5 H138A/T105F double mutant or just the CSN5 T105F single mutant for the assay. Given that the CSN5 H138A mutation is known to abolish deneddylation activity, this clarification is essential. Additionally, even though the CSN5 H138A mutant reportedly loses activity, could the authors still use this mutant to perform the same deneddylation activity assay for consistency and comparison?

Thank you for pointing this out. We apologise for the confusion. In the deneddylation activity assay, we used the CSN^{5T105F} single mutant, in which the catalytic histidine (H138) remains intact. To clarify this, we have corrected the manuscript text from "CSN^{5H138/T105F}" to "CSN^{5T105F}" to accurately reflect that only a single point mutation (T105F) was introduced.

Regarding your suggestion to include the CSN^{5H138A} mutant as a control for comparison, we agree that such an assay might be informative. However, as the complete loss of deneddylation activity in CSN^{5H138A} has been well established and widely reported (Cope, Suh et al. 2002), we focused our assay efforts on variants that retain activity .

• In Extended Data Fig. 9, the results presented for the deneddylation assay are incomplete. All CSN mutations indicated in panel c should be supplemented

with SDS-PAGE (4–12%) analysis, coupled with phosphate radioactive imaging, to provide a more comprehensive and detailed dataset, similar in panel a and b.

To provide a more complete and consistent presentation of the deneddylation assay data across all panels, we have now included the corresponding SDS-PAGE (4–12%) analyses and radioactive phosphate imaging for all CSN mutants (Extended Data Fig. 13a-f).

Extended Data Fig.13. In vitro De-neddylation activity assays.

a-f, SDS-PAGE (4–12%) analysis combined with phosphorimaging of in vitro de-neddylation assays. The assays compare CSN5 variants, CSN^{5WT} (**a**), CSN^{5E122K} (**c**), CSN^{5M117R} (**d**), CSN^{5L157G/F161G} (**e**), and CSN^{5T105F} (**f**), with His-PKA-N⁸CUL1/RBX1 as substrate. Panel **b** shows CSN^{5WT} acting on a mutant substrate His-PKA-N⁸CUL1K^{726A/Q728A}/RBX1. **g**, Quantification of de-neddylation activity measured as the increasing ratio of free NEDD8 to total NEDD8 (free NEDD8 + CUL1-bound NEDD8) over time. The de-neddylation activity is indicated by the slope of the linear regression curve, with mutations in CSN or CUL1 distinguished by colour-coded data points. **h**, SDS-PAGE (4-12%) and Coomassie staining of in vitro de-neddylation assays comparing 1: N⁸CUL1/RBX1; 2: N⁸CUL1^{R741E}/RBX1; and 3: N⁸CUL1^{R717E}/RBX1.

• For the cryo-EM analysis, please provide the models to map the Fourier Shell Correlation (FSC) for the individual structures. Including this data will enhance the transparency and rigour of the structural analysis.

To improve the transparency and rigour of our structural analysis, we have calculated model–map FSC curves with Phenix version 1.21.1 (Adams, Afonine et al. 2010) for all individual structures. These data have been included as additional figures (Extended Data Fig. 3 and 23), now provided below and in the revised manuscript.

Also, we have revised the main text as below:

Page 5, lines 140-142: *Fourier Shell Correlation (FSC) analysis confirmed good model-to-map agreement with minimal overfitting (Extended Data Fig. 3).*

Extended Data Fig. 3. Models-to-map Fourier Shell Correlation (FSC) analysis for the CSN^{5H138A}-N8SCF complexes.

a, FSC curves for the pre-activated CSN^{5H138A}-N8SCF complex. **b**, FSC curves for the activated CSN^{5H138A}-N8SCF complex. For each structure, the FSC was calculated between the refined atomic model and the corresponding cryo-EM map with Phenix version 1.21.1 (Adams, Afonine et al. 2010). Both masked (orange) and unmasked (blue) correlations are shown. The resolution at the FSC = 0.5 criterion is indicated by the vertical dashed line.

Extended Data Fig. 23. Models-to-map Fourier Shell Correlation (FSC) analysis for the CSN^{E104A}-SCF dissociation states.

a, FSC curves for dissociation-state-1 CSN-SCF complex. **b**, FSC curves for dissociation-state-2 CSN-SCF complex. **c**, FSC curves for dissociation-state-3 CSN-SCF complex. **d**, FSC curves for dissociation-state-4 CSN-SCF complex. **e**, FSC curves for dissociated CSN^PO complex. For each structure, the FSC was calculated between the refined atomic model and the corresponding cryo-EM map using the *phenix.validation_cryoem* tool. Both masked (orange) and unmasked (blue) correlations are shown. The resolution at the FSC = 0.5 criterion is indicated by the vertical dashed line.

- **Have the authors tried local refinement on the N-terminal regions of CUL1, SKP1 and SKP2 to increase the quality of the maps?**

Yes, we performed local refinement to improve map quality for the N-terminal regions of CUL1, SKP1, and SKP2.

The data processing is shown in new Extended Data Fig. 2, local 3D classification followed by refinement of the pre-activated dataset (Class A) resulted in a map at 3.8 Å resolution with improved density in the N-terminal regions of these components (left panel of Extended Data Fig. 2a and b). A similar strategy was applied to the activated sub-dataset (Class B) (right panel), yielding a 3.5 Å map with enhanced density in the same regions (right panel of Extended Data Fig. 2a and b).

However, this approach required extensive particle binning: 75% of the particles were binned in the pre-activated dataset and 66% in the activated dataset. This led to a noticeable compromise in the local resolution of the CSN5–NEEDD8–CUL1^{WHB} interface (Extended Data Fig. 2c-d), which is central to the deneddylation mechanism.

Given this trade-off, we prioritised the higher-resolution global reconstructions for model building, even though the density for the N-terminal regions of CUL1, SKP1, and SKP2 was suboptimal in those maps. For completeness, we are happy to deposit the locally refined maps with improved N-terminal density alongside the final reconstructions to support further exploration by interested researchers.

Extended Data Fig. 2. Focused 3D classification of the substrate receptor region in the CSN^{5H138A}_N8SCF dataset.

a, Workflow of 3D local classification targeting the substrate receptor (SKP1–SKP2–CKS1) in both Class A and Class B sub-datasets. The red circle highlights the mask used for focused classification. **b**, Resolution estimates of the final maps in (a). **c-d**, Local resolution estimates of the map generated from Class A and Class B, respectively.

- **Could the authors summarize the discoveries presented in this manuscript alongside the previously reported findings on the CSN-neddylated CRL2 complex (PMID: 31444342)? Adding a comparative summary in the discussion section would help provide a more cohesive understanding of the CSN working mechanisms against diverse CRL.**

In the revised Discussion section, we now include a direct comparison in the discussion between the findings from this study and those reported in the 2019 *Nature Communications* paper on the CSN–neddylated CRL2 complex (PMID: 31444342) as well as other CSN-CRL structural research (Cavadini, Fischer et al. 2016, Mosadeghi, Reichermeier et al. 2016, Faull, Lau et al. 2019, Hu, Zhang et al. 2024). We trust that this comparison further helps contextualise the significance of our findings within the broader understanding of CSN–CRL regulation and have included this summary in the revised manuscript as requested.

Discussion (Page 23, lines 735-749): *Our results build upon and significantly expand prior mechanistic studies of CSN-mediated regulation of CRLs, encompassing CRL1 (Mosadeghi, Reichermeier et al. 2016), CRL2 (Faull, Lau et al. 2019), CRL3 (Hu, Zhang et al. 2024), and CRL4 (Cavadini, Fischer et al. 2016). A unifying theme across these studies is the conserved role of CSN2 and CSN4 in clamping both the CUL1^{CTD} and RBX1^{RING}, along with the release mechanism of the CSN5/CSN6 heterodimer from CSN4. In our work, these key interfaces are resolved at near-atomic resolution, providing new structural insight. Notably, our study structurally characterises the interface between RBX1^{RING} and the CUL1^{WHB} domain, an interaction essential for positioning NEDD8 in a catalytically competent configuration, and one that we have now experimentally validated. Faull et. al., previously proposed that NEDD8 triggers remodelling of the CSN5 active site (Faull, Lau et al. 2019). Our structural data support and extend this model, showing that although NEDD8 facilitates CSN5 activation indirectly, it is the interaction between CSN5 and the CUL1^{WHB} domain that directly drives remodelling of the active site, initiating a two-step mechanism.*

- **Please provide marker in Extended Data Fig1a and Fig17a.**

We have now added molecular weight markers to Extended Data Fig. 1a and Extended Data Fig. 22a (old 17a). We hope these additions help improve the clarity and interpretability of the results.

• Given that the dimensions of the protein assemblies described in this paper are approximately 150 × 170 angstroms, please confirm whether the scale bars in Extended Data Fig. 1c and Extended Data Fig. 17 are accurate.

We thank the reviewer for drawing our attention to this point. The 2D class averages shown in Extended Data Fig. 1c and Extended Data Fig. 17. were generated from images that had been two-fold binned (pixel size = 2.16 Å, not the unbinned 1.08 Å). Consequently, the original 50 Å scale bars were incorrect. We have replaced them with the correct 100 Å scale bars (Extended Data Fig. 1c and 22c). We apologise for the oversight.

• Please indicate the corresponding figures between line 252-263. It is easier to follow the text and figures.

We have now added references to the corresponding figures on page 10, lines 286–301 to improve clarity and help guide the reader through the results more effectively. These changes are reflected in the revised manuscript.

“Structurally, the RBX1^{RING} consists of two zinc-chelating loops (RBX1^{RING-loop1} and RBX1^{RING-loop2}), an α -helix (RBX1^{RING-helix3}) and a three-stranded β -sheet, alongside an insertion motif (RBX1^{RING-insertion}: residues 50–70) stabilised by a third zinc ion (Extended Data Fig. 8f). At the CSN2-RBX1^{RING} interface, a shallow groove in the RBX1^{RING} accommodates CSN2^{arm}. This groove, predominantly formed by the RBX1^{RING-helix3}, RBX1^{RING-loop1} and RBX1^{RING-loop2}, features residues RBX1^{A43}, RBX1^{I44}, RBX1^{F79}, RBX1^{W87}, RBX1^{P95} and RBX1^{L96}, which create a hydrophobic environment for CSN2^{I291} (Fig. 1e). Additional stability comes from two salt bridges, CSN2^{E260}-RBX1^{R99} and CSN2^{D298}-RBX1^{H77}, along with a hydrogen bond between the backbone carbonyl group of CSN2^{G290} and side chain RBX1^{R46} (Fig. 1e). On the opposite side of the clamp, CSN4^{arm} links to RBX1 via RBX1^{RING-loop1} and the

RBX1^{RING-insertion} motif, forming further stabilising interactions. The binding surface is predominantly hydrophobic, with residues RBX1^{V38}, RBX1^{V39}, RBX1^{I49} and RBX1^{M50} creating a pocket for CSN4^{L173} (Fig. 1f). Additionally, the imidazole ring of RBX1^{H48} engages in a hydrogen bond with the side chain of CSN4^{N169}, further strengthening the interface (Fig. 1f)."

- **Extended Data Fig.8n “Rbx1” should be “RBX1”; Fig 2J, “Csn5” should be “CSN5”; ED Fig2c, “csn8” should be “CSN8”.**

Thank you for pointing this out. We have corrected the capitalisation of all protein names in the corresponding figures to maintain consistency with standard nomenclature. The revised figures have been updated in the revised manuscript as new Extended Data Fig. 12j, Extended Data Fig.4c.

- **The figure legend for Extended Data Fig 7e does not match the label in the figure.**

We have corrected the figure legend to ensure consistency with the figure panel, as shown in the revised manuscript (new Extended Data 11e).

- **In Extended Data Fig. 8d and 8j, the reported affinities should be similar, as the M117R mutation was specifically designed to disrupt the CSN5-NEDD8 interaction. Since non-neddylated protein is used in this context, the mutation is not expected to influence the affinity. Please clarify or address this discrepancy.**

We agree that the CSN^{5M117R} mutation was originally designed to disrupt the CSN5–NEDD8 interface within the CSN-^{N8}SCF complex and indeed resulted in a 12-fold reduction in binding affinity, as determined by our original, steady-state SPR data analysis. Interestingly, this mutation also led to a ~3-fold decrease in binding affinity to non-neddylated CUL/RBX1 ($K_d = 760$ nM for CSN^{5H138A/M117R} vs. 260 nM for CSN^{5H138A}). While this reduction is relatively modest, we further investigated its impact by reanalysing existing SPR datasets for this mutant and performing

additional kinetic measurements, now included in Extended Data Fig. 5i and 14b (new). Kinetic analysis revealed that CSN^{5H138A/M117R} had no significant effect on the dissociation rate (k_{off}) but showed a ~3-fold decrease in the association rate (k_{on}) with non-neddylated CUL1/RBX1, consistent with its reduced affinity (Table 2 and Extended Data Fig. 5i and 14b). A comparable reduction in k_{on} rate was also observed for experiments performed with neddylated CUL1/RBX1, suggesting that the M117R mutation may induce subtle conformational changes that influence initial complex formation, even in the absence of NEDD8. Notably, the most pronounced effect of CSN^{5H138A/M117R} was observed in the k_{off} when NEDD8 was present, supporting the importance of the CSN5-NEDD8 interface in stabilising the neddylated complex.

CSN	CUL1/RBX1	K _d (nM)		k _{on} (10 ⁶ M ⁻¹ s ⁻¹)	k _{off} (s ⁻¹)
		steady-state	K _d (nM) kinetics		
5H138A	WT	260 ± 30	220 ± 90	0.74 ± 0.04	0.16 ± 0.06
5H138A/M117R	WT	760 ± 20	760 ± 20	0.19 ± 0.02	0.14 ± 0.02

CSN	N ⁸ CUL1/RBX1	K _d (nM) steady-	K _d (nM)	k _{on} (10 ⁶ M ⁻¹ s ⁻¹)	k _{off} (s ⁻¹)
		state	kinetics		
					0.00097 ±
5H138A	WT	10 ± 3	2.6 ± 0.04	0.37 ± 0.2	0.0004
5H138A/M117R	WT	120 ± 9	80 ± 30	0.095 ± 0.02	0.0074 ± 0.003

We have now clarified this point in the revised text.

Page12-13, lines 383-387: *Kinetic SPR analysis revealed that the CSN^{5H138A/M117R} mutant exhibited an approximately ~30-fold reduction in binding affinity for neddylated CUL1/RBX1, resulting from a ~4-fold decrease in the k_{on} rate and an ~8-fold increase in the k_{off} rate, relative to CSN^{5H138A}. Interestingly, the same mutation also led to a relatively modest ~3-fold reduction in binding affinity for non-neddylated CUL1/RBX1, resulting from a ~3-fold slower association rate (Table 2-3, Extended Data Fig. 12e-f, Extended Data Fig. 14b, Extended Data Fig. 5g-i).*

Page13, lines 400-403: *Second, although disrupting the CSN5-NEDD8 I44 interface through the CSN^{5M117R} mutation severely impaired binding and catalysis, the modest impact on binding to the non-neddylated substrate suggests that CSN^{5M117} may also influence complex formation independently of NEDD8.*

• **Line 343-344, while CSN5 E122 does not directly mediate NEDD8 binding, it likely plays a crucial role in stabilizing NEDD8 for efficient catalysis, possibly through interactions with NEDD8 H68. This sentence is confusing. Please revise.**

We have revised the sentence in the main text for clarity. The revised version reads:

Page 12, lines 372-379: *To elucidate a functional role for CSN5^{E122}, we performed kinetic analysis of the SPR data (Table 2, and Extended Data Fig. 14a, 5c). While the association rate constant (k_{on}) to neddylated CUL1/RBX1 was similar to that of CSN^{5H138A}, the dissociation rate constant (k_{off}) was accelerated nearly 6-fold, indicating that CSN5^{E122} is not directly involved in substrate recognition, but instead contributes to stabilising the enzyme-substrate complex, possibly through an interaction with NEDD8^{H68}. The weakened complex stability observed in this mutant likely accounts for its reduced deneddylation activity.*

Reviewer #2 (Remarks to the Author):

Reviewer #3 (Remarks to the Author):

Ding and colleagues present a comprehensive structural analysis of the COP9 signalosome (CSN) interaction with its substrate, the SCF ubiquitin ligase complex. Using cryo-electron microscopy, the authors have captured multiple states of this molecular machinery, including pre-activated, activated, and four distinct dissociation intermediates. Their findings provide unprecedented

insight into the mechanisms of CSN-mediated deneddylation, including the first structural visualization of CSNAP, a previously uncharacterized component of the CSN complex. While the structural work is a real tour de force, I have several significant concerns about the data interpretation and experimental validation, but I would still recommend publication once they are addressed

We thank the reviewer for the thoughtful and constructive summary of our work. We truly appreciate these comments, which have been valuable in guiding our revisions and improving the clarity and completeness of the manuscript. All points have been carefully addressed in the detailed responses below.

Major Concerns:

1. The authors designate structures as "pre-activated," "activated," and "dissociation" states with insufficient experimental evidence. The assignment of these states to a sequential reaction coordinate is largely speculative and not supported by kinetic data. The authors provide no compelling evidence that these aren't simply different conformations of the complex that exist in equilibrium or, in some cases, potentially damaged particles. Without kinetic

experiments or complementary biophysical approaches, the sequential nature of these states remains unproven, undermining the mechanistic model.

We agree that definitive validation for a stepwise reaction sequence would require additional, complementary biophysical approaches, including time-resolved cryo-EM or kinetic trapping. These approaches are beyond the scope of the present study but remain active goals for future investigation, as acknowledged in the original manuscript (see Discussion, page24 lines 749-764).

Our use of the terms “pre-activated”, “activated” and “dissociation” is not meant to imply a fully resolved or experimentally validated reaction pathway. Rather, these designations are intended to provide a mechanistic framework based on the structural snapshots we obtained, interpreted within their defined biochemical contexts. Each structure was assembled under specific biochemical conditions designed to favour distinct functional states, allowing us to derive a plausible, hypothesis-driven model for understanding CSN-CRL regulation.

While we acknowledge that our interpretations are model based, we believe that the biochemical context in which these complexes were assembled, coupled with the structural features observed at high resolution, lends strong support to our proposed descriptions. In response to the reviewer’s point 1, we have now revised the manuscript in the following ways:

Activated state:

To enable structural studies of the active CSN-^{N8}SCF complex, we reconstituted the 9-subunit CSN^{5H138A} enzyme with its neddylated SCF substrate. The H138A mutation in the CSN5 active site abolishes deneddylation activity (Cope, Suh et al. 2002) (page 4, lines 120-121) and is a well-established structural tool to trap the activated transition state of CSN by stably retaining NEDD8 over the time scale of an experiment (Enchev, Scott et al. 2012, Cavadini, Fischer et al. 2016, Mosadeghi, Reichermeier et al. 2016, Faull, Lau et al. 2019, Hu, Zhang et al. 2024). In the resulting CSN^{5H138A}-^{N8}SCF complex, NEDD8 is positioned within the CSN5 active

site, with its isopeptide bond located ~ 3 Å from the catalytic zinc-coordinating residues (lines 500-502, fig.2d, right panel). This geometry is consistent with a catalytically poised conformation, as observed in other MPN+ enzymes such as RPN11 and AMSH-LP (lines 486-491). The use of the catalytically impaired CSN^{5H138A} mutant supports the conclusion that our complex represents a trapped, activated intermediate (see Extended Data Fig. 19).

Pre-activated state:

The structure designated as pre-activated CSN^{5H138A_N8}SCF reveals distinct structural hallmarks of activation, including a clear remodelling of the CSN5 Ins-1 loop away from the catalytic site and a coordinated engagement of CUL1^{WHB} and conjugated NEDD8 with CSN5. These features suggest that the substrate is properly aligned in readiness for catalysis, consistent with a trajectory towards full enzymatic activation. While we do not claim this represents the sole intermediate along the activation pathway, we chose the term “pre-activated” as a general term to reflect its preparatory conformation (see Results, Fig. 2c-d left panel).

Dissociation states:

The four CSN-SCF dissociation intermediates were obtained from a complex assembled with the CSN^{5E104A} mutant, previously characterised by us (Mosadeghi, Reichermeier et al. 2016). This mutation is known to inhibit product release, resulting in a product-inhibited form of the enzyme. Specifically, CSN^{5E104A} binds deneddylated CRL, the reaction product, with significantly higher affinity than wild-type CSN ($K_d = 26$ nM vs. 310 nM) and exhibits a markedly slower dissociation rate ($k_{off} = 0.13$ s⁻¹ vs. 1.1 s⁻¹ for wild type; see Figure 3C in Mosadeghi, Reichermeier *et al.*, 2016). To aid the reader, we have included additional details of this kinetic work within the text.

Page18, lines 547-550: *Specifically, CSN^{5E104A} binds deneddylated CRL, the reaction product, with more than 10-fold higher affinity than wild-type CSN and exhibits an 8-fold reduction in the product dissociation rate (Mosadeghi, Reichermeier et al. 2016).*

These kinetic findings strongly support the interpretation that the resolved conformations in our cryo-EM dataset correspond to post-catalytic complexes that are stalled along the dissociation pathway. All four states were resolved at reasonably high resolution and display distinct structural differences with a trajectory toward complex disassembly. Based on the biochemical context of the CSN^{5E104A} mutant and the structural features of the maps, we believe this evidence provides a strong rationale for interpreting these as dissociation intermediates, but with appropriate caveats clearly stated. Therefore, we explicitly acknowledge in the revised manuscript that alternative interpretations are possible, namely that some of these states could reflect early assembly intermediates rather than post-catalytic ones. We have revised the Discussion accordingly to present a more balanced and transparent interpretation as below:

Discussion, page 24, lines 755–764: *While we favour the interpretation that these structures correspond to sequential dissociation intermediates, we explicitly acknowledge that some may additionally reflect early association states, given the inherent reversibility of protein-protein interactions. Nonetheless, the defined assembly conditions, high-resolution features, and specificity of the CSN-SCF interactions observed in the absence of NEDD8 support their classification as bona fide dissociation intermediates. Ultimately, complementary approaches such as time-resolved cryo-EM (Maeots, Lee et al. 2020, Maeots and Enchev 2022, Märt-Erik Mäeots and Rodriguez Molina 2025) will be essential to fully elucidate the temporal dynamics of the CSN–CRL regulatory cycle.*

We hope these clarifications address the reviewer's concerns and reinforce that while the sequence of states remains a model, it is grounded in a combination of structural, biochemical, and mechanistic reasoning.

2. Despite providing detailed structural information about CSNAP's integration into the CSN complex, the authors present no functional data whatsoever.

Critical questions remain entirely unaddressed:

Does CSNAP affect deneddylation activity? Does it alter binding kinetics?

Does it influence complex stability? A basic comparison of 8-subunit versus 9-

subunit CSN would provide essential functional context for the structural observations.

These are indeed important points that have attracted the attention of the field and been the subject of previous work. Since the identification of CSNAP as the ninth subunit of CSN (Rozen, Fuzesi-Levi et al. 2015), its biochemical role has been systematically studied. *Füzesi-Levi et al. (2020)*, in collaboration with our laboratory, conducted extensive *in vitro* and cellular analysis demonstrating that CSNAP does not impact CSN's deneddylation activity (Fuzesi-Levi, Fainer et al. 2020). Based on those findings, we did not pursue additional activity assays related to CSNAP in the current study.

However, CSNAP was previously reported to reduce the binding affinity of CSN for both neddylated and non-neddylated CUL1/RBX1 (Fuzesi-Levi, Fainer et al. 2020). To further examine the effect of CSNAP on substrate binding, we performed detailed SPR kinetic analysis comparing CSN^{5H138A} complexes with and without CSNAP (referred to as 9CSN^{5H138A} and 8CSN^{5H138A}, respectively) against both forms of CUL1/RBX1. The results are presented in Tables 2-3 and Extended Data Fig. 5a-h.

CSN	^{N8} CUL1/RBX1	K _d (nM) steady-state	K _d (nM) kinetics	k _{on} (10 ⁶ M ⁻¹ s ⁻¹)	k _{off} (s ⁻¹)
8CSN5H138A	WT	6 ± 3	1.6 ± 0.3	1.0 ± 0.4	0.0016 ± 0.0005
9CSN5H138A	WT	10 ± 3	2.6 ± 0.04	0.37 ± 0.2	0.00097 ± 0.0004

CSN	CUL1/RBX1	K _d (nM) steady-state	K _d (nM) kinetics	k _{on} (10 ⁶ M ⁻¹ s ⁻¹)	k _{off} (s ⁻¹)
8CSN5H138A	WT	90 ± 8	77 ± 20	0.62 ± 0.3	0.044 ± 0.01
9CSN5H138A	WT	260 ± 30	220 ± 90	0.74 ± 0.04	0.16 ± 0.06

Our data indicate that CSNAP modulates the binding kinetics and stability of CSN-substrate interactions. Consistent with previous findings (Fuzesi-Levi, Fainer et al.

2020), incorporation of CSNAP into the CSN complex reduces binding affinity for both neddylated and non-neddylated CUL1/RBX1. For the neddylated substrate, 9CSN^{5H138A} exhibits a slightly slower association rate ($k_{on} = 0.37 \times 10^6 \text{ M}^{-1}\text{s}^{-1}$) compared to 8CSN^{5H138A} ($k_{on} = 1.0 \times 10^6 \text{ M}^{-1}\text{s}^{-1}$), while the dissociation rate remains largely unchanged. Conversely, for the non-neddylated substrate, the association rates are similar between the two complexes, but the dissociation rate is significantly higher for 9CSN^{5H138A} ($k_{off} = 0.16 \text{ s}^{-1}$) compared to 8CSN^{5H138A} ($k_{off} = 0.04 \text{ s}^{-1}$).

These kinetic findings suggest that CSNAP reduces overall affinity for CRL substrates by modulating complex stability, specifically, by decreasing the association rate for neddylated CRLs and increasing the dissociation rate for non-neddylated CRLs. Importantly, this decreased binding affinity does not compromise the structural integrity of the complex: all seven cryo-EM reconstructions of 9-subunit CSN reveal well-defined density for CSNAP, indicating stable incorporation in both neddylated and non-neddylated complexes. Furthermore, removal of CSNAP does not disrupt the integrity of the remaining eight CSN subunits, as evidenced by both cryo-EM analysis and biochemical purification profiles (Lingaraju, Bunker et al. 2014).

Notably, our ability to obtain seven near-atomic resolution cryo-EM maps may, in part, reflect CSNAP's stabilising effect on the overall rigidity of the CSN complex, offering a potential advantage over previous structural studies of CSN lacking CSNAP (Enchev, Scott et al. 2012, Cavadini, Fischer et al. 2016, Mosadeghi, Reichermeier et al. 2016, Faull, Lau et al. 2019, Hu, Zhang et al. 2024).

Although CSNAP does not directly participate in catalysis, our data support its role as a regulatory subunit that fine-tunes the dynamics of CSN-substrate interactions. These findings have been incorporated into the revised manuscript and supporting figures as below.

Page 7, Lines 203-217: *To investigate the effect of CSNAP on binding, we performed surface plasmon resonance (SPR) kinetic measurements comparing CSN^{5H138A} complexes with and without CSNAP (referred to as 9CSN^{5H138A} and 8CSN^{5H138A}, respectively) with both neddylated and non-neddylated CUL1/RBX1 (Table 2 and Extended Data Fig. 5a-h). Consistent with prior findings (Fuzesi-Levi,*

Fainer et al. 2020), incorporation of CSNAP into the CSN complex reduces the binding affinity for both neddylated and non-neddylated CUL1/RBX1. For the neddylated substrate, 9CSN^{5H138A} exhibits a slower association rate ($k_{on} = 0.37 \times 10^6 M^{-1}s^{-1}$) compared to 8CSN^{5H138A} ($k_{on} = 1.0 \times 10^6 M^{-1}s^{-1}$), while the dissociation rate remains largely unchanged. Conversely, for the non-neddylated substrate, the association rates are similar between the two complexes, but the dissociation rate is significantly higher for 9CSN^{5H138A} ($k_{off} = 0.16 s^{-1}$) compared to 8CSN^{5H138A} ($k_{off} = 0.04 s^{-1}$). These kinetic findings suggest that CSNAP reduces overall affinity for CRL substrates by modulating complex stability, specifically, by decreasing the association rate for neddylated CRLs and increasing the dissociation rate for non-neddylated CRLs.

Page 8, Lines 232-238: Our complementary kinetics analysis indicates that CSNAP modulates CSN -substrate interaction dynamics. Specifically, incorporation of CSNAP into the CSN complex reduces the association rate with neddylated CUL1/RBX1 and increases the dissociation rate from non-neddylated CUL1/RBX1. These findings support a model in which CSN acts as a regulatory subunit that fine-tunes substrate binding and release without directly contributing to deneddylation activity.

Extended Data Fig. 5. Comparative SPR analysis of CSN variants binding to neddylated and non-neddylated CUL1/RBX1

a, c, e, g, SPR sensorgrams showing binding of CSN^{5H138A} variants (with or without CSNAP) to immobilised StrepII^{2X}-tagged neddylated (**a, c**) or non-neddylated (**e, g**) CUL1/RBX1. Sensorgrams were analysed by global fitting using either a 1:1 Langmuir binding model (**a, c**) or a 1:1 binding model with drift correction (**e, g**), to

extract association (k_{on}) and dissociation (k_{off}) rate constants and calculate kinetic K_d values (black = experimental; red = fit). **b, d, f, h**, Corresponding steady-state binding curves derived from (**a, c, e, g**). Data were analysed by fitting a hyperbolic one-site binding model to determine steady-state K_d . For panels **b** and **f**, responses were normalised to the R_{max} value to account for variability in immobilisation levels across triplicates and for the graph adjusted to the median R_{max} . Raw responses were used in panels **d** and **h**. **i**, SPR sensorgrams showing binding of CSN^{5H138A/M117R} to immobilised *Strept12x*CUL1/RBX1. Sensorgrams were analysed by global fitting using either a 1:1 binding model with drift correction, to extract association (k_{on}) and dissociation (k_{off}) rate constants and calculate kinetic K_d values (black = experimental; red = fit). All the data represent mean \pm SD from three independent experiments.

To investigate whether CSNAP influences the structure of the CSN complex, we compared our 9-subunit CSN^{apo} structure with the previously determined 8-subunit CSN crystal structure (PDB: 4D10) (Lingaraju, Bunker et al. 2014), as detailed in lines 783-802 and Extended Data Fig. 33 (formerly Extended Data Fig. 27). Subtle conformational differences are observed in regions near the CSN3–CSN8 interface, where CSNAP is located, as well as within the PCI Ring. These local shifts suggest that CSNAP may modulate structural dynamics in this region, although the resolution of both structures limits definitive interpretation (Extended Data Fig. 33b-c).

Together with our kinetic data, this structural comparison supports the conclusion that CSNAP does not impact the core assembly of the CSN complex. Rather, it acts as a regulatory component modulating substrate engagement. These insights add important functional context to the role of CSNAP within the holo-CSN complex.

3. The cryo-EM maps show clear signs of preferred orientation, as evident from the angular distribution plots. While DeepEMhancer has been used for post-processing, the potential artifacts introduced by preferred orientation are hidden to some extent But they should still be adequately acknowledged or addressed. A 3D FSC analysis would help quantify anisotropy in the maps.

Preferred orientations have been a well-documented challenge across all reported cryo-EM studies of CSN-CRLs. We acknowledge that also in this work the angular distribution plots indicate evidence of preferred particle orientation in our cryo-EM datasets. To rigorously assess the impact of this anisotropy on map quality, as suggested by the reviewer, we performed 3D Fourier Shell Correlation (3D FSC) analysis using the Remote 3DFSC Processing Server (<https://3dfsc.salk.edu/upload/>) (Tan, Baldwin et al. 2017).

The resulting plots are now included in Extended Data Figs. 1f, i, and 22g, j, m, p, s, respectively. This analysis confirms the presence of moderate anisotropy, but with overall sphericity values within acceptable limits. Importantly, no severe directional artefacts were observed that would compromise interpretation. Additionally, local resolution estimates and direct visual inspection of the density maps in critical regions, such as the CSN5 active site and the CSN-NEDD8 interface, demonstrate that these areas are well-resolved and suitable for reliable model building.

We have now acknowledged this limitation and included the 3D FSC results in the Extended Data Figs.1 and 22.

Extended Data Fig.1. Reconstitution, cryo-electron microscopy, and single particle analysis of CSN^{5H138A_N8}SCF.

e, Euler angle distribution plots of pre-activated CSN^{5H138A_N8}SCF complex. **f**, Directional 3DFSC plots of pre-activated CSN^{5H138A_N8}SCF complex (Tan, Baldwin et al. 2017). **g**, Local resolution estimates of pre-activated CSN^{5H138A_N8}SCF complex. **h**, Euler angle distribution plots of activated CSN^{5H138A_N8}SCF complex. **i**, Directional 3DFSC plots of activated CSN^{5H138A_N8}SCF complex (Tan, Baldwin et al. 2017). **j**, Local resolution estimates of activated CSN^{5H138A_N8}SCF complex.

Extended Data Fig.22. Cryo-EM and single particle analysis on CSN^{5E104A}-SCF dissociation states.

e, Resolution estimates of maps resolved from the CSN^{E104A}-SCF dataset. *f*, Euler angle distribution plots for dissociation-state-1. *g*, Directional 3DFSC plots of dissociation-state-1. *h*, Local resolution estimates for dissociation-state-1. *i*, Euler angle distribution plots for dissociation-state-2. *j*, Directional 3DFSC plots of dissociation-state-2. *k*, Local resolution estimates of dissociation-state-2. *l*, Euler angle distribution plots for dissociation-state-3. *m*, Directional 3DFSC plots of dissociation-state-3. *n*, Local resolution estimates of dissociation-state-3. *o*, Euler angle distribution plots of dissociation-state-4. *p*, Directional 3DFSC plots of dissociation-state-4. *q*, Local resolution estimates of dissociation-state-4. *r*, Euler angle distribution plots of free CSN (CSN^{Apo}). *s*, Directional 3DFSC plots of free CSN (CSN^{Apo}). *t*, Local resolution estimates of free CSN (CSN^{Apo}). Resolutions for all maps in this figure were estimated using the gold-standard FSC 0.143 criterion.

Moreover, the local resolution estimation appears overly optimistic, particularly in peripherally located regions where resolution is likely significantly lower than 5 Å as indicated.

In response, we have recalculated the local resolution using an updated scale bar to ensure a more accurate estimation. The revised local resolution maps are now included in Extended Data Fig. 1g, j.

4. The methods section lacks critical details regarding image processing decisions. The implementation of cryoDRGN is inadequately described - specifically,

how particles were selected for different classes,

We have revised the Methods section to clarify the cryoDRGN workflow, as well as the figure legend for Extended Data Fig. 1d.

Page 36, Lines 995-997: The cryoDRGN-generated maps informed the design of local masks centred around the NEDD8 region. Focused classification was done with RELION-4.0.

Page 2-3 in SuppFigs, Lines 8-16: Extended Data Fig. 1c: Single particle analysis workflow for the CSN-^{N8}SCF dataset, resulting in 3D reconstructions of pre- and activated CSN^{5H138A-N8}SCF. CryoDRGN-based heterogeneity analysis is shown in the inset. Left: UMAP projection of the latent space. Middle: Representative cryoDRGN-generated structures at indicated UMAP coordinates. Right: Overlay of structures 3 and 8 highlighting distinct NEDD8 positions (circled in red). Densities from structures 3 and 8 were combined to create a mask for 3D local classification. 3D local classification was performed in RELION-4.0 (Kimanius, Dong et al. 2021) utilising a NEDD8-specific mask (generated from classification results from CryoDRGN (Zhong, Bepler et al. 2021)).

Please note that we did not use cryoDRGN for direct particle classification. Instead, we employed cryoDRGN to explore conformational heterogeneity within the dataset, particularly focusing on the NEDD8 moiety, which exhibited the greatest degree of structural flexibility. The cryoDRGN-generated maps informed the design of local masks in RELION-4.0 centred around the NEDD8 region.

These masks were subsequently applied in focused 3D classification using RELION. Particle subsets were then selected from the resulting classes based on improvements in local resolution within the CSN5/CSN6–NEDD8 region.

what criteria determined inclusion or exclusion,

The primary criterion guiding particle inclusion or exclusion was the local map quality of the NEDD8-CSN5 catalytic interface, which represents the core functional region relevant to our mechanistic studies. Our classification strategy focused on exploring conformational variability in the vicinity of NEDD8 and the CSN5 active site.

Therefore, particle selection was not based on global conformational differences of the CRL scaffold, but rather on the quality of local resolution at the CSN5/CSN6–NEDD8 interface.

And whether additional conformational states might exist in the dataset but were not pursued.

During our analysis, we did observe a small subset of particles that appeared to correspond to alternative conformational states. However, these were represented at low resolution, likely due to high intrinsic flexibility and limited particle numbers. Given their poor interpretability and relatively low representation within the dataset, we indeed did not pursue further structural analysis of these states. We have now clarified this rationale in the revised Methods section to ensure transparency in our analysis strategy.

What about the other clusters? These details are essential for evaluating the robustness of the structural classifications.

The additional clusters identified through cryoDRGN analysis primarily correspond to conformational variability within the CUL1–SKP1–SKP2 region, which is known to exhibit flexibility relative to the CSN core (as reported for instance in Mosadeghi et al. 2016). In addition, we observed a small number of clusters corresponding to minor particle subsets, which may represent alternative states. However, again, due to low particle numbers and insufficient resolution, we did not pursue further classification of these states.

Our focused analysis prioritised clusters that provided the highest resolution and most reliable density in the CSN5-NEDD8 region.

5. The modelling of the CSN5Ins-1 loop in the pre-activated state is problematic. This region shows fragmented density yet serves as the basis for significant mechanistic interpretations. Given the central importance of this loop to the authors' mechanistic arguments, a more rigorous approach to

modelling this region is required, with explicit acknowledgement of the limitations imposed by map quality.

We fully acknowledge the limitations in modelling the CSN5^{Ins-1} loop due to its intrinsic flexibility and the corresponding weak density in our cryo-EM maps. In the original manuscript (lines 439-441), we stated, *“Due to the inherent flexibility of the CSN5^{Ins-1} region, we were unable to accurately model all side chains within the loop (Extended Data Fig. 13a).”*

In response to the reviewer’s suggestion for a more rigorous approach to modelling this region, we have now quantified the local resolution estimation of CSN5^{Ins-1} and calculated Q-scores for each residue in both pre-activated and activated CSN^{5H138A-N⁸SCF} structures. These data are presented in Extended Data Fig. 18 (previously Extended Data Fig. 13) and help to illustrate the resolution constraints that informed our model building. Side chains such as CSN5^{E101} and CSN5^{E104} were modelled conservatively as approximations, and this is explicitly stated in the figure legend.

Given the mechanistic significance of this loop, we complemented our structural studies with functional validation. Specifically, we performed a mutagenesis experiment targeting CUL1^{R741E}, designed based on our interpretation of the CSN5^{Ins-1} interaction interface observed in the pre-activated CSN^{5H138A-N⁸SCF} complex. As shown in Extended Data Fig. 13d (new), the CUL1^{R741E} mutation led to a marked reduction in deneddylation activity compared with wild-type^{N⁸CUL1/RBX1}, supporting the functional relevance of our proposed model despite local resolution limitations.

Extended Data Fig. 18. Structural versatility of CSN5^{Ins-1}.

a, Local resolution estimation of CSN5^{Ins-1} and CUL1^{R741}, CUL1^{R717} in pre-activated CSN^{5H138A-N8} SCF. **b**, Cryo-EM density and molecular model of CSN5^{Ins-1}, as well as CUL1^{R741} and CUL1^{R717} in pre-activated CSN^{5H138A-N8} SCF. The side chains of CSN5^{E101} and CSN5^{E104} are modelled as approximations due to density limitations. For validation, the Q-score (Pintilie, Zhang et al. 2020) is calculated for the backbone and quoted under each residue. **c**, Local resolution estimation of CSN5^{Ins-1} in activated CSN^{5H138A-N8} SCF. **d**, Cryo-EM density and molecular model of CSN5^{Ins-1} in activated CSN^{5H138A-N8} SCF. **e**, Structural comparison of CSN5^{Ins-1} from pre-activated and activated CSN^{5H138A-N8} SCF, alongside the isolated crystal structure of CSN5 (PDB: 4F70), demonstrating the conformational adaptability of this loop.

Adams, P. D., P. V. Afonine, G. Bunkoczi, V. B. Chen, I. W. Davis, N. Echols, J. J. Headd, L. W. Hung, G. J. Kapral, R. W. Grosse-Kunstleve, A. J. McCoy, N. W. Moriarty, R. Oeffner, R. J. Read, D. C. Richardson, J. S. Richardson, T. C. Terwilliger and P. H. Zwart (2010). "PHENIX: a comprehensive Python-based system for macromolecular structure solution." Acta Crystallogr D Biol Crystallogr **66**(Pt 2): 213-221.

Baek, K., D. T. Krist, J. R. Prabu, S. Hill, M. Klugel, L. M. Neumaier, S. von Gronau, G. Kleiger and B. A. Schulman (2020). "NEDD8 nucleates a multivalent cullin-RING-UBE2D ubiquitin ligation assembly." Nature **578**(7795): 461-466.

Cavadini, S., E. S. Fischer, R. D. Bunker, A. Potenza, G. M. Lingaraju, K. N. Goldie, W. I. Mohamed, M. Faty, G. Petzold, R. E. Beckwith, R. B. Tichkule, U. Hassiepen, W. Abdulrahman, R. S. Pantelic, S. Matsumoto, K. Sugasawa, H. Stahlberg and N. H. Thoma (2016). "Cullin-RING ubiquitin E3 ligase regulation by the COP9 signalosome." Nature **531**(7596): 598-603.

Cope, G. A., G. S. Suh, L. Aravind, S. E. Schwarz, S. L. Zipursky, E. V. Koonin and R. J. Deshaies (2002). "Role of predicted metalloprotease motif of Jab1/Csn5 in cleavage of Nedd8 from Cul1." Science **298**(5593): 608-611.

Enchev, R. I., D. C. Scott, P. C. da Fonseca, A. Schreiber, J. K. Monda, B. A. Schulman, M. Peter and E. P. Morris (2012). "Structural basis for a reciprocal regulation between SCF and CSN." Cell Rep **2**(3): 616-627.

Faull, S. V., A. M. C. Lau, C. Martens, Z. Ahdash, K. Hansen, H. Yebenes, C. Schmidt, F. Beuron, N. B. Cronin, E. P. Morris and A. Politis (2019). "Structural basis of Cullin 2 RING E3 ligase regulation by the COP9 signalosome." Nat Commun **10**(1): 3814.

Fuzesi-Levi, M. G., I. Fainer, R. Ivanov Enchev, G. Ben-Nissan, Y. Levin, M. Kupervaser, G. Friedlander, T. M. Salame, R. Nevo, M. Peter and M. Sharon (2020). "CSNAP, the smallest CSN subunit, modulates proteostasis through cullin-RING ubiquitin ligases." Cell Death Differ **27**(3): 984-998.

Hao, B., N. Zheng, B. A. Schulman, G. Wu, J. J. Miller, M. Pagano and N. P. Pavletich (2005). "Structural basis of the Cks1-dependent recognition of p27(Kip1) by the SCF(Skp2) ubiquitin ligase." Mol Cell **20**(1): 9-19.

Hu, Y., Z. Zhang, Q. Mao, X. Zhang, A. Hao, Y. Xun, Y. Wang, L. Han, W. Zhan, Q. Liu, Y. Yin, C. Peng, E. M. Y. Moresco, Z. Chen, B. Beutler and L. Sun (2024). "Dynamic molecular architecture and substrate recruitment of cullin3-RING E3 ligase CRL3(KBTBD2)." Nat Struct Mol Biol **31**(2): 336-350.

Kimanius, D., L. Dong, G. Sharov, T. Nakane and S. H. W. Scheres (2021). "New tools for automated cryo-EM single-particle analysis in RELION-4.0." Biochem J **478**(24): 4169-4185.

Lingaraju, G. M., R. D. Bunker, S. Cavadini, D. Hess, U. Hassiepen, M. Renatus, E. S. Fischer and N. H. Thoma (2014). "Crystal structure of the human COP9 signalosome." Nature **512**(7513): 161-165.

Maeots, M. E. and R. I. Enchev (2022). "Structural dynamics: review of time-resolved cryo-EM." Acta Crystallogr D Struct Biol **78**(Pt 8): 927-935.

Maeots, M. E., B. Lee, A. Nans, S. G. Jeong, M. M. N. Esfahani, S. Ding, D. J. Smith, C. S. Lee, S. S. Lee, M. Peter and R. I. Enchev (2020). "Modular microfluidics enables kinetic insight from time-resolved cryo-EM." Nat Commun **11**(1): 3465.

Märt-Erik Mäeots, S. T., Mohammad M. N. Esfahani, Juan B. and J. A. C. Rodriguez Molina, Aran Amin, Albane Imbert, Radoslav I. Enchev (2025). "Chronobot: Deep learning guided time-resolved cryo-EM

captures molecular choreography of RecA in homology search." bioRxiv preprint.

Mosadeghi, R., K. M. Reichermeier, M. Winkler, A. Schreiber, J. M. Reitsma, Y. Zhang, F. Stengel, J. Cao, M. Kim, M. J. Sweredoski, S. Hess, A. Leitner, R. Aebersold, M. Peter, R. J. Deshaies and R. I. Enchev (2016). "Structural and kinetic analysis of the COP9-Signalosome activation and the cullin-RING ubiquitin ligase deneddylation cycle." Elife **5**.

Pintilie, G., K. Zhang, Z. Su, S. Li, M. F. Schmid and W. Chiu (2020). "Measurement of atom resolvability in cryo-EM maps with Q-scores." Nat Methods **17**(3): 328-334.

Rozen, S., M. G. Fuzesi-Levi, G. Ben-Nissan, L. Mizrachi, A. Gabashvili, Y. Levin, S. Ben-Dor, M. Eisenstein and M. Sharon (2015). "CSNAP Is a Stoichiometric Subunit of the COP9 Signalosome." Cell Rep **13**(3): 585-598.

Tan, Y. Z., P. R. Baldwin, J. H. Davis, J. R. Williamson, C. S. Potter, B. Carragher and D. Lyumkis (2017). "Addressing preferred specimen orientation in single-particle cryo-EM through tilting." Nat Methods **14**(8): 793-796.

Zhong, E. D., T. Bepler, B. Berger and J. H. Davis (2021). "CryoDRGN: reconstruction of heterogeneous cryo-EM structures using neural networks." Nat Methods **18**(2): 176-185.